# Scalable and Certifiable Graph Unlearning: Overcoming the Approximation Error Barrier

**Lu Yi & Zhewei Wei***
Renmin University of China
{yilu, zhewei}@ruc.edu.cn

## Abstract

Graph unlearning has emerged as a pivotal research area for ensuring privacy protection, given the widespread adoption of Graph Neural Networks (GNNs) in applications involving sensitive user data. Among existing studies, certified graph unlearning is distinguished by providing robust privacy guarantees. However, current certified graph unlearning methods are impractical for large-scale graphs because they necessitate the costly re-computation of graph propagation for each unlearning request. Although numerous scalable techniques have been developed to accelerate graph propagation for GNNs, their integration into certified graph unlearning remains uncertain as these scalable approaches introduce approximation errors into node embeddings. In contrast, certified graph unlearning demands bounded model error on exact node embeddings to maintain its certified guarantee.

To address this challenge, we present ScaleGUN, the first approach to scale certified graph unlearning to billion-edge graphs. ScaleGUN integrates the approximate graph propagation technique into certified graph unlearning, offering certified guarantees for three unlearning scenarios: node feature, edge, and node unlearning. Extensive experiments on real-world datasets demonstrate the efficiency and unlearning efficacy of ScaleGUN. Remarkably, ScaleGUN accomplishes $(\epsilon, \delta) = (1, 10^{-4})$ certified unlearning on the billion-edge graph ogbn-papers100M in 20 seconds for a 5,000 random edge removal request – of which only 5 seconds are required for updating the node embeddings – compared to 1.91 hours for retraining and 1.89 hours for re-propagation. Our code is available at https://github.com/luyi256/ScaleGUN.

## 1 Introduction

Graph Neural Networks (GNNs) have been widely adopted in various applications that involve sensitive user data, such as recommender systems (He et al., 2020), online social networks (Qiu et al., 2018), and financial networks (Weber et al., 2019). With the increasing demands for privacy protection, recent regulations have been established to ensure "the right to be forgotten" of users (Dang, 2021). Advances in privacy protection have led to the emergence of *graph unlearning* (Said et al., 2023), which aims to remove certain user data from trained GNNs. A straightforward approach is to retrain the model from scratch without the data to remove. However, retraining incurs substantial computation costs, rendering it impractical for frequent removal requests. Consequently, various graph unlearning methods attempt to enhance efficiency while maintaining performance and privacy protection comparable to retraining. Among them, *certified graph unlearning* (Chien et al., 2022a) has garnered increasing attention due to its solid theoretical guarantees for privacy protection.

To achieve certified graph unlearning, an unlearning mechanism must be proved to ensure that the unlearned model is *approximately* equivalent to the retrained model with respect to their probability distributions. Specifically, certified unlearning is guaranteed if the gradient residual norm of the loss function, $\|\mathcal{L}(\mathbf{w}^-, \mathcal{D}')\|$, is bounded for the unlearned model $\mathbf{w}^-$ and the updated dataset $\mathcal{D}'$ post-unlearning (Guo et al., 2019). This can be accomplished by introducing noise perturbation into the loss function, thereby rendering the unlearned and retrained models approximately indistinguishable in their probability distributions. Intuitively, the gradient residual norm measures the *model error*

---

*Zhewei Wei is the corresponding author.

of the unlearned model, representing the discrepancy between the unlearned and retrained model. When the model error is limited in a certain bound relative to the noise level, the noise can effectively *mask* the error, thereby ensuring certified guarantees. Thus, previous studies attempt to achieve certified graph unlearning by perturbing the loss and subsequently deriving the gradient residual norm bound by non-trivial theoretical analysis. CGU (Chien et al., 2022a) takes the first step to achieve certified graph unlearning on the SGC (Wu et al., 2019) model and CEU Wu et al. (2023b) further extends this work to batch edge unlearning. To satisfy the requirements of certified unlearning, the aforementioned studies typically re-compute graph propagation, such as $(\mathbf{D}^{-\frac{1}{2}}\mathbf{A}\mathbf{D}^{-\frac{1}{2}})^2\mathbf{X}$ in a 2-hop SGC, and subsequently perform a single Newton update step on the model parameters. This approach demonstrates substantial efficiency gains compared to retraining on small graphs.

**Motivations.** However, we observe that the existing certified graph unlearning methods struggle to scale to large graphs by conducting evaluations on ogbn-papers100M dataset (Hu et al., 2020), which comprises 111M nodes and 1.6B edges. Figure 1 (a) shows the costs of existing unlearning methods when removing 5,000 random edges with a $(\epsilon, \delta) = (1, 10^{-4})$ certified guarantee. Re-computing graph propagation for a 2-hop SGC model requires over 6,000 seconds while retraining the SGC model after graph propagation takes less than 80 seconds. Furthermore, by excluding the propagation cost, unlearning can be finished in under 20 seconds via a single update step of model parameters.

Since existing methods require re-propagation for each unlearning request, addressing one such request demands requires about 6,000 seconds, whose cost is expensive and almost comparable to retraining. The high cost of existing solutions contradicts the objective of unlearning, implying that reducing propagation costs is imperative to scale certified graph unlearning to large graphs.

Intuitively, scalable techniques developed for graph learning (Chiang et al., 2019; Zeng et al., 2019) can be utilized to accelerate the propagation process in certified graph unlearning. However, the existing scalable approaches introduce

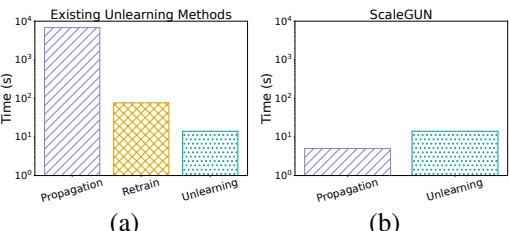

(a)         (b)

Figure 1: Time costs per 5,000-random-edge batch removal for existing unlearning methods v.s. Scale-GUN on ogbn-papers100M (2-hop SGC model).

approximation error into the node embeddings, i.e., leverage an *approximate* dataset $\hat{\mathcal{D}}'$ rather than the *exact* $\mathcal{D}'$. Conversely, deriving a certified guarantee for unlearning requires bounding the model error, the gradient residual norm of the loss function $\|\mathcal{L}(\mathbf{w}^-, \mathcal{D}')\|$ on the *exact* $\mathcal{D}'$. *This discrepancy leaves the impact of approximation error on the model error of unlearning unclear, making it uncertain whether existing scalable approaches can be directly integrated into certified graph unlearning.*

**Contributions.** In this study, we address the research question: *How can we integrate existing scalable techniques into certified graph unlearning to improve scalability while maintaining certified guarantees?* We verify the feasibility of this integration and propose ScaleGUN, the first approach for Scalable and certifiable Graph UNlearning, ensuring both efficiency and privacy guarantees for billion-edge graphs. Our main contributions are as follows.

- Uncovering the impact of approximation error on certified guarantees. We integrate the approximate propagation technique from decoupled models (Zhang et al., 2016; Zheng et al., 2022) into certified graph unlearning and derive theoretical guarantees for three graph unlearning scenarios: node feature, edge, and node unlearning. By non-trivial theoretical analysis, we reveal that the approximation error only marginally increases the model error of unlearning, while still ensuring that the total model error remains bounded. Therefore, the total model error can also be masked by noise perturbation, ensuring the certified unlearning guarantee.

- Lazy local propagation framework for Generalized PageRank. Existing approximate propagation techniques for dynamic Personalized PageRank cannot be applied to commonly used models in certified graph unlearning, such as SGC. To address the gap, we extend these techniques to Generalized PageRank, facilitating efficient propagation for layered propagation schemes.

- Empirical studies. Extensive experiments demonstrate that ScaleGUN achieves superior performance, efficiency, and privacy protection trade-offs on large graphs. Specifically, ScaleGUN takes only 5 seconds to update embeddings and 20 seconds in total to execute the aforementioned

unlearning request on ogbn-papers100M, as shown in Figure 1. Additionally, we study the impact of approximation error on certified unlearning by analyzing the approximate propagation parameter $r_{\max}$. The results reveal that the approximation error can be controlled by $r_{\max}$ to simultaneously achieve impressive model performance and unlearning efficiency.

## 2 PRELIMINARIES

**Notation.** We consider an undirected graph $\mathcal{G} = (\mathcal{V}, \mathcal{E})$ with node set $\mathcal{V}$ of size $n$ and edge set $\mathcal{E}$ of size $m$. We denote the size of the training set as $n_t$. Each node is associated with an $F$-dimensional feature vector $\mathbf{x}$, and the feature matrix is denoted by $\mathbf{X} \in \mathbb{R}^{n \times F}$. $\mathbf{A}$ and $\mathbf{D}$ are the adjacency matrix and the diagonal degree matrix of $\mathcal{G}$ with self-loops, respectively. $\mathcal{N}(u)$ represents the neighbors of node $u$. Let $\mathcal{A}$ be a (randomized) learning algorithm that trains on graph-structured data $\mathcal{D} = (\mathbf{X}, \mathbf{Y}, \mathbf{A}) \in \mathcal{X}$, where $\mathbf{Y}$ are the labels of the training nodes and $\mathcal{X}$ represents the space of possible datasets. $\mathcal{A}$ outputs a model $h \in \mathcal{H}$, where $\mathcal{H}$ is the hypothesis set, that is, $\mathcal{A} : \{\mathcal{D}\} \to \mathcal{H}$. Suppose that $\mathcal{D}'$ is the new graph dataset resulting from a desired removal. An unlearning method $\mathcal{M}$ applied to $\mathcal{A}(\mathcal{D})$ will output a new model $h \in \mathcal{H}$ according to the unlearning request, that is $\mathcal{M}(\mathcal{A}(\mathcal{D}), \mathcal{D}, \mathcal{D}') \to \mathcal{H}$. We add a hat to a variable to denote the approximate version of it, e.g., $\hat{\mathbf{Z}}$ is the approximation of the embedding matrix $\mathbf{Z}$. For simplicity, we denote the data as $\hat{\mathcal{D}}$ to indicate that $\hat{\mathbf{Z}}$ takes the place of $\mathbf{Z}$ during the learning and unlearning process. We add a prime to a variable to denote the variable after the removal, e.g., $\mathcal{D}'$ is the new dataset post-unlearning.

**Certified removal.** Guo et al. (2019) define *certified removal* for unstructured data as follows. Given $\epsilon, \delta > 0$, the original dataset $\mathcal{D}$, a removal request resulting in $\mathcal{D}'$, a removal mechanism $\mathcal{M}$ guarantees $(\epsilon, \delta)$-certified removal for a learning algorithm $\mathcal{A}$ if $\forall \mathcal{T} \subseteq \mathcal{H}, \mathcal{D} \subseteq \mathcal{X}$,

$$\mathbb{P}(\mathcal{M}(\mathcal{A}(\mathcal{D}), \mathcal{D}, \mathcal{D}') \in \mathcal{T}) \leq e^\epsilon \mathbb{P}(\mathcal{A}(\mathcal{D}') \in \mathcal{T}) + \delta, \text{ and}$$
$$\mathbb{P}(\mathcal{A}(\mathcal{D}') \in \mathcal{T}) \leq e^\epsilon \mathbb{P}(\mathcal{M}(\mathcal{A}(\mathcal{D}), \mathcal{D}, \mathcal{D}') \in \mathcal{T}) + \delta.$$

This definition states the likelihood ratio between the retrained model on $\mathcal{D}'$ and the unlearned model via $\mathcal{M}$ is *approximately* equivalent in terms of their probability distributions. Guo et al. (2019) also introduced a certified unlearning mechanism for linear models trained on unstructured data. Consider $\mathcal{A}$ trained on $\mathcal{D} = \{(\mathbf{x}_1, y_1), \cdots, (\mathbf{x}_n, y_n)\}$ aiming to minimize the empirical risk $\mathcal{L}(\mathbf{w}; \mathcal{D}) = \sum_{i=1}^{n} l\left(\mathbf{w}^\top \mathbf{x}_i, y_i\right) + \frac{\lambda n}{2} \|\mathbf{w}\|_2^2$, where $l$ is a convex loss function that is differentiable everywhere. Let $\mathbf{w}^\star = \mathcal{A}(\mathcal{D})$ be the unique optimum such that $\nabla \mathcal{L}(\mathbf{w}^\star, \mathcal{D}) = 0$. Guo et al. propose the *Newton update removal mechanism*: $\mathbf{w}^- = \mathbf{w}^\star + \mathbf{H}_{\mathbf{w}^\star}^{-1} \Delta$, where $\Delta = \lambda \mathbf{w}^\star + \nabla l((\mathbf{w}^\star)^\top \mathbf{x}, y)$ when $(\mathbf{x}, y)$ is removed, and $\mathbf{H}_{\mathbf{w}^\star}$ is the Hessian matrix of $\mathcal{L}(\cdot, \mathcal{D}')$ at $\mathbf{w}^\star$. Furthermore, to mask the direction of the gradient residual $\nabla \mathcal{L}(\mathbf{w}^-, \mathcal{D}')$, they introduce a perturbation in the empirical risk: $\mathcal{L}_{\mathbf{b}}(\mathbf{w}, \mathcal{D}) = \mathcal{L}(\mathbf{w}, \mathcal{D}) + \mathbf{b}^\top \mathbf{w}$, where $\mathbf{b} \in \mathbb{R}^F$ is a random vector sampled from a specific distribution. Then, one can secure a certified guarantee by using the following theorem:

**Theorem 2.1** (Theorem 3 in (Guo et al., 2019)). *Let $\mathcal{A}$ be the learning algorithm that returns the unique optimum of the loss $\mathcal{L}_{\mathbf{b}}(\mathbf{w}; \mathcal{D})$. Suppose that a removal mechanism $\mathcal{M}$ returns $\mathbf{w}^-$ with $\|\nabla \mathcal{L}(\mathbf{w}^-; \mathcal{D}')\|_2 \leq \epsilon'$ for some computable bound $\epsilon' > 0$. If $\mathbf{b} \sim \mathcal{N}(0, c\epsilon'/\epsilon)^d$ with $c > 0$, then $\mathcal{M}$ guarantees $(\epsilon, \delta)$-certified removal for $\mathcal{A}$ with $\delta = 1.5 \cdot e^{-c^2/2}$.*

**Certified graph unlearning.** CGU (Chien et al., 2022a) generalizes the certified removal mechanism to graph-structured data on SGC and the Generalized PageRank extensions. The empirical risk is modified to $\mathcal{L}(\mathbf{w}; \mathcal{D}) = \sum_{i=1}^{n_t} l\left(\mathbf{e}_i^\top \mathbf{Z} \mathbf{w}, \mathbf{e}_i^\top \mathbf{Y}\right) + \frac{\lambda n_t}{2} \|\mathbf{w}\|_2^2$, where $\mathbf{e}_i^\top \mathbf{Z} = \mathbf{e}_i^\top (\mathbf{D}^{-1} \mathbf{A})^k \mathbf{X}$, $\mathbf{e}_i^\top \mathbf{Y}$ is the embedding vector and the label of node $i$, respectively, and $n_t$ denotes the number of training nodes. Removing a node alters its neighbors' embeddings, consequently affecting their loss values. Therefore, CGU generalizes the mechanism in (Guo et al., 2019) by revising $\Delta$ as $\nabla \mathcal{L}(\mathbf{w}^\star, \mathcal{D}) - \nabla \mathcal{L}(\mathbf{w}^\star, \mathcal{D}')$. If no graph structure is present, implying $\mathbf{Z}$ is independent of the graph structure, CGU aligns with that of (Guo et al., 2019). Based on the modifications, CGU derives the certified unlearning guarantee for three types of removal requests: node feature, edge, and node unlearning.

**Approximate dynamic PPR propagation techniques via Forward Push.** Forward Push (Andersen et al., 2006; Yang et al., 2024) is a canonical technique designed to accelerate graph propagation computations for decoupled models (Wang et al., 2021; Chen et al., 2020). The technique is a localized version of cumulative power iteration, performing one *push* operation for a single node

at a time. Take Forward Push for $\mathbf{z} = \mathbf{P}^\ell \mathbf{x}$ and $\mathbf{P} = \mathbf{A}\mathbf{D}^{-1}$ as an example, where $\mathbf{x}$ is the graph signal vector and $\ell$ is the number of propagation steps. Forward Push maintains a reserve vector $\boldsymbol{q}^{(\ell)}$ and a residue vector $\boldsymbol{r}^{(\ell)}$ for each level $\ell$. For any node $u \in \mathcal{V}$, $\boldsymbol{q}^{(\ell)}(u)$ represents the current estimation of $(\mathbf{P}^\ell \mathbf{x})(u)$, and $\boldsymbol{r}^{(\ell)}(u)$ holds the residual mass to be distributed to subsequent levels. When $\boldsymbol{r}^{(\ell)}(u)$ exceeds a predefined threshold $r_{\max}$, it is added to $\boldsymbol{q}^{(\ell)}(u)$ and distributed to the residues of $u$'s neighbors in the next level, i.e., $\boldsymbol{r}^{(\ell+1)}(v)$ is increased by $\boldsymbol{r}^{(\ell)}(u)/d(u)$ for $v \in \mathcal{N}(u)$. After that, $\boldsymbol{r}^{(\ell)}(u)$ is set to 0. Ignoring small residues that contribute little to the estimate, Forward Push achieve a balance between estimation error and efficiency. Several dynamic PPR algorithms have been proposed to solve PPR-based propagation in evolving graphs (Zhang et al., 2016; Guo et al., 2021). We focus on InstantGNN (Zheng et al., 2022) as the lazy local propagation framework in our ScaleGUN draws inspiration from it. Specifically, InstantGNN observed that the invariant property $\hat{\boldsymbol{\pi}} + \alpha \boldsymbol{r} = \alpha \mathbf{x} + (1 - \alpha)\mathbf{P}\hat{\boldsymbol{\pi}}$ holds during the propagation process for the propagation scheme $\boldsymbol{\pi} = \sum_{\ell=0}^{\infty} \alpha(1 - \alpha)^\ell \mathbf{P}^\ell \mathbf{x}$. Upon an edge arrival or removal, this invariant is disrupted due to the revised $\mathbf{P}$ or $\mathbf{x}$. InstantGNN updates the residue vector $\boldsymbol{r}$ for affected nodes to maintain the invariant, then applies Forward Push to meet the error requirement. Since only a few nodes are affected by one edge change, the updates are local and efficient.

## 3 LAZY LOCAL PROPAGATION

This section introduces our lazy local propagation framework that generates approximate embeddings with a bounded $L_2$-error. To effectively capture the propagation scheme prevalent in current GNN models and align with the existing graph unlearning methods, we adopt the Generalized PageRank (GPR) approach (Li et al., 2019) as the propagation scheme, $\mathbf{Z} = \sum_{\ell=0}^{L} w_\ell \left( \mathbf{D}^{-\frac{1}{2}} \mathbf{A} \mathbf{D}^{-\frac{1}{2}} \right)^\ell \mathbf{X}$, where $w_\ell$ is the weight of the $\ell$-th order propagation matrix and $\sum_{\ell=0}^{L} |w_\ell| \le 1$. Note that this propagation scheme differs from that of CGU (Chien et al., 2022a), which uses an asymmetric normalized matrix $\mathbf{P} = \mathbf{D}^{-1}\mathbf{A}$ and fixes $w_\ell = 1/L$. We introduce the initial propagation method for a signal vector $\mathbf{x}$ and the corresponding update method. The approximate embedding matrix $\hat{\mathbf{Z}}$ can be obtained by parallel adopting the method for each signal vector and putting the results together. Details and pseudo-codes of our propagation framework are provided in Appendix D.

*Remark.* The lazy local propagation framework draws inspiration from the dynamic PPR method of InstantGNN (Zheng et al., 2022) and extends it to the GPR propagation scheme. To the best of our knowledge, our framework is the first to achieve efficient embeddings update for the layered propagation scheme in evolving graphs, making it readily applicable to various models, such as SGC, GBP (Chen et al., 2020), and GDC (Gasteiger et al., 2019). Utilizing the GPR scheme also enhances the generalizability of ScaleGUN, allowing one to replace our propagation framework with dynamic PPR methods to achieve certified unlearning for PPR-based GNNs. Note that InstantGNN cannot be directly applied to certified unlearning because the impact of approximation on the certified unlearning guarantees has not been well studied. This paper's main contribution is addressing this gap. We establish certified guarantee for GNNs that use approximate embeddings, as detailed in Section 4. The differences between our propagation framework and existing propagation methods are further detailed in Appendix D.1.

For the initial propagation, we adopt Forward Push with $\mathbf{P} = \mathbf{A}\mathbf{D}^{-1}$, drawing from the observation that $(\mathbf{D}^{-\frac{1}{2}} \mathbf{A} \mathbf{D}^{-\frac{1}{2}})^\ell = \mathbf{D}^{-\frac{1}{2}} (\mathbf{A}\mathbf{D}^{-1})^\ell \mathbf{D}^{\frac{1}{2}}$. Initially, we normalize $\mathbf{x}$ to ensure $\left\| \mathbf{D}^{\frac{1}{2}} \mathbf{x} \right\|_1 \le 1$, and set the residue vector $\boldsymbol{r}^{(0)} = \mathbf{D}^{\frac{1}{2}} \mathbf{x}$. Applying Forward Push, we derive $\hat{\mathbf{z}} = \sum_{\ell=0}^{L} w_\ell \mathbf{D}^{-\frac{1}{2}} \boldsymbol{q}^{(\ell)}$ as the approximation of $\mathbf{z} = \sum_{\ell=0}^{L} w_\ell \left( \mathbf{D}^{-\frac{1}{2}} \mathbf{A} \mathbf{D}^{-\frac{1}{2}} \right)^\ell \mathbf{x}$. Any non-zero $\boldsymbol{r}^{(\ell)}$ holds the weight mass not yet passed on to subsequent levels, leading to the approximation error between $\hat{\mathbf{z}}$ and $\mathbf{z}$. During the propagation process, it holds that

$$\mathbf{z} = \sum_{\ell=0}^{L} w_\ell \mathbf{D}^{-\frac{1}{2}} \left( \boldsymbol{q}^{(\ell)} + \sum_{t=0}^{\ell} (\mathbf{A}\mathbf{D}^{-1})^{\ell-t} \boldsymbol{r}^{(t)} \right). \tag{1}$$

By examining $\sum_{\ell=0}^{L} w_\ell \mathbf{D}^{-\frac{1}{2}} \sum_{t=0}^{\ell} (\mathbf{A}\mathbf{D}^{-1})^{\ell-t} \boldsymbol{r}^{(t)}$, we establish the error bound of $\hat{\mathbf{z}}$ as follows.

**Lemma 3.1.** *Given graph $\mathcal{G}$ with $n$ nodes and the threshold $r_{\max}$, the approximate embedding $\hat{\mathbf{z}}$ for signal vector $\mathbf{x}$ satisfies that $\|\hat{\mathbf{z}} - \mathbf{z}\|_2 \le \sqrt{n} L r_{\max}$.*

Inspired by the update method for the PPR-based propagation methods on dynamic graphs (Zheng et al., 2022), we identify the invariant property for the GPR scheme as follows.

**Lemma 3.2.** *For each signal vector* $\mathbf{x}$*, the reserve vectors* $\{\boldsymbol{q}^{(\ell)}\}$ *and the residue vectors* $\{\boldsymbol{r}^{(\ell)}\}$ *satisfy the following invariant property for all* $u \in \mathcal{V}$ *during the propagation process:*

$$
\begin{cases}
\boldsymbol{q}^{(\ell)}(u) + \boldsymbol{r}^{(\ell)}(u) = \mathbf{D}(u)^{\frac{1}{2}}\mathbf{x}(u), & \ell = 0 \\
\boldsymbol{q}^{(\ell)}(u) + \boldsymbol{r}^{(\ell)}(u) = \sum_{t \in \mathcal{N}(u)} \dfrac{\boldsymbol{q}^{(\ell-1)}(t)}{d(t)}, & 0 < \ell \leq L
\end{cases}
\tag{2}
$$

Upon a removal request, we adjust $\{\boldsymbol{r}^{(\ell)}\}$ locally for the affected nodes to maintain the invariant property. Consider an edge removal scenario, for example, where edge $(u, v)$ is targeted for removal. Only node $u$, node $v$, and their neighbors fail to meet Equation (2) due to the altered degrees of $u$ and $v$. Take the modification for node $u$ as an example. For level 0, we update $\boldsymbol{r}^{(0)}(u)$ reflecting changes in $\mathbf{D}^{\frac{1}{2}}\mathbf{x}$. For subsequent levels, $\boldsymbol{r}^{(\ell)}(u)$ is updated to reflect that the right side of $u$'s equations exclude $\frac{\boldsymbol{q}^{(\ell-1)}(v)}{d(v)}$. For $u$'s neighbors, their residues are updated accordingly, since one term on the right, $\frac{\boldsymbol{q}^{(\ell-1)}(u)}{d(u)}$, shifts to $\frac{\boldsymbol{q}^{(\ell-1)}(u)}{d(u)-1}$. Post-adjustment, the invariant property is preserved across all nodes. Then we invoke Forward Push to secure the error bound and acquire the updated $\hat{\mathbf{z}}$. Removing a node can be treated as multiple edge removals by eliminating all edges connected to the node. Moreover, feature removal for node $u$ can be efficiently executed by setting $\boldsymbol{r}^{(0)}(u)$ as $-\boldsymbol{q}^{(0)}(u)$. The following theorem illustrates the average cost for each removal request.

**Theorem 3.3** (Average Cost)**.** *For a sequence of* $m$ *removal requests that remove all edges of the graph, the amortized cost per edge removal is* $O\left(L^2 d\right)$*. For a sequence of* $K$ *random edge removals, the expected cost per edge removal is* $O\left(L^2 d\right)$*.*

Note that the propagation step $L$ and the average degree $d$ are both typically a small constant in practice. $L$ is commonly set to 2 in GCN (Kipf & Welling, 2016), SGC, GAT (Velickovic et al., 2017). Many real-world networks are reported to be scale-free, characterized by a small average degree (Barabási, 2013). For instance, the citation network ogbn-arxiv and ogbn-papers100M exhibit average degrees of 6 and 14, respectively. Consequently, this setup generally allows for constant time complexity for each removal request.

## 4 SCALABLE AND CERTIFIABLE UNLEARNING MECHANISM

This section presents the certified graph unlearning mechanism of ScaleGUN based on the approximate embeddings derived from our lazy local propagation framework. Following existing works, we first study linear models with a strongly convex loss function and focus on binary node classification problems. We define the empirical risk as $\mathcal{L}(\mathbf{w}, \mathcal{D}) = \sum_{i \in [n_t]} \left( l(\mathbf{e}_i^\top \hat{\mathbf{Z}}\mathbf{w}, \mathbf{e}_i^\top \mathbf{Y}) + \frac{\lambda}{2} \|\mathbf{w}\|^2 \right)$, where $l$ is a convex loss that is differentiable everywhere, $\mathbf{e}_i^\top \hat{\mathbf{Z}}$ and $\mathbf{e}_i^\top \mathbf{Y}$ represents the approximate embedding vector and the label of node $i$, respectively. Without loss of generality, we assume that the training set contains the first $n_t$ nodes. Suppose that $\mathbf{w}^*$ is the unique optimum of the original graph and an unlearning request results in a new graph $\mathcal{D}'$. Our unlearning approach produces a new model $\mathbf{w}^-$ as follows: $\mathbf{w}^- = \mathbf{w}^\star + \hat{\mathbf{H}}_{\mathbf{w}^\star}^{-1}\Delta$, where $\Delta = \nabla \mathcal{L}(\mathbf{w}^\star, \hat{\mathcal{D}}) - \nabla \mathcal{L}(\mathbf{w}^\star, \hat{\mathcal{D}}')$ and $\hat{\mathbf{H}}_{\mathbf{w}^\star}$ is the Hessian matrix of $\mathcal{L}(\cdot, \hat{\mathcal{D}}')$ at $\mathbf{w}^\star$. We also introduce a perturbation in the loss function following (Guo et al., 2019) to hide information: $\mathcal{L}_{\mathbf{b}}(\mathbf{w}, \mathcal{D}) = \mathcal{L}(\mathbf{w}, \mathcal{D}) + \mathbf{b}^\top \mathbf{w}$, where $\mathbf{b} \in \mathbb{R}^F$ is a noise vector sampled from a specific distribution.

Compared to the unlearning mechanism, $\mathbf{w}^- = \mathbf{w}^\star + \mathbf{H}_{\mathbf{w}^\star}^{-1}\left(\nabla \mathcal{L}(\mathbf{w}^\star, \mathcal{D}) - \nabla \mathcal{L}(\mathbf{w}^\star, \mathcal{D}')\right)$, proposed by Chien et al. (2022a), our primary distinction lies in the approximation of embeddings. This also poses the main theoretical challenges in proving certified guarantee. According to Theorem 2.1, an unlearning mechanism can ensure $(\epsilon, \delta)$-certified graph unlearning if $\|\nabla \mathcal{L}(\mathbf{w}^-, \mathcal{D}')\|$ is bounded. Bounding $\|\nabla \mathcal{L}(\mathbf{w}^-, \mathcal{D}')\|$ is the main challenge to develop a certified unlearning mechanism. One of the leading contributions of (Guo et al., 2019; Chien et al., 2022b) is to establish the bounds for their proposed mechanisms. In the following, we elaborate on the bounds of $\|\nabla \mathcal{L}(\mathbf{w}^-, \mathcal{D}')\|$ of ScaleGUN under three graph unlearning scenarios: node feature, edge, and node unlearning. Before that, we make the following assumptions.

**Assumption 4.1.** For any dataset $\mathcal{D} \in \mathcal{X}$, $i \in [n]$ and $\mathbf{w} \in \mathbb{R}^F$: (1) $\left\|\nabla l(\mathbf{e}_i^\top \mathbf{Z}\mathbf{w}, \mathbf{e}_i^\top \mathbf{Y})\right\| \le c$; (2) $l'$ is $c_1$-bounded; (3) $l'$ is $\gamma_1$-Lipschitz; (4) $l''$ is $\gamma_2$-Lipschitz; (5) $\left\|\mathbf{e}_i^\top \mathbf{X}\right\| \le 1$.

Note that these assumptions are also needed for (Chien et al., 2022a). Assumptions (1)(3)(5) can be avoided when working with the data-dependent bound in Theorem 4.5. We first focus on a single instance unlearning and extend to multiple unlearning requests in Section 4.3.

## 4.1 NODE FEATURE UNLEARNING

We follow the definitions of all three graph unlearning scenarios in (Chien et al., 2022a). In the node feature unlearning case, the feature and label of a node are removed, resulting in $\mathcal{D}' = (\mathbf{X}', \mathbf{Y}', \mathbf{A})$, where $\mathbf{X}'$ and $\mathbf{Y}'$ is identical to $\mathbf{X}$ and $\mathbf{Y}$ except the row of the removed node is zero. Without loss of generality, we assume that the removed node is node $u$ in the training set. Our conclusion remains valid even when the removed node is not included in the training set.

**Theorem 4.2** (Worst-case bound of node feature unlearning). *Suppose that Assumption 4.1 holds and the feature of node $u$ is to be unlearned. If $\forall j \in [F]$, $\left\|\hat{\mathbf{Z}}\mathbf{e}_j - \mathbf{Z}\mathbf{e}_j\right\| \le \epsilon_1$, we have*

$$\left\|\nabla\mathcal{L}(\mathbf{w}^-, \mathcal{D}')\right\| \le \left(\frac{c\gamma_1}{\lambda}F + c_1\sqrt{F(n_t - 1)}\right)\left(\epsilon_1 + \frac{8\gamma_1 F}{\lambda(n_t - 1)} \cdot \sqrt{d(u)}\right).$$

The feature dimension $F$ affects the outcome as the analysis is conducted on one dimension of the embedding $\mathbf{Z}_j$ rather than on $\mathbf{e}_i^\top \mathbf{Z}$. In real-world datasets, $F$ is typically a small constant. Note that $\epsilon_1$ is exactly $\sqrt{n}Lr_{\max}$ according to Lemma 3.1. To ensure that the norm will not escalate with the training set size $n_t$, we typically set $\epsilon_1 = O(\frac{1}{\sqrt{n_t}})$, which implies that $r_{\max} = O(1/\sqrt{n_t n})$. The bound can be viewed as comprising two components: the component resulting from approximation, $\left(\frac{c\gamma_1}{\lambda}F + c_1\sqrt{F(n_t - 1)}\right)\epsilon_1$, and the component resulting from unlearning, $\left(\frac{c\gamma_1}{\lambda}F + c_1\sqrt{F(n_t - 1)}\right)\frac{8\gamma_1 F}{\lambda(n_t - 1)} \cdot \sqrt{d(u)}$. In the second component, the norm increases if the unlearned node possesses a high degree, as removing a large-degree node's feature impacts the embeddings of many other nodes. This component is not affected by $L$, the propagation step, due to the facts that $(\mathbf{D}^{-\frac{1}{2}}\mathbf{A}\mathbf{D}^{-\frac{1}{2}})^\ell = \mathbf{D}^{-\frac{1}{2}}(\mathbf{A}\mathbf{D}^{-1})^\ell\mathbf{D}^{\frac{1}{2}}$ and that $\mathbf{A}\mathbf{D}^{-1}$ is left stochastic.

**Analytical challenges.** It is observed that $\|\nabla\mathcal{L}(\mathbf{w}^-, \mathcal{D}')\|$ originates from the exact embeddings of $D'$, whereas $\mathbf{w}^-$ is derived from approximate embeddings in ScaleGUN. Thus, the first challenge is to establish the connection between $\mathcal{D}'$ and $\hat{\mathcal{D}}'$ in $\|\nabla\mathcal{L}(\mathbf{w}^-, \mathcal{D}')\|$. To address this issue, we employ the Minkowski inequality to adjust the norm: $\|\nabla\mathcal{L}(\mathbf{w}^-, \mathcal{D}')\| \le \left\|\nabla\mathcal{L}(\mathbf{w}^-, \mathcal{D}') - \nabla\mathcal{L}(\mathbf{w}^-, \hat{\mathcal{D}}')\right\| + \left\|\nabla\mathcal{L}(\mathbf{w}^-, \hat{\mathcal{D}}')\right\|$. The first norm depicts the difference between the gradient of $\mathbf{w}^-$ on the exact and approximate embeddings, which is manageable through the approximation error. The second norm, $\left\|\nabla\mathcal{L}(\mathbf{w}^-, \hat{\mathcal{D}}')\right\|$, signifies the error of $\mathbf{w}^-$ as the minimizer of $\mathcal{L}(\cdot, \hat{\mathcal{D}}')$, given as

$$\left\|\nabla\mathcal{L}(\mathbf{w}^-, \hat{\mathcal{D}}')\right\| = \left\|(\hat{\mathbf{H}}_{\mathbf{w}_\eta} - \hat{\mathbf{H}}_{\mathbf{w}^\star})\hat{\mathbf{H}}_{\mathbf{w}^\star}^{-1}\Delta\right\|,$$

where $\hat{\mathbf{H}}_{\mathbf{w}_\eta}$ is the Hessian of $\mathcal{L}(\cdot, \hat{\mathcal{D}}')$ at $\mathbf{w}_\eta = \mathbf{w}^\star + \eta\hat{\mathbf{H}}_{\mathbf{w}^\star}^{-1}\Delta$ for some $\eta \in [0, 1]$. This introduces the second challenge: bounding $\|\Delta\|$. Bounding $\|\Delta\|$ is intricate, especially for graph unlearning scenarios. Here, $\Delta$ represents the difference between the gradient of $\mathbf{w}^*$ on pre- and post-removal datasets, i.e., $\hat{\mathbf{Z}}$ and $\hat{\mathbf{Z}}'$. Therefore, establishing the mathematical relationship between $\mathbf{Z}$ and $\mathbf{Z}'$ is the crucial point. Although prior studies have explored the bound of $\left\|\mathbf{e}_i^\top(\mathbf{Z} - \mathbf{Z}')\right\|$, their interest primarily lies in $\mathbf{P} = \mathbf{D}^{-1}\mathbf{A}$ (Chien et al., 2022a) or 1-hop feature propagation (Wu et al., 2023b), which cannot be generalized to our GPR propagation scheme with $\mathbf{P} = \mathbf{D}^{-\frac{1}{2}}\mathbf{A}\mathbf{D}^{-\frac{1}{2}}$. We address this challenge innovatively by taking advantage of our lazy local propagation framework as follows.

**bounding $\|\Delta\|$ via the propagation framework.** Assuming node $u$ is the $n_t$-th node in the training set, for the case of feature unlearning, we have

$$\Delta = \lambda\mathbf{w}^\star + \nabla l(\hat{\mathbf{Z}}, \mathbf{w}^\star, n_t) + \sum_{i \in [n_t - 1]}\left(\nabla l(\hat{\mathbf{Z}}, \mathbf{w}^\star, i) - \nabla l(\hat{\mathbf{Z}}', \mathbf{w}^\star, i)\right),$$

where $l(\hat{\mathbf{Z}}, \mathbf{w}^\star, i)$ is short for $l(\mathbf{e}_i^\top \hat{\mathbf{Z}} \mathbf{w}^\star, \mathbf{e}_i^\top \mathbf{Y})$. Creatively, we bound $\|\mathbf{Z} \mathbf{e}_j - \mathbf{Z}' \mathbf{e}_j\|$ for all $j \in [F]$ via the lazy local propagation framework. Let $\mathbf{z} = \mathbf{Z} \mathbf{e}_j$ and $\mathbf{z}' = \mathbf{Z}' \mathbf{e}_j$ for brevity. In our propagation framework, right after adjusting the residues upon a removal, $\{q^{(\ell)}\}$ remains unchanged. Moreover, Equation (1) holds during the propagation process. Thus, we have

$$\mathbf{z} - \mathbf{z}' = \sum_{t=0}^{\ell} w_\ell \sum_{t=0}^{\ell} \mathbf{D}^{-\frac{1}{2}} (\mathbf{A} \mathbf{D}^{-1})^{\ell - t} \left( \boldsymbol{r}^{(t)} - \boldsymbol{r}'^{(t)} \right).$$

To derive $\boldsymbol{r}^{(t)} - \boldsymbol{r}'^{(t)}$, notice that Equation (1) is applicable across all $r_{\max}$ configurations. Setting $r_{\max} = 0$ to eliminate error results in $\boldsymbol{r}^{(t)}$ becoming a zero vector. Thus, for the feature unlearning scenario, $\boldsymbol{r}^{(0)} - \boldsymbol{r}'^{(0)}$ is precisely $-\boldsymbol{q}^{(0)}(u)\mathbf{e}_u$, as only node $u$'s residue is modified. For $t > 0$, $\boldsymbol{r}^{(t)} - \boldsymbol{r}'^{(t)}$ are all zero vectors since these residues are not updated. This solves the second challenge. The bounds for edge and node unlearning scenarios can be similarly derived via the lazy local framework, significantly streamlining the proof process compared to direct analysis of $\mathbf{Z} \mathbf{e}_j - \mathbf{Z}' \mathbf{e}_j$.

## 4.2 EDGE UNLEARNING AND NODE UNLEARNING

In the edge unlearning case, we remove an edge $(u, v)$, resulting in $\mathcal{D}' = (\mathbf{X}, \mathbf{Y}, \mathbf{A}')$, where $\mathbf{A}'$ is identical to $\mathbf{A}$ except that the entries for $(u, v)$ and $(v, u)$ are set to zero. The node feature and labels remain unchanged. The conclusion remains valid regardless of whether $u, v$ are in the training set.

**Theorem 4.3** (Worst-case bound of edge unlearning). *Suppose that Assumption 4.1 holds, and the edge $(u, v)$ is to be unlearned. If $\forall j \in [F]$, $\left\| \hat{\mathbf{Z}} \mathbf{e}_j - \mathbf{Z} \mathbf{e}_j \right\| \leq \epsilon_1$, we can bound $\|\nabla \mathcal{L}(\mathbf{w}^-, \mathcal{D}')\|$ by*

$$\frac{4c\gamma_1 F}{\lambda n_t} + \left( \frac{c\gamma_1}{\lambda} F + c_1 \sqrt{F n_t} \right) \left( \epsilon_1 + \frac{2\gamma_1 F}{\lambda n_t} (2\epsilon_1 + \frac{4}{\sqrt{d(u)}} + \frac{4}{\sqrt{d(v)}}) \right).$$

We observe that the worst-case bound diminishes when the two terminal nodes of the removed edge have a large degree. This reduction occurs because the impact of removing a single edge from a node with many edges is relatively minor.

In the node unlearning case, removing a node $u$ results in $\mathcal{D}' = (\mathbf{X}', \mathbf{Y}', \mathbf{A}')$, where the entries regarding the removed node in all three matrices are set to zero. The bound can be directly inferred from Theorem 4.3 since unlearning node $u$ equates to eliminating all edges connected to node $u$. This upper bound indicates that the norm is related to the degree of node $u$ and that of its neighbors.

**Theorem 4.4** (Worst-case bound of node unlearning). *Suppose that Assumption 4.1 holds and node $u$ is removed. If $\forall j \in [F]$, $\left\| \hat{\mathbf{Z}} \mathbf{e}_j - \mathbf{Z} \mathbf{e}_j \right\| \leq \epsilon_1$, we can bound $\|\nabla \mathcal{L}(\mathbf{w}^-, \mathcal{D}')\|$ by*

$$\frac{4c\gamma_1 F}{\lambda(n_t - 1)} + \left( \frac{c\gamma_1}{\lambda} F + c_1 \sqrt{F(n_t - 1)} \right) \left( \epsilon_1 + \frac{2\gamma_1 F}{\lambda(n_t - 1)} (2\epsilon_1 + 4\sqrt{d(u)} + \sum_{w \in \mathcal{N}(u)} \frac{4}{\sqrt{d(w)}}) \right).$$

## 4.3 UNLEARNING ALGORITHM

In this subsection, we introduce the unlearning procedure of ScaleGUN. More practical considerations are deferred to Appendix B.1, including feasible loss functions and batch unlearning.

**Data-dependent bound.** The worst-case bounds may be loose in practice. Following the existing certified unlearning studies (Guo et al., 2019; Chien et al., 2022a), we examined the data-dependent norm as follows. Similar to the worst-case bounds, the data-dependent bound can also be understood as two components: the first term $2c_1 \left\| \mathbf{1}^\top \boldsymbol{R} \right\|$ incurred by the approximation error and the second term $\gamma_2 \left\| \hat{\mathbf{Z}}' \right\| \left\| \hat{\mathbf{H}}_{\mathbf{w}^\star}^{-1} \Delta \right\| \left\| \hat{\mathbf{Z}}' \hat{\mathbf{H}}_{\mathbf{w}^\star}^{-1} \Delta \right\|$ incurred by unlearning. The second term is similar to that of the existing works, except that it is derived from the approximate embeddings.

**Theorem 4.5** (Data-dependent bound). *Let $\boldsymbol{R}^{(t)} \in \mathbb{R}^{n \times F}$ denote the residue matrix at level $t$, where the $j$-th column $\boldsymbol{R}^{(t)} \mathbf{e}_j$ represents the residue at level $t$ for the $j$-th signal vector. Let $\boldsymbol{R}$ be defined as the sum $\sum_{t=0}^{L} \boldsymbol{R}^{(t)}$. We can establish the following data-dependent bound:*

$$\left\| \nabla \mathcal{L}(\mathbf{w}^-, \mathcal{D}') \right\| \leq 2c_1 \left\| \mathbf{1}^\top \boldsymbol{R} \right\| + \gamma_2 \left\| \hat{\mathbf{Z}}' \right\| \left\| \hat{\mathbf{H}}_{\mathbf{w}^\star}^{-1} \Delta \right\| \left\| \hat{\mathbf{Z}}' \hat{\mathbf{H}}_{\mathbf{w}^\star}^{-1} \Delta \right\|.$$

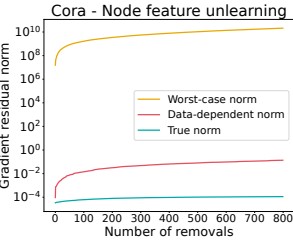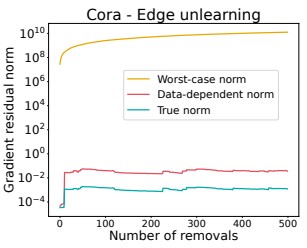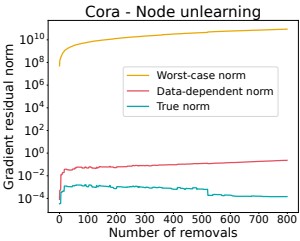

Figure 2: Comparison of the bounds of the gradient residual: Worst-case bound (Theorem 4.2, 4.3, 4.4 for node feature, edge, node unlearning, respectively), data-dependent bound (Theorem 4.5) and the true value on Cora dataset.

**Sequential unlearning algorithm.** Multiple unlearning requests can be executed sequentially. The unlearning process is similar to the existing works (Chien et al., 2022a; Wu et al., 2023b), except that we employ the lazy local propagation framework for the initial training and each removal. Specifically, we select the noise standard deviation $\alpha$ and privacy parameter $\epsilon, \delta$, and compute the "privacy budget". Once the accumulated data-dependent norm exceeds the budget, we retrain the model and reset the accumulated norm. Notably, only the component attributable to unlearning, i.e., the second term in Theorem 4.5, needs to be accumulated. This is because the first term represents the error caused by the current approximation error and does not depend on the previous results. We provide the pseudo-code and illustrate more details in Appendix B.2.

## 5 EXPERIMENTS

In this section, we evaluate the performance of ScaleGUN on real-world datasets, including three small graph datasets: Cora (Sen et al., 2008), Citeseer (Yang et al., 2016), and Photo (McAuley et al., 2015); as well as three large graph datasets: ogbn-arxiv, ogbn-products, and ogbn-papers100M (Hu et al., 2020). Consistent with prior certified unlearning research, we employ LBFGS as the optimizer for linear models. The public splittings are used for all datasets. Unless otherwise stated, we set $L = 2, \epsilon = 1$ for all experiments, averaging results across 5 trials with random seeds. By default, we set $\delta$ as $\frac{1}{\#\text{edges}}$ for edge unlearning and $\frac{1}{\#\text{nodes}}$ for node/feature unlearning in the linear model utility experiments to meet the typical privacy requirements (Chien et al., 2024; Sajadmanesh et al., 2023). Following (Chien et al., 2022a), $w_\ell$ is set to 1 for $\ell = 2$ and 0 for other values. The propagation matrix is set to $\mathbf{P} = \mathbf{D}^{-\frac{1}{2}} \mathbf{A} \mathbf{D}^{-\frac{1}{2}}$ across all the methods for fair comparison. Our benchmarks include CGU (Chien et al., 2022a), CEU (Chen et al., 2022), and the standard retrained method. The experimental results under this setting, additional experiments and detailed configurations are available in Appendix C.

**Evaluations.** We evaluate the performance of ScaleGUN in terms of efficiency, model utility (performance), and unlearning efficacy. The efficiency is measured by the average total cost per removal and the average propagation cost per removal. The model utility is evaluated by the accuracy of node classification. Given the lack of standardized methods for evaluating unlearning efficacy in graphs, we utilize two attack methods to assess this efficacy. For edge unlearning, we apply the task of forgetting adversarial data as described by (Wu et al., 2023a). For node unlearning, we employ the Deleted Data Replay Test (DDRT) as outlined in (Cong & Mahdavi).

**Bounds on the gradient residual norm.** Figure 2 validates the bounds on the gradient residual norm: the worst-case bounds, the data-dependent bounds, and the true value for all three unlearning scenarios on the Cora dataset. For simplicity, the standard deviation $\alpha$ is set to 0 for the noise $\mathbf{b}$. The results demonstrate that both the worst-case bounds and the data-dependent bounds validly upper bound the true value, and the worst-case bounds are looser than the data-dependent bounds.

**Efficiency and model utility on linear models.** We examine the unlearning performance of ScaleGUN on large graph datasets here and provide the results on small graphs in Appendix C. For large datasets, randomly selecting edges to remove is insufficient to affect model accuracy significantly. We propose a novel approach to select a set of *vulnerable edges* for removal. Inspired by Theorem 4.3, we find that edges linked to nodes with small degrees are more likely to yield larger gradient residual norms, thus severely impacting performance. Consequently, we

Table 1: Test accuracy (%), total unlearning cost (s) and propagation cost (s) per batch edge removal for **linear** models (large graphs).

| | ogbn-arxiv | | | | | ogbn-products | | | | | ogbn-papers100M | |
|---|---|---|---|---|---|---|---|---|---|---|---|---|
| $N$ | Retrain | CGU | CEU | ScaleGUN | $N(\times 10^3)$ | Retrain | CGU | CEU | ScaleGUN | $N(\times 10^3)$ | Retrain | ScaleGUN |
| 0 | 57.83 | 57.84 | 57.84 | 57.84 | 0 | 56.24 | 56.17 | 56.17 | 56.04 | 0 | 59.99 | 59.82 |
| 25 | 57.83 | 57.84 | 57.84 | 57.84 | 1 | 56.23 | 56.16 | 56.16 | 56.03 | 2 | 59.71 | 59.71 |
| 50 | 57.82 | 57.83 | 57.83 | 57.83 | 2 | 56.22 | 56.15 | 56.15 | 56.02 | 4 | 59.55 | 59.90 |
| 75 | 57.82 | 57.82 | 57.82 | 57.82 | 3 | 56.21 | 56.15 | 56.15 | 56.15 | 6 | 59.89 | 59.32 |
| 100 | 57.81 | 57.81 | 57.81 | 57.81 | 4 | 56.20 | 56.14 | 56.14 | 56.24 | 8 | 59.46 | 59.59 |
| 125 | 57.81 | 57.81 | 57.81 | 57.81 | 5 | 56.19 | 56.13 | 56.13 | 56.17 | 10 | 59.26 | 59.15 |
| Total | 2.66 | 3.36 | 3.42 | **1.08** | Total | 101.90 | 87.79 | 88.90 | **11.69** | Total | 6764.31 | **54.79** |
| Prop | 1.73 | 2.78 | 2.91 | **0.85** | Prop | 98.48 | 86.81 | 87.95 | **10.81** | Prop | 6703.44 | **9.02** |

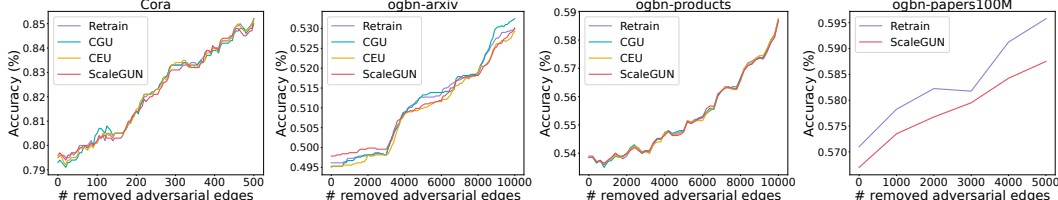

Figure 3: Comparison of unlearning efficacy for **linear** models: Model accuracy v.s. the number of removed adversarial edges.

randomly choose a set of low-degree nodes from the test set and then select edges that connect these nodes to other low-degree nodes bearing identical labels. Table 1 presents the test accuracy, average total unlearning cost, and average propagation cost for batch edge unlearning in linear models applied to large graph datasets. The results highlight ScaleGUN's impressive performance, with its advantages becoming more pronounced as the graph size increases.

Furthermore, Table 2 shows the node and feature unlearning results on ogbn-papers100M, where 2000 nodes are removed at a time. The results demonstrate that ScaleGUN achieves competitive accuracy compared to the retrained model while significantly reducing the total unlearning and propagation costs under the node feature and node unlearning scenarios. Notably, on the ogbn-papers100M dataset, ScaleGUN demonstrates a speed advantage of $1000\times$ over retraining in terms of propagation cost under both edge and feature unlearning scenarios.

Table 2: Test accuracy (%), total unlearning cost (s) and propagation cost (s) per node feature/node removal for **linear** models on ogbn-papers100M.

| Feature Unlearning | | | Node Unlearning | | |
|---|---|---|---|---|---|
| $N(\times 10^3)$ | Retrain | ScaleGUN | $N(\times 10^3)$ | Retrain | ScaleGUN |
| 0 | 59.99 | 59.72 | 0 | 59.99 | 59.72 |
| 2 | 59.99 | 59.72 | 2 | 59.99 | 59.75 |
| 4 | 59.99 | 59.65 | 4 | 59.99 | 59.58 |
| 6 | 59.99 | 59.47 | 6 | 59.99 | 59.80 |
| 8 | 59.99 | 59.54 | 8 | 59.99 | 59.56 |
| 10 | 59.99 | 59.45 | 10 | 60.00 | 59.63 |
| Total | 5400.45 | **45.29** | Total | 5201.88 | **60.08** |
| Prop | 5352.84 | **6.89** | Prop | 5139.09 | **21.61** |

**Efficiency and model utility on deep models.** ScaleGUN can also achieve superior unlearning performance on deep models, such as decoupled models and spectral GNNs (Chien et al., 2020; He et al., 2022), despite lacking certified guarantees. Table 3 presents ScaleGUN's performance on decoupled models, utilizing a structure where two-hop propagations are followed by a two-layer Multi-Layer Perceptron (MLP). The formulation is expressed as $\hat{\mathbf{Y}} = \text{LogSoftmax}(\sigma(\mathbf{P}^2\mathbf{X}\mathbf{W})\mathbf{W})$. We introduce perturbation $\mathbf{B}^\top\mathbf{W}$ to each learnable parameter $\mathbf{W}$, utilizing Adam as the optimizer.

The propagation cost is excluded as its pattern is consistent with that in Table 1. This suggests that ScaleGUN can be employed in shallow networks to achieve higher model accuracy when certified guarantee are not required. Furthermore, spectral GNNs are a significant subset of GNNs rooted in spectral graph theory, known for their superior expressive power. We also apply ScaleGUN on the spectral GNN model to demonstrate its versatility in Appendix C.

Table 3: Test accuracy (%), total unlearning cost ($s$) per batch edge removal for **decoupled** models.

| | ogbn-products | | | ogbn-papers100M | |
|---|---|---|---|---|---|
| $N(\times 10^3)$ | Retrain | ScaleGUN | $N(\times 10^3)$ | Retrain | ScaleGUN |
| 0 | 74.16 | 74.25 | 0 | 63.39 | 63.13 |
| 1 | 74.15 | 74.25 | 2 | 63.21 | 63.05 |
| 2 | 74.16 | 74.24 | 4 | 63.13 | 62.97 |
| 3 | 74.12 | 74.24 | 6 | 63.05 | 62.89 |
| 4 | 74.18 | 74.23 | 8 | 62.95 | 62.80 |
| 5 | 74.10 | 74.22 | 10 | 62.85 | 62.72 |
| Total | 174.23 | **14.19** | Total | 7958.83 | **10.49** |

**Edge unlearning efficacy.** We adopt the task of forgetting adversarial data to evaluate ScaleGUN's edge unlearning efficacy. First, we introduce the measurement for edge unlearning efficacy. Specifically, we introduce adversarial edges into the graph, ensuring the terminal nodes bear different labels.

Table 4: Deleted Data Replay Test: The ratio of incorrectly labeled nodes, $r_d, r_a$, after unlearning.

| Model | $r_d = \frac{|\{i \in \mathcal{V}_d|\hat{y}_i=c\}|}{|\mathcal{V}_d|}$ (%, ↓) | | | $r_a = \frac{|\{i \in \mathcal{V}|\hat{y}_i=c\}|}{|\mathcal{V}|}$ (%, ↓) | | |
|---|---|---|---|---|---|---|
| | Cora | ogbn-arxiv | ogbn-products | Cora | ogbn-arxiv | ogbn-products |
| Origin | 100 | 90.72 | 100 | 52.13 | 1.01 | 45.59 |
| Retrain | 0 | 0 | 0 | 0 | 0 | 0 |
| CGU | 0 | 0 | 0 | 0.09 | 0 | 0 |
| ScaleGUN | 0 | 0 | 0 | 0.08 | 0 | 0 |

These adversarial edges are used to deceive the model into making incorrect predictions, diminishing its performance. The objective of unlearning is to delete these adversarial edges and recuperate the model's performance. With the removal of more adversarial edges, the accuracy of unlearning methods is expected to rise, following the trend of retraining. For Cora, we randomly selected 500 edges connecting two distinct labeled terminal nodes to serve as adversarial edges. However, this straightforward method proves inadequate in affecting model accuracy for large graphs. Consequently, we adopt a more refined approach tailored for large graphs, which is detailed in Appendix C. Figure 3 illustrates how the model accuracy varies as the number of removed adversarial edges increases. The results affirm that CGU, CEU, and ScaleGUN exhibit effectiveness in edge unlearning.

**Node unlearning efficacy.** To measure the node unlearning efficacy, we conduct Deleted Data Replay Test on Cora, ogbn-arxiv, and ogbn-products for linear models in Table 4. First, we choose a set of nodes $\mathcal{V}_d$ to be unlearned from the training set and add 100-dimensional binary features to the original node features. For nodes in $\mathcal{V}_d$, these additional features are set to 1; for other nodes, they are set to 0. The labels of nodes in $\mathcal{V}_d$ are assigned to a new class $c$. An effective unlearning method is expected not to predict the unlearned nodes as class $c$ after unlearning $\mathcal{V}_d$, meaning that $\hat{y}_i \neq c$ for $i \in \mathcal{V}_d$. We report the ratio of incorrectly labeled nodes $r_d = \frac{|\{i \in \mathcal{V}_d|\hat{y}_i=c\}|}{|\mathcal{V}_d|}$ in the original model (which does not unlearn any nodes) and target models after unlearning. We set $|\mathcal{V}_d| = 100, 125, 50000$ for Cora, ogbn-arxiv, and ogbn-products, respectively. We also report $r_a = \frac{|\{i \in \mathcal{V}|\hat{y}_i=c\}|}{|\mathcal{V}|}$ to assess whether the impact of the unlearned nodes is completely removed from the graph. The results show that ScaleGUN successfully removes the impact of the unlearned nodes, with only a slight discrepancy compared to retraining. Moreover, we also evaluate unlearning efficacy by Membership Inference Attack (MIA), however, as argued in (Chien et al., 2022a), MIA is not suitable to evaluate graph unlearning efficacy. We defer the MIA results to Appendix C.

**Effects of $r_{\max}$.** ScaleGUN introduces the parameter $r_{\max}$ to manage the approximation error. To investigate the impact of $r_{\max}$, we conduct experiments on the ogbn-arxiv dataset, removing 100 random edges, one at a time. Table 5 displays the initial model accuracy before any removal, total unlearning cost, propagation cost per edge removal, and average number of retraining throughout the unlearning process. Decreasing $r_{\max}$ improves model accuracy and reduces the approximation error, leading to fewer retraining times and lower unlearning costs.

Beyond a certain threshold, the model accuracy stabilizes, and the unlearning cost is minimized. Further reduction in $r_{\max}$ is unnecessary as it would increase the initial computational cost. Notably, the propagation cost remains stable, consistent with Theorem 3.3, which indicates that the average propagation cost is independent of $r_{\max}$. Table 5 suggests that selecting an appropriate $r_{\max}$ can achieve both high model utility and efficient unlearning.

Table 5: Initial test accuracy, total unlearning cost and propagation cost per edge removal by varying $r_{\max}$ on ogbn-arxiv with linear model.

| $r_{\max}$ | 1e-5 | 1e-7 | 1e-9 | 5e-10 | 1e-10 | 1e-15 |
|---|---|---|---|---|---|---|
| Acc (%) | 55.23 | 57.84 | 57.84 | 57.84 | 57.84 | 57.84 |
| Total (s) | 3.30 | 3.60 | 3.31 | 1.81 | 0.92 | 0.93 |
| Prop (s) | 0.72 | 0.76 | 0.75 | 0.71 | 0.77 | 0.77 |
| #Retrain | 100 | 100 | 85.33 | 56.67 | 0 | 0 |

## 6 CONCLUSION

This paper introduces ScaleGUN, the first certified graph unlearning mechanism that scales to billion-edge graphs. We introduce the approximate propagation technique from decoupled models into certified unlearning and reveal the impact of approximation error on the certified unlearning guarantees by non-trivial theoretical analysis. Certified guarantees are established for all three graph unlearning scenarios: node feature, edge, and node unlearning. Empirical studies of ScaleGUN on real-world datasets showcase its efficiency, model utility, and unlearning efficacy in graph unlearning.

## ACKNOWLEDGEMENTS

The work was partially done at Gaoling School of Artificial Intelligence, Beijing Key Laboratory of Big Data Management and Analysis Methods, MOE Key Lab of Data Engineering and Knowledge Engineering, and Pazhou Laboratory (Huangpu), Guangzhou, Guangdong 510555, China. This research was supported in part by National Science and Technology Major Project (2022ZD0114802), by National Natural Science Foundation of China (No. 92470128, No. U2241212), by Beijing Outstanding Young Scientist Program No.BJJWZYJH012019100020098, by Huawei-Renmin University joint program on Information Retrieval. We also wish to acknowledge the support provided by the fund for building world-class universities (disciplines) of Renmin University of China, by Engineering Research Center of Next-Generation Intelligent Search and Recommendation, Ministry of Education, by Intelligent Social Governance Interdisciplinary Platform, Major Innovation & Planning Interdisciplinary Platform for the "Double-First Class" Initiative, Public Policy and Decision-making Research Lab, and Public Computing Cloud, Renmin University of China.

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

## A    OTHER RELATED WORKS

Machine unlearning, first introduced by Cao & Yang (2015), aims to eliminate the impact of selected training data from trained models to enhance privacy protection.

**Exact unlearning.**  Exact unlearning seeks to generate a model that mirrors the performance of a model retrained from scratch. The straightforward method is to retrain the model upon a data removal request, which is often deemed impractical due to high computational and temporal costs. To bypass the burdensome retraining process, several innovative solutions have been proposed. Ginart et al. (2019) focused on $k$-means clustering unlearning, while Karasuyama & Takeuchi (2010) addressed the unlearning problem for support vector machines. Bourtoule et al. (2021) proposed the SISA (sharded, isolated, sliced, and aggregated) method, which partitions the training data into shards and trains on each shard separately. Upon a removal request, only the affected shards need to be retrained, thus enhancing the performance significantly. GraphEraser (Chen et al., 2022) extended the SISA approach to graph-structured data. Projector and GraphEditor (Cong & Mahdavi; 2023) address exact unlearning for linear GNNs with Ridge regression as their objective.

**Approximate unlearning.**  Approximate unlearning, on the other hand, introduces probabilistic or heuristic methods to reduce unlearning costs further. Within this realm, certified unlearning, the primary subject of this study, stands as a subclass of approximate methods distinguished by its probabilistic assurances. Besides the certified unlearning works previously mentioned, Pan et al. (2023) extended the certified unlearning to graph scattering transform. GIF (Wu et al., 2023a) developed graph influence function based on influence function for unstructured data (Koh & Liang, 2017). Golatkar et al. (2020) proposed heuristic-based selective forgetting in deep networks. GN-NDelete (Cheng et al., 2023) introduced a novel layer-wise operator for optimizing deleted edge consistency and neighborhood influence in graph unlearning. Concurrently, Li et al. (2024) achieved effective and general graph unlearning through a mutual evolution design, and Wang et al. (2023) proposed the first general framework for addressing the inductive graph unlearning problem.

**Differentially Privacy v.s. Certified unlearning.** Differentially Privacy (DP) aims to ensure that an adversary cannot infer whether the model was trained on the original dataset or the dataset with any **single data** sample removed, based on the model's output. Certified unlearning aims to remove the impact of the **specific** data sample(s) so that the model behaves as if the sample was never included. Thus, a DP model inherently provides certified unlearning for any single data sample. However, most DP models suffer from performance degradation even for loose privacy constraints (Abadi et al., 2016; Chaudhuri et al., 2011). Certified unlearning can balance model utility and computational cost, presenting an alternative to retraining and DP (Chien et al., 2022b). This also explains the growing interest in certified unlearning as privacy protection demands rise. Additionally, there are some nuances between DP and certified unlearning. For example, DP does not obscure the total dataset size and the number of deletions any given model instance has processed, but this should not be leaked in certified unlearning (Ginart et al., 2019).

## B    DETAILS OF CERTIFIABLE UNLEARNING MECHANISM

### B.1    PRACTICAL ASPECTS

**Least-squares and logistic regression on graphs.**  Similar to (Chien et al., 2022a), our unlearning mechanism can achieve certifiable unlearning with least-squares and binary logistic

regression. ScaleGUN performs exact unlearning for least-squares since the Hessian of loss function is independent of $\mathbf{w}$. For binary logistic regression, we define the empirical risk as $l(\mathbf{e}_i^\top \hat{\mathbf{Z}} \mathbf{w}, \mathbf{e}_i^\top \mathbf{Y}_i) = -\log(\sigma(\mathbf{e}_i^\top \mathbf{Y}_i \mathbf{e}_i^\top \hat{\mathbf{Z}} \mathbf{w}))$, where $\sigma$ denotes the sigmoid function. As shown in (Guo et al., 2019; Chien et al., 2022a), Assumption 4.1 holds with $c = 1, \gamma_1 = \gamma_2 = 1/4, c_1 = 1$ for logistic regression. Following (Chien et al., 2022a), ScaleGUN can adapt the "one-versus-all other classes" strategy for multiclass logistic regression.

**Batch unlearning.** In the batch unlearning scenario, multiple instances may be removed at a time. In the worst-case bounds of the gradient residual norm, the component attributable to approximation error remains unchanged across three kinds of unlearning requests and for any number of unlearning instances. This stability is due to our lazy local propagation framework, which ensures the approximation error $\epsilon_1$ does not vary with the number of removed instances. Regarding the component related to unlearning, which is mainly determined by $\|\mathbf{Z}\mathbf{e}_j - \mathbf{Z}'\mathbf{e}_j\|$, we can establish the corresponding bounds by aggregating the bounds for each unlearning instance, utilizing the Minkowski inequality. The data-dependent bound remains unchanged and can be computed directly.

**Limitations of existing certified unlearning mechanisms.** The existing certified unlearning studies, including ScaleGUN, are limited to linear models with a strongly convex loss function. Achieving certified graph unlearning in nonlinear models is a significant yet challenging objective. Existing approximate unlearning methods on deep models are heuristics without theoretical guarantees (Golatkar et al., 2020; McAuley et al., 2015). On the one hand, the theoretical foundation of existing certified unlearning mechanisms is the Newton update. However, the thorough examination of the Newton update in deep networks remains an unresolved issue (Xu et al., 2018). Koh & Liang (2017) demonstrates that the Newton update performs well in non-convex shallow CNN networks. However, performance may decline in deeper architectures as the convexity of the loss function is significantly compromised (Wu et al., 2023a). On the other hand, note that a certified unlearning model is approximately equivalent to retraining in terms of their probability distributions. This indicates that the unlearning model must be able to approximate the new optimal point of empirical risk within a certain margin of error after data removal. However, existing studies (Wang & Pilanci, 2023) have proven that even approximating the optimal value of a 2-layer ReLU neural network is NP-hard in the worst cases. Therefore, the path to certified unlearning in nonlinear models remains elusive, even for unstructured data. While ScaleGUN cannot achieve certified unlearning for deep models, our empirical results demonstrate that ScaleGUN can achieve competitive model utility using deep models as backbones when certified guarantees are not required, as shown in Table 3 and Table 13. In summary, ScaleGUN can perform certified unlearning for linear GNNs and serves as an effective heuristic unlearning method for deep models.

**Potential improvements and future directions.** As suggested in (Guo et al., 2019; Chien et al., 2022a), pretraining a nonlinear feature extractor on public datasets can significantly improve overall model performance. If no public datasets are available, a feature extractor with differential privacy (DP) guarantees can be designed. DP-based methods provide privacy guarantees for nonlinear models, indicating the potential to leverage DP concepts to facilitate certified unlearning in deep models. We also observe that ScaleGUN is capable of data removal from deep models to a certain extent (see Figure 7 in Appendix C), even without theoretical guarantees. This observation suggests the possibility of formulating more flexible certification criteria to assess the efficacy of current heuristic approaches.

### B.2 DETAILS OF SEQUENTIAL UNLEARNING ALGORITHM

According to the privacy requirement and Theorem 2.1, we select the noise standard deviation $\alpha$, privacy parameters $\epsilon, \delta$ and compute the "privacy budget" $\alpha\epsilon/\sqrt{2\log(1.5/\delta)}$. Initially, compute the approximate embeddings by the lazy local propagation framework and train the model from scratch. For each unlearning request, update the embeddings and then employ our unlearning mechanism. Specifically, compute the first component of the data-dependent bound, i.e., $2c_1 \left\| \mathbf{1}^\top \mathbf{R} \right\|$ in Theorem 4.5 and accumulate the second component $\gamma_2 \left\| \hat{\mathbf{Z}}' \right\| \left\| \hat{\mathbf{H}}_{\mathbf{w}^\star}^{-1} \Delta \right\| \left\| \hat{\mathbf{Z}}' \hat{\mathbf{H}}_{\mathbf{w}^\star}^{-1} \Delta \right\|$ for each unlearning request. Once the budget is exhausted, retrain the model from scratch. Note that we do not need to re-propagate even when retraining the model. Algorithm 1 provides the pseudo-code of ScaleGUN, where the removal request $R_i$ can be one instance or a batch of instances.

---

**Algorithm 1:** ScaleGUN

---

**Input:** Graph data $\mathcal{D} = (\mathbf{X}, \mathbf{Y}, \mathbf{A})$, sequence of removal requests $R = \{R_1, \cdots, R_k\}$, loss function $l$, parameters $L, \{w_\ell\}, r_{\max}, \alpha, \gamma_2, \epsilon, \delta$

1  Compute the embedding matrix $\hat{\mathbf{Z}}$ by Algorithm 2 with initialized $\{\boldsymbol{q}^{(\ell)}\}$ and $\{\boldsymbol{r}^{(\ell)}\}$ ;

2  $\mathbf{w} \leftarrow$ the model trained on the training set of $\hat{\mathcal{D}}$ with the approximate embeddings $\hat{\mathbf{Z}}$;

3  Accumulated unlearning error $\beta \leftarrow 0$;

4  **for** $R_i \in R$ **do**

5     $\hat{\mathbf{Z}}', \boldsymbol{R} \leftarrow$ the embedding matrix updated by the lazy local propagation framework and the corresponding residue matrix according to the removal request $R_i$;

6     $\hat{\mathcal{D}}' \leftarrow$ the updated dataset according to $R_i$;

7     $\Delta \leftarrow \nabla \mathcal{L}(\mathbf{w}, \hat{\mathcal{D}}) - \mathcal{L}(\mathbf{w}, \hat{\mathcal{D}}')$;

8     $\hat{\mathbf{H}}_{\mathbf{w}} \leftarrow \nabla^2 \mathcal{L}(\mathbf{w}, \hat{\mathcal{D}}')$;

9     $\beta \leftarrow \beta + \gamma_2 \left\| \hat{\mathbf{Z}}' \right\| \left\| \hat{\mathbf{H}}_{\mathbf{w}^\star}^{-1} \Delta \right\| \left\| \hat{\mathbf{Z}}' \hat{\mathbf{H}}_{\mathbf{w}^\star}^{-1} \Delta \right\|$;

10     **if** $\beta + 2c_1 \left\| \mathbf{1}^\top \boldsymbol{R} \right\| > \alpha\epsilon / \sqrt{2 \log(1.5/\delta)}$ **then**

11        $\mathbf{w} \leftarrow$ the model retrained on the training set of $\hat{\mathcal{D}}'$;

12        $\beta \leftarrow 0$;

13     **else**

14        $\mathbf{w} = \mathbf{w} + \hat{\mathbf{H}}_{\mathbf{w}}^{-1} \Delta$;

---

Table 6: Statistics of datasets.

| Dataset | $n$ | $m$ | $|C|$ | $F$ | train/val/test |
|---|---|---|---|---|---|
| Cora | 2708 | 5278 | 7 | 1433 | 1208/500/1000 |
| Citeseer | 3327 | 4552 | 6 | 3703 | 1827/500/1000 |
| Photo | 7650 | 119081 | 8 | 745 | 6150/500/1000 |
| ogbn-arxiv | 169,343 | 1,166,243 | 40 | 128 | 90941/29799/48603 |
| ogbn-products | 2,449,029 | 61,859,140 | 47 | 128 | 196615/39323/2213091 |
| ogbn-papers100M | 111,059,956 | 1,615,685,872 | 172 | 128 | 1,207,179/125,265/125,265 |

## C  DETAILS OF EXPERIMENTS AND ADDITIONAL EXPERIMENTS

### C.1  EXPERIMENTS SETTINGS

We use cumulative power iteration to compute exact embeddings for the retrain method, CGU, and CEU. Each dimension of the embedding matrix is computed in parallel with a total of 40 threads. We adopt the "one-versus-all" strategy to perform multi-class node classification following (Chien et al., 2022a). We conduct all experiments on a machine with an Intel(R) Xeon(R) Silver 4114 CPU @ 2.20GHz CPU, 1007GB memory, and Quadro RTX 8000 GPU in Linux OS. The statistics of the datasets are summarized in Table 6. Table 7 and Table 8 summarize the parameters used in the linear and deep experiments, respectively. The parameters are consistently utilized across all methods.

**Adversarial edges selection.** In Figure 3, we show the model accuracy as the number of removed adversarial edges increases. We add 500 adversarial edges to Cora, with 5 random edges being removed in each removal request. For ogbn-arxiv, ogbn-products, and ogbn-papers100M, we add $10^4$ adversarial edges, with batch-unlearning size as 100, 200, and 2000, respectively. The details of selecting adversarial edges are as follows. For the Cora dataset, we randomly select edges connecting two distinct labeled terminal nodes and report the accuracy on the whole test set. For large datasets, randomly selecting adversarial edges proves insufficient to impact model accuracy. Hence, we initially identify a small set of nodes from the test set, then add adversarial edges by linking these nodes to other nodes with different labels. Specifically, we first randomly select 2000 nodes from the original test set to create a new test set $\mathcal{V}_t$. We then repeat the following procedure until we collect enough

Table 7: Parameters used in the linear model experiments.

| Dataset | $r_{\max}$ | $\lambda$ | $\alpha$ |
|---|---|---|---|
| Cora | 1e-7 | 1e-2 | 0.1 |
| Citeseer | 1e-8 | 1e-2 | 0.1 |
| Photo | 1e-8 | 1e-4 | 3.0 |
| ogbn-arxiv | 1e-8 | 1e-4 | 0.1 |
| ogbn-products | 1e-8 | 1e-4 | 0.1 |
| ogbn-papers100M | 1e-9 | 1e-8 | 15.0 |

Table 8: Parameters used in the deep model experiments.

| Dataset | $\lambda$ | $\alpha$ | learning rate | hidden size | batch size |
|---|---|---|---|---|---|
| ogbn-arxiv | 5e-4 | 0.1 | 1e-3 | 1024 | 1024 |
| ogbn-products | 1e-4 | 0.01 | 1e-4 | 1024 | 512 |
| ogbn-papers100M | 1e-8 | 5.0 | 1e-4 | 256 | 8192 |

edges: randomly select a node $u$ from $\mathcal{V}_t$, randomly select a node $v$ from the entire node set $\mathcal{V}$, and if $u$ and $v$ have different labels, we add the adversarial edge $(u, v)$ to the original graph.

**Vulnerable edges selection.** In Table 1, we show the unlearning results in large graph datasets. For these datasets, we selected *vulnerable edges* connected to low-degree nodes in order to observe changes in test accuracy after unlearning. The details of selecting vulnerable edges are as follows. We first determine all nodes with a degree below a specified threshold in the test set. Specifically, the threshold is set to 10 for the ogbn-arxiv and ogbn-products datasets, and 6 for the ogbn-papers100M dataset. We then shuffle the set of these low-degree nodes and iterate over each node $v$: For each node $v$, if any of its neighbors share the same label as $v$, the corresponding edge is included in the unlearned edge set. The iteration ends until we collect enough edges.

## C.2  ADDITIONAL EXPERIMENTS ON NODE FEATURE, EDGE AND NODE UNLEARNING.

**Node feature and node unlearning experiments.** We conduct experiments on node feature and node unlearning for linear models and choose Cora, ogbn-arixv, and ogbn-papers100M as the representative datasets. Following the setting in (Chien et al., 2022b), we set $\delta = 1e - 4$. Table 9 shows the model accuracy, average total cost, and average propagation cost per removal on Cora and ogbn-arxiv. For Cora, we remove the feature of one node at a time for node feature unlearning tasks. While for ogbn-arxiv, we remove the feature of 25 nodes. The node unlearning task is set in the same way. The results demonstrate that ScaleGUN achieves comparable model utility with retraining while significantly reduce the unlearning costs in node feature and node unlearning tasks.

**Edge unlearning results on small graphs.** Table 10 presents test accuracy, average total unlearning cost, and average propagation cost for edge unlearning in linear models applied to small graph datasets. For each dataset, 2000 edges are removed, with one edge removed at a time. The table reports the initial and intermediate accuracy after every 500-edge removal. In this experiment, we also set $\delta = 1e - 4$, following the setting in (Chien et al., 2022b). ScaleGUN is observed to maintain competitive accuracy when compared to CGU and CEU while notably reducing both the total and propagation costs. It is important to note that the difference in unlearning cost (Total$-$Prop) among the three methods is relatively minor. However, CGU and CEU incur the same propagation cost as retraining, which is $10\times \sim 30\times$ higher than that of ScaleGUN.

**Model utility when removing more than 50% training nodes.** To validate the model utility when the unlearning ratio reaches 50%, we remove $5 \times 10^4$ and $10^5$ training nodes/features from ogbn-arxiv and ogbn-products in Figure 4. The batch-unlearning sizes are set to 25 and $10^3$ nodes/features, respectively. We set $\delta = 1e - 4$ in the experiment, following the setting in (Chien et al., 2022b). The results show that ScaleGUN closely matches the retrained model's utility even when half of the training nodes/features are removed.

Table 9: Test accuracy (%), total unlearning cost (s) and propagation cost (s) per node feature/feature removal for **linear** models on Cora and ogbn-arxiv.

| | Feature Unlearning | | | | | | | Node Unlearning | | | | | | | |
| | Cora | | | ogbn-arxiv | | | | Cora | | | ogbn-arxiv | | | |
| $N$ | Retrain | CGU | ScaleGUN | $N$ | Retrain | CGU | ScaleGUN | $N$ | Retrain | CGU | ScaleGUN | $N$ | Retrain | CGU | ScaleGUN |
|---|---|---|---|---|---|---|---|---|---|---|---|---|---|---|---|
| 0 | 84.9 | 84.9 | 84.9 | 0 | 57.84 | 57.84 | 57.84 | 0 | 84.9 | 84.9 | 84.9 | 0 | 57.84 | 57.84 | 57.84 |
| 200 | 84.17 | 83.4 | 83.4 | 25 | 57.83 | 57.84 | 57.84 | 200 | 83.5 | 82.97 | 82.97 | 25 | 57.83 | 57.83 | 57.84 |
| 400 | 84.4 | 82.43 | 82.43 | 50 | 57.83 | 57.83 | 57.83 | 400 | 82.43 | 81.53 | 81.53 | 50 | 57.83 | 57.83 | 57.83 |
| 600 | 83.03 | 81.73 | 81.73 | 75 | 57.84 | 57.84 | 57.84 | 600 | 81.87 | 80.7 | 80.7 | 75 | 57.84 | 57.84 | 57.84 |
| 800 | 81.9 | 77.83 | 77.83 | 100 | 57.83 | 57.85 | 57.84 | 800 | 80.1 | 76.7 | 76.7 | 100 | 57.83 | 57.84 | 57.84 |
| $T_1$ | 0.44 | 0.20 | 0.10 | $T_1$ | 2.59 | 2.33 | 1.49 | $T_1$ | 0.44 | 0.20 | 0.11 | $T_1$ | 2.66 | 2.04 | 1.38 |
| $T_2$ | 0.11 | 0.10 | 0.01 | $T_2$ | 1.55 | 1.56 | 0.77 | $T_2$ | 0.10 | 0.10 | 0.01 | $T_2$ | 1.66 | 1.43 | 0.81 |

Table 10: Test accuracy (%), total unlearning cost (s), and propagation cost (s) per one-edge removal for **linear** models (small graphs).

| | Cora | | | | | Citeseer | | | | | Photo | | | |
| $N(\times 500)$ | Retrain | CGU | CEU | ScaleGUN | $N(\times 500)$ | Retrain | CGU | CEU | ScaleGUN | $N(\times 500)$ | Retrain | CGU | CEU | ScaleGUN |
|---|---|---|---|---|---|---|---|---|---|---|---|---|---|---|
| 0 | 85.30 | 84.10 | 84.10 | 84.10 | 0 | 79.30 | 78.80 | 78.80 | 78.80 | 0 | 91.67 | 89.93 | 89.93 | 89.93 |
| 1 | 85.20 | 83.30 | 83.30 | 83.30 | 1 | 78.60 | 78.37 | 78.37 | 78.37 | 1 | 91.60 | 89.90 | 89.90 | 89.90 |
| 2 | 84.10 | 82.30 | 82.30 | 82.30 | 2 | 78.30 | 78.00 | 78.00 | 78.00 | 2 | 91.60 | 89.90 | 89.90 | 89.90 |
| 3 | 83.00 | 81.80 | 81.80 | 81.80 | 3 | 77.93 | 77.43 | 77.43 | 77.43 | 3 | 91.57 | 89.67 | 89.67 | 89.67 |
| 4 | 82.40 | 81.40 | 81.40 | 81.40 | 4 | 77.60 | 77.10 | 77.10 | 77.10 | 4 | 91.63 | 89.67 | 89.67 | 89.67 |
| Total | 0.74 | 0.28 | 0.32 | **0.12** | Total | 0.99 | 0.81 | 0.76 | **0.55** | Total | 3.08 | 0.35 | 0.40 | **0.07** |
| Prop | 0.15 | 0.15 | 0.17 | **0.01** | Prop | 0.34 | 0.29 | 0.27 | **0.03** | Prop | 0.31 | 0.26 | 0.29 | **0.01** |

## C.3 PARAMETER STUDIES.

**Trade-off between privacy, model utility, and unlearning efficiency.** According to Theorem 2.1 and our sequential unlearning algorithm, there is a trade-off amongst privacy, model utility, and unlearning efficiency. Specifically, $\epsilon$ and $\delta$ controls the privacy requirement. To achieve $(\epsilon, \delta)$-certified unlearning, we can adjust the standard deviation $\alpha$ of the noise **b**. An overly large $\alpha$ introduces too much noise and may degrade the model utility. However, a smaller $\alpha$ may lead to frequent retraining and increase the unlearning cost due to the privacy budget constraint. We empirically analyze the trade-off between privacy and model utility, by removing 800 nodes from Cora, one at a time.

To control the frequency of retraining, we fix $\alpha\epsilon = 0.1$ and vary the standard derivation $\alpha$ of the noise **b**. The results are shown in Figure 5. Here, we set $\delta = 1/\#\text{nodes}$ and $r_{\max} = 10^{-10}$. We observe that the model accuracy decreases as $\alpha$ increases (i.e., $\epsilon$ decreases), which agrees with our intuition.

**Varying batch-unlearning size.** To validate the model efficiency and model utility under varying batch-unlearning sizes, we vary the batch-unlearning size in $\{1\times10^3, 3\times10^3, 5\times10^3, 10\times10^3\}$ for ogbn-products and $\{2 \times 10^3, 10 \times 10^3, 50\times10^3\}$ for ogbn-papers100M in Table 11. Following the setting in (Chien et al., 2022b), we set $\delta = 1e - 4$. The average total cost and

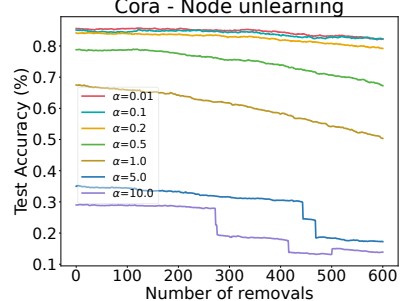

Figure 5: Varying $\alpha$ ($\alpha\epsilon = 0.1$ fixed): Test accuracy v.s. the number of removed nodes on Cora.

propagation cost for five batches and the accuracy after unlearning five batches are reported. For comparison, the corresponding retraining results are provided, where $1 \times 10^3$ and $2 \times 10^3$ nodes are removed per batch for ogbn-products and ogbn-papers100M, respectively. ScaleGUN maintains superior efficiency over retraining, even when the batch-unlearning size becomes 4%-5% of the training set in large graphs (e.g., $50 \times 10^3$ nodes in a batch out of $1.2 \times 10^6$ total training nodes of ogbn-papers100M). The test accuracy demonstrates that ScaleGUN preserves model utility under varying batch-unlearning sizes.

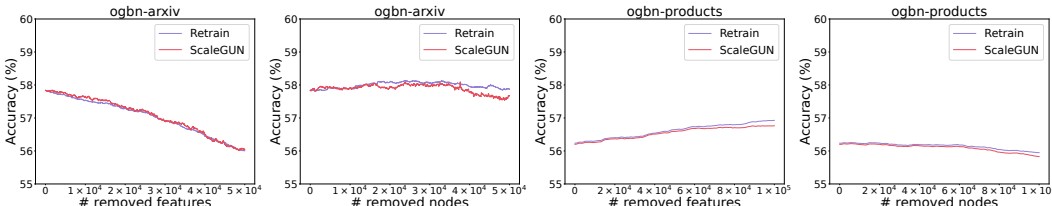

Figure 4: Unlearning more than 50% of training nodes/features: Test accuracy on ogbn-arxiv and ogbn-products.

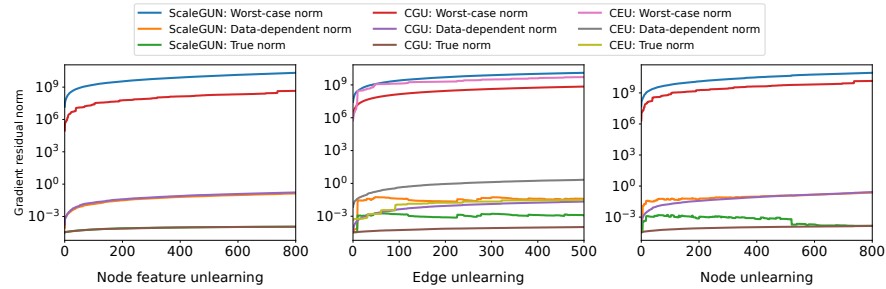

Figure 6: Comparison of the bounds of the gradient residual: Worst-case bound, data-dependent bound and the true value on Cora dataset for ScaleGUN, CGU and CEU.

### C.4 CGU, CEU AND SCALEGUN'S BOUNDS OF THE GRADIENT RESIDUAL.

Figure 6 shows the worst-case bound, data-dependent bound, and the true value of the gradient residual for ScaleGUN, CGU, and CEU on the Cora dataset. For ScaleGUN, CGU and CEU, the worst-case bounds and the data-dependent bounds validly upper bound the true value, and the worst-case bounds are looser than the data-dependent bounds. This confirms the correctness of the bounds for all methods. The worst-case bound of ScaleGUN is larger than that of CGU and slightly larger than that of CEU. This difference arises because ScaleGUN introduces approximation error, which is not present in CGU and CEU. However, note that this large worst-case bound does not impact the update efficiency of ScaleGUN, as decisions on retraining during sequential unlearning rely on data-dependent bounds rather than worst-case bounds. In edge and node unlearning scenarios, the data-dependent bound of ScaleGUN is slightly larger than that of CGU in the early stages of sequential unlearning. This occurs because the approximation error introduced by ScaleGUN dominates the model error when only a small amount of data has been unlearned. As the number of unlearning requests increases, the unlearning error accumulates, eventually dominating the model error, while the approximation error does not accumulate since we meet the approximation error bound in each unlearning step. Consequently, the data-dependent bound of ScaleGUN becomes comparable to that of CGU over time. In the edge unlearning scenario, the data-dependent bound of CEU is larger than that of both ScaleGUN and CGU, primarily due to CEU employing a different unlearning mechanism. The true norm of ScaleGUN is slightly larger than that of CGU in edge and node unlearning scenarios, again due to the introduction of approximation error by ScaleGUN.

### C.5 ADDITIONAL VALIDATION OF UNLEARNING EFFICACY.

**Unlearning efficacy on deep models.** We also evaluate the unlearning efficacy for deep models. Figure 7 illustrates how the model accuracy changes as the number of removed adversarial edges increases. The adversarial edges are the same as those used for linear models, selected according to the method detailed in Appendix C.1. The deep models employed are consistent with those in Table 3, defined as $\hat{Y} = \text{LogSoftmax}(\sigma(\mathbf{P}^2\mathbf{XW})\mathbf{W})$. The results indicate that ScaleGUN's performance improves with the removal of more adversarial edges, behaving similarly to retraining. This highlights ScaleGUN's ability to effectively unlearn adversarial edges, even in shallow non-linear networks.

**Membership Inference Attack (MIA).** As noted in (Chien et al., 2022b), MIA has distinct design goals from unlearning. Even after unlearning a training node, the attack model may still recognize the unlearned node in the training set if there are other similar training nodes.

Table 11: Varying batch-unlearning size: Test accuracy (%), average total cost ($s$), and average propagation cost ($s$) per batch node removal.

| | ogbn-products | | | | | ogbn-papers100M | | | |
|---|---|---|---|---|---|---|---|---|---|
| Metric | ScaleGUN | | | | Retrain | ScaleGUN | | | Retrain |
| | $1 \times 10^3$ | $3 \times 10^3$ | $5 \times 10^3$ | $10 \times 10^3$ | $1 \times 10^3$ | $2 \times 10^3$ | $10 \times 10^3$ | $50 \times 10^3$ | $2 \times 10^3$ |
| Acc | 56.25 | 56.25 | 56.07 | 56.14 | 56.24 | 59.49 | 59.71 | 58.76 | 59.45 |
| Total | 24.00 | 39.07 | 45.58 | 56.14 | 92.44 | 54.85 | 352.35 | 657.99 | 5201.88 |
| Prop | 21.97 | 37.25 | 43.91 | 54.72 | 91.54 | 15.49 | 313.11 | 621.62 | 5139.09 |

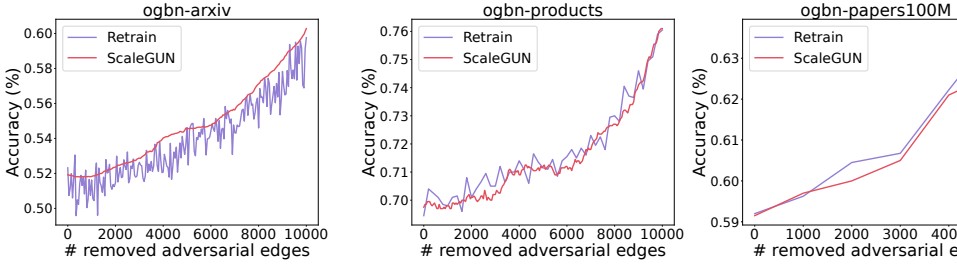

Figure 7: Comparison of unlearning efficacy for **deep** model: Model accuracy v.s. the number of removed adversarial edges.

Nonetheless, we follow (Chien et al., 2022b; Cheng et al., 2023) to conduct MIA (Olatunji et al., 2021) on Cora and ogbn-arxiv in Table 12. The core idea is that if the target model effectively unlearns a training node, the MI attacker will determine that the node is not present in the training set, i.e., the presence probability of the unlearned nodes decreases.

We remove 100 training nodes from Cora and 125 from ogbn-arxiv. We then report the ratio of the presence probability of these unlearned nodes between the original model (which does not unlearn any nodes) and target models after unlearning. A reported ratio greater than 1.0 indicates that the target model has successfully removed some information about the unlearned nodes, with higher values indicating better performance. Table 12 indicates that ScaleGUN offers privacy-preserving performance comparable to retraining in the context of MIA.

Table 12: MIA: The ratio of the presence probability of the unlearned nodes between the original model and target models after unlearning.

| Target model | Cora (↑) | ogbn-arxiv (↑) |
|---|---|---|
| Retrain | 1.698±0.21 | 1.232±0.388 |
| CGU | 1.526±0.120 | 1.231±0.388 |
| ScaleGUN | 1.524±0.130 | 1.232±0.388 |

## C.6 APPLY SCALEGUN TO SPECTRAL GNNS.

Besides decoupled GNNs that incorporate MLPs, ScaleGUN can also be applied to spectral GNNs (Chien et al., 2020; He et al., 2022), which are a significant subset of existing GNN models and have shown superior performance in many popular benchmarks. For example, ChebNetII (He et al., 2022) can be expressed as $Y = f_\theta(Z), Z = \sum_{k=0}^{K} w_k T_k(\hat{L})X$ when applied to large graphs, where $T_k$ are the Chebyshev polynomials, $f_\theta$ is a MLP and $w_k$ are learnable parameters. We can maintain $T_k(\hat{L})X$ using the lazy local framework and perform unlearning on $f_\theta$ and $w_k$, similar to that of Table 3. Table 13 demonstrated the unlearning results using ChebNetII as the backbone on ogbn-arxiv, mirroring the unlearning settings with Table 1. This suggests that ScaleGUN can achieve competitive performance when no certified guarantees are required.

Table 13: Test accuracy (%), total unlearning cost (s), and propagation cost (s) per batch edge removal using **ChebNetII** as the backbone on ogbn-arxiv.

| | Retrain | ScaleGUN |
|---|---|---|
| 0 | 71.99 | 71.99 |
| 25 | 72.08 | 71.98 |
| 50 | 72.01 | 71.98 |
| 75 | 71.8 | 71.96 |
| 100 | 72.3 | 71.94 |
| 125 | 71.79 | 71.93 |
| Prop | 14.96 | 3.13 |
| Total | 58.51 | 5.84 |

**Licenses of the datasets.** Ogbn-arxiv and ogbn-papers100M are both under the ODC-BY license. Cora, Citeseer, Photo, and ogbn-products are under the CC0 1.0 license, Creative Commons license, Open Data Commons Attribution License, and Amazon license, respectively.

# D  DETAILS OF LAZY LOCAL PROPAGATION FRAMEWORK

## D.1  DIFFERENCES BETWEEN SCALEGUN AND EXISTING DYNAMIC PROPAGATION METHODS

Existing dynamic propagation methods focus on the PPR-based propagation scheme, while ScaleGUN adopts the GPR-based one. Note that these dynamic propagation methods based on PPR cannot be straightforwardly transformed to accommodate GPR. Specifically, PPR-based methods can simply maintain a reserve vector $q$ and a residue vector $r$. In contrast, GPR-based propagation requires $q^{(\ell)}$ and $r^{(\ell)}$ for each level $\ell$. This difference also impacts the theoretical analysis of correctness and time complexity. Therefore, we have developed tailored algorithms and conducted a comprehensive theoretical analysis of the GPR-based scheme.

Choosing GPR enhances the generalizability of our framework. Specifically, our framework computes the embedding $\mathbf{Z} = \sum_{\ell=0}^{L} w_\ell \mathbf{P}^\ell \mathbf{X}$, wherein PPR, the weights $w_\ell$ are defined as $\alpha(1-\alpha)^\ell$ with $\alpha$ being the decay factor and $L$ tending towards infinity. Such a formulation allows the theoretical underpinnings of ScaleGUN to be directly applicable to PPR-based models, illustrating the broader applicability of our approach. One could replace the propagation part in ScaleGUN with the existing dynamic PPR approaches and obtain a certifiable unlearning method for PPR-based models. Conversely, confining our analysis to PPR would notably limit the expansiveness of our results.

**Limitations of InstantGNN to achieve certified unlearning.** InstantGNN (Zheng et al., 2022) provides an incremental computation method for the graph embeddings of dynamic graphs, accommodating insertions/deletions of nodes/edges and modifications of node features. However, simply applying InstantGNN for unlearning does not lead to certified unlearning, as this requires a thorough analysis of the impact of approximation error on the certified unlearning guarantee. Certified unlearning mandates that the gradient residual norm on exact embeddings, i.e., $||\nabla L(w^-, D')||$, remains within the privacy limit, as detailed in Theorem 2.1. InstantGNN, which relies on approximate embeddings for training, lacks a theoretical analysis of $||\nabla L(w^-, D')||$. Whether InstantGNN retrains the model for each removal or employs an adaptive training strategy for better performance, it lacks solid theoretical backing. Thus, InstantGNN is considered a heuristic method without theoretical support.

## D.2  ALGORITHMS OF LAZY LOCAL PROPAGATION FRAMEWORK

**Initial propagation.** Algorithm 2 illustrates the pseudo-code of Forward Push on a graph $\mathcal{G}$ with $\mathbf{P} = \mathbf{A}\mathbf{D}^{-1}$ and $L$ propagation steps. Initially, we normalize $\mathbf{x}$ to ensure $\left\|\mathbf{D}^{\frac{1}{2}}\mathbf{x}\right\|_1 \leq 1$, and let $q^{(\ell)} = r^{(\ell)} = \mathbf{0}$ for $0 \leq \ell \leq L$, except for $r^{(0)} = \mathbf{D}^{\frac{1}{2}}\mathbf{x}$ when $\ell = 0$. After employing Algorithm 2, we obtain $\hat{\mathbf{z}} = \sum_{\ell=0}^{L} w_\ell \mathbf{D}^{-\frac{1}{2}} q^{(\ell)}$ as the approximation of $\mathbf{z}$.

---

**Algorithm 2:** BasicProp

**Input:** Graph $\mathcal{G}$, level $L$, threshold $r_{\max}$, and initialized $\{q^{(\ell)}, r^{(\ell)}\}, 0 \leq \ell \leq L$

**Output:** New estimated vectors $\{q^{(\ell)}, r^{(\ell)}\}$

1 **for** $\ell$ *from 0 to* $L - 1$ **do**
2      **for** $u \in V$ *with* $|r^{(\ell)}(u)| > r_{\max}$ **do**
3          **for** *each* $v \in \mathcal{N}(u)$ **do**
4              $r^{(\ell+1)}(v)+ = \frac{r^{(\ell)}(u)}{d(u)}$;
5          $q^{(\ell)}(u)+ = r^{(\ell)}(u)$ and $r^{(\ell)}(u) \leftarrow 0$ ;
6 $q^{(L)} \leftarrow q^{(L)} + r^{(L)}$ and $r^{(L)} \leftarrow \mathbf{0}^{n \times F}$;
7 **return** $\{q^{(\ell)}, r^{(\ell)}\}, 0 \leq \ell \leq L$

---

---

**Algorithm 3:** UpdateEmbeddingMatrix(EdgeRemoval)

---

**Input:** Graph $\mathcal{G}$, level $L$, edge $e = (u, v)$ to be removed, threshold $r_{\max}$, $\{w_\ell\}$ and $\{\boldsymbol{q}^{(\ell)}, \boldsymbol{r}^{(\ell)}\}$, $0 \le \ell \le L$

**Output:** $\tilde{\mathbf{Z}} \in \mathbb{R}^n$

1  $\mathcal{G}' \leftarrow$ delete $(u, v)$ from $\mathcal{G}$ ;

2  $\boldsymbol{r}^{(0)}(u) \leftarrow \boldsymbol{r}^{(0)}(u) + (d(u)^{\frac{1}{2}} - (d(u) + 1)^{\frac{1}{2}})\mathbf{x}(u)$;

3  $\boldsymbol{r}^{(0)}(v) \leftarrow \boldsymbol{r}^{(0)}(v) + (d(v)^{\frac{1}{2}} - (d(v) + 1)^{\frac{1}{2}})\mathbf{x}(v)$;

4  **for** $\ell$ *from* 1 *to* $L$ **do**

5      $\boldsymbol{r}^{(\ell)}(u) \leftarrow \boldsymbol{r}^{(\ell)}(u) - \frac{\boldsymbol{q}^{(\ell-1)}(v)}{d(v)+1}$;

6      **for** $w \in \mathcal{N}(u)$ **do**

7          $\boldsymbol{r}^{(\ell)}(w) \leftarrow \boldsymbol{r}^{(\ell)}(w) + \frac{\boldsymbol{q}^{(\ell-1)}(u)}{d(u)(d(u)+1)}$;

8      $\boldsymbol{r}^{(\ell)}(v) \leftarrow \boldsymbol{r}^{(\ell)}(v) - \frac{\boldsymbol{q}^{(\ell-1)}(u)}{d(u)+1}$;

9      **for** $w \in \mathcal{N}(v)$ **do**

10          $\boldsymbol{r}^{(\ell)}(w) \leftarrow \boldsymbol{r}^{(\ell)}(w) + \frac{\boldsymbol{q}^{(\ell-1)}(v)}{d(v)(d(v)+1)}$;

11  $\{\boldsymbol{q}^{(\ell)}, \boldsymbol{r}^{(\ell)}\} \leftarrow$ BasicProp$(\mathcal{G}', L, r_{\max}, \{\boldsymbol{q}^{(\ell)}, \boldsymbol{r}^{(\ell)}\})$;

12  **return** $\sum_{\ell=0}^{L} w_\ell \mathbf{D}^{-\frac{1}{2}} \boldsymbol{q}^{(\ell)}$

---

**Efficient Removal.** Algorithm 3 details the update process for edge removal. Upon a removal request, we first adjust $\{\boldsymbol{r}^{(\ell)}\}$ locally for the affected nodes to maintain the invariant property. Post-adjustment, the invariant property is preserved across all nodes. Then, Algorithm 2 is invoked to reduce the error further and returns the updated $\hat{\mathbf{z}}$. Note that the degree of node $u$ and node $v$ is revised in Line 1.

**Batch update.** Inspired by (Zheng et al., 2022), we introduce a parallel removal algorithm upon receiving a batch of removal requests. Specifically, we initially update the graph structure to reflect all removal requests and then compute the final adjustments for each affected node. Notably, the computation is conducted only once, thus significantly enhancing efficiency. We extend Algorithm 3 to accommodate batch-edge removal, simultaneously enabling parallel processing for multiple edges. The core concept involves adjusting both the reserves and residues for nodes impacted by the removal, deviating from Algorithm 3 where only the residues are modified. The benefit of altering the reserves is that only the reserves and residues for nodes $u$ and $v$ need updates when removing the edge $(u, v)$. Consequently, this allows the removal operations for all edges to be executed in parallel. Algorithm 4 details the batch update process for edge removal. Batch-node removal and batch-feature removal can be similarly implemented.

### D.3 THEORETICAL ANALYSIS OF LAZY LOCAL PROPAGATION FRAMEWORK

Denote $\{\Delta \boldsymbol{r}^{(\ell)}\}$ as the adjustments to residues before invoking Forward Push (Algorithm 2) for further reducing the error. For instance, we have $\Delta \boldsymbol{r}^{(\ell)}(u) = \frac{\boldsymbol{q}^{(\ell-1)}(v)}{d(v)+1}$ for $\ell > 0$. Let $d$ be the average degree of the graph. We introduce the following theorems regarding the time complexity of our lazy local propagation framework.

**Theorem D.1** (Initialization Cost, Update Cost, and Total Cost). *Let $\hat{\mathbf{z}}_0$ be the initial embedding vector generated from scratch. $\hat{\mathbf{z}}_i$ and $\{\boldsymbol{r}_i^{(\ell)}\}$ represents the approximate embedding vector and the residues after the $i$-th removal, respectively. Each embedding vector $\hat{\mathbf{z}}_i$ satisfies Lemma 3.1. Given the threshold $r_{\max}$, the cost of generating $\hat{\mathbf{z}}_0$ from scratch is $O\left(\left(L - \sum_{\ell=0}^{L-1}(L - \ell)\left\|\boldsymbol{r}_0^{(\ell)}\right\|_1\right) \cdot \frac{d}{r_{\max}}\right)$. The cost for updating $\hat{\mathbf{z}}_{i-1}$ and $\{\boldsymbol{r}_{i-1}^{(\ell)}\}$ to generate $\hat{\mathbf{z}}_i$ is*

$$O\left(d + \frac{d}{r_{\max}} \cdot \sum_{\ell=0}^{L-1}(L - \ell)\left(\left\|r_{i-1}^{(\ell)}\right\|_1 + \left\|\Delta \boldsymbol{r}_i^{(\ell)}\right\|_1 - \left\|r_i^{(\ell)}\right\|_1\right)\right),$$

---

**Algorithm 4:** BatchUpdate(EdgeRemoval)

---

**Input:** Graph $\mathcal{G}$, level $L$, edges to be unlearned $S = \{e_1, \cdots, e_k\}$, weight coefficients $w_\ell$,
   threshold $r_{\max}$, $(\boldsymbol{q}^{(\ell)}, \boldsymbol{r}^{(\ell)})$, $0 \le \ell \le L$

**Output:** $\hat{\mathbf{Z}} \in \mathbb{R}^n$

1 Update $\mathcal{G}$ according to $S$, and let $\Delta_u$ be the number of removed neighbors of node $u$;
2 $V_a \leftarrow \{u \mid \text{the degree of node } u \text{ has changed}\}$;
3 **parallel for** $u \in V_a$ **do**
4    $\boldsymbol{q}^{(0)}(u) \leftarrow \frac{d(u)}{d(u)+\Delta(u)} \boldsymbol{q}^{(0)}(u)$;
5    $\boldsymbol{r}^{(0)}(u) \leftarrow d(u)^{\frac{1}{2}} \mathbf{x}(u) - \boldsymbol{q}^{(0)}(u)$;
6 **for** $\ell$ *from* 1 *to* $L$ **do**
7    **parallel for** $u \in V_a$ **do**
8       $\boldsymbol{q}^{(\ell)}(u) \leftarrow \frac{d(u)}{d(u)+\Delta(u)} \boldsymbol{q}^{(\ell)}(u)$;
9       $\boldsymbol{r}^{(\ell)}(u) \leftarrow \boldsymbol{r}^{(\ell)}(u) + \frac{\Delta(u)}{d(u)} \boldsymbol{q}^{(\ell)}(u)$;
10      **for** *each* $v \in \{removed\ neighbors\ of\ node\ u\}$ **do**
11         $\boldsymbol{r}^{(\ell)}(u) \leftarrow \boldsymbol{r}^{(\ell)}(u) - \frac{\boldsymbol{q}^{(\ell-1)}(v)}{d(v)}$;
12 $\{\boldsymbol{q}^{(\ell)}, \boldsymbol{r}^{(\ell)}\} \leftarrow \mathsf{BasicProp}(\mathcal{G}', L, r_{\max}, \{w_\ell\}, \{\boldsymbol{q}^{(\ell)}, \boldsymbol{r}^{(\ell)}\})$;
13 **return** $\sum_{\ell=0}^{L} w_\ell \mathbf{D}^{-\frac{1}{2}} \boldsymbol{q}^{(\ell)}$

---

*where $\{\Delta \boldsymbol{r}_i^{(\ell)}\}$ represent the adjustments of residues for the $i$-th removal. For a sequence of $K$ removal requests, the total cost of initialization and $K$ removals is*

$$O\left(\frac{Ld}{r_{\max}} + Kd + \sum_{i=1}^{K} \sum_{\ell=0}^{L-1} \left\|\Delta \boldsymbol{r}_i^{(\ell)}\right\|_1 \frac{Ld}{r_{\max}}\right).$$

## E    PROOFS OF THEORETICAL RESULTS

**Notations for proofs.** For the sake of readability, we use $l(\mathbf{Z}, \mathbf{w}, i)$ to denote $l(\mathbf{e}_i^\top \mathbf{Z}\mathbf{w}, \mathbf{e}_i^\top Y)$. $\mathbf{Z}_i$ represents the embedding column vector of node $i$. When referring to the embedding row vector of $\mathbf{Z}$, we use $\mathbf{e}_i^\top \mathbf{Z}$. $\mathbf{Z}_{ij}$ denotes the term at the $i$-th row and the $j$-th column of $\mathbf{Z}$. $|\mathbf{v}|$ represents the vector with the absolute value of each element in vector $\mathbf{v}$.

### E.1    PROOF OF EQUATION (1) AND LEMMA 3.2

Equation (1) is presented as follows:

$$\mathbf{z} = \sum_{\ell=0}^{L} w_\ell \mathbf{D}^{-\frac{1}{2}} \left(\boldsymbol{q}^{(\ell)} + \sum_{t=0}^{\ell} (\mathbf{A}\mathbf{D}^{-1})^{\ell-t} \boldsymbol{r}^{(t)}\right).$$

**Lemma.** *For each feature vector $\mathbf{x}$, the reserve vectors $\{\boldsymbol{q}^{(\ell)}\}$ and the residue vectors $\{\boldsymbol{r}^{(\ell)}\}$ satisfy the following invariant property during the propagation process:*

$$\begin{cases} \boldsymbol{q}^{(\ell)}(u) + \boldsymbol{r}^{(\ell)}(u) = \mathbf{D}(u)^{\frac{1}{2}} \mathbf{x}(u), & \ell = 0 \\ \boldsymbol{q}^{(\ell)}(u) + \boldsymbol{r}^{(\ell)}(u) = \sum_{t \in \mathcal{N}(u)} \frac{\boldsymbol{q}^{(\ell-1)}(t)}{d(t)}, & 0 < \ell \le L \end{cases}$$

For simplicity, we reformulate the invariant into the vector form:

$$\begin{cases} \boldsymbol{q}^{(\ell)} + \boldsymbol{r}^{(\ell)} = \mathbf{D}^{\frac{1}{2}} \mathbf{x}, & \ell = 0 \\ \boldsymbol{q}^{(\ell)}(u) + \boldsymbol{r}^{(\ell)}(u) = \mathbf{A}\mathbf{D}^{-1} \boldsymbol{q}^{(\ell-1)}, & 0 < \ell \le L \end{cases} \tag{3}$$

We prove that Equation (1) and Lemma 3.2 hold during the propagation process by induction. At the beginning of the propagation, we initialize $\boldsymbol{q}^{(\ell)} = \mathbf{0}$ for all $\ell$, $\boldsymbol{r}^{(0)} = \mathbf{D}^{\frac{1}{2}}\mathbf{x}$ and $\boldsymbol{r}^{(\ell)} = \mathbf{0}$ for all $\ell > 1$. Therefore, Equation (3) and Equation (1) hold at the initial state. Consider the situation of level 0. Since we set $\boldsymbol{r}^{(0)}(u) = 0$ and $\boldsymbol{q}^{(0)}(u) = \boldsymbol{r}^{(0)}(u)$ once we push the residue $\boldsymbol{r}^{(\ell)}(u)$. The sum of $\boldsymbol{r}^{(0)}(u)$ and $\boldsymbol{q}^{(0)}(u)$ for any node $u$ remains unchanged during the propagation. Thus, Equation (3) holds for level 0. Consider the situation of level $\ell > 0$. Assuming that Equation (3) and Equation (1) holds after a specific push operation. Consider a push operation on node $u$ from level $k$ to level $k + 1$, $0 \leq k < L$. Let $y = \boldsymbol{r}^{(k)}(u)$, $\mathbf{e}_u$ is the one-hot vector with only 1 at the node $u$ and 0 elsewhere. According to Line 3 to 5 in Algorithm 2, the push operation can be described as follows:

$$
\begin{aligned}
\boldsymbol{r}'^{(k+1)} &= \boldsymbol{r}^{(k+1)} + \mathbf{A}\mathbf{D}^{-1} \cdot y\mathbf{e}_u \\
\boldsymbol{q}'^{(k)} &= \boldsymbol{q}^{(k)} + y\mathbf{e}_u \\
\boldsymbol{r}'^{(k)} &= \boldsymbol{r}^{(k)} - y\mathbf{e}_u
\end{aligned}
\tag{4}
$$

Therefore, we have

$$
\begin{aligned}
\boldsymbol{q}'^{(k+1)} + \boldsymbol{r}'^{(k+1)} &= \boldsymbol{q}^{(k+1)} + \boldsymbol{r}^{(k+1)} + \mathbf{A}\mathbf{D}^{-1} \cdot y\mathbf{e}_u \\
\mathbf{A}\mathbf{D}^{-1}\boldsymbol{q}'^{(k)} &= \mathbf{A}\mathbf{D}^{-1}(\boldsymbol{q}^{(k)} + y\mathbf{e}_u)
\end{aligned}
$$

Note that $\boldsymbol{q}^{(k+1)} + \boldsymbol{r}^{(k+1)} = \mathbf{A}\mathbf{D}^{-1}\boldsymbol{q}^{(k)}$ is satistied before this push operation. Thus, $\boldsymbol{q}'^{(k+1)} + \boldsymbol{r}'^{(k+1)} = \mathbf{A}\mathbf{D}^{-1}\boldsymbol{q}'^{(k)}$, and thus Equation (3) holds after this push operation. For Equation (1), the right side of the equation is updated to:

$$
\sum_{\ell}^{L} w_\ell \mathbf{D}^{-\frac{1}{2}} \left( \boldsymbol{q}'^{(\ell)} + \sum_{t=0}^{\ell} (\mathbf{A}\mathbf{D}^{-1})^{\ell-t} \boldsymbol{r}'^{(t)} \right).
$$

Plugging the updated $\boldsymbol{q}'^{(k)}, \boldsymbol{r}'^{(k)}, \boldsymbol{r}'^{(k+1)}$ into the equation, we derive that Equation (1) holds after this push operation. This completes the proof of Equation (1) and Lemma 3.2.

## E.2 Proof of Lemma 3.1

**Lemma.** *Given the threshold $r_{\max}$, the $L_2$-error between $\mathbf{z}$ and $\hat{\mathbf{z}}$ is bounded by*

$$
\|\hat{\mathbf{z}} - \mathbf{z}\|_2 \leq \sqrt{n}Lr_{\max}.
\tag{5}
$$

*Proof.* As stated in Equation (1), we have

$$
\mathbf{z} - \hat{\mathbf{z}} = \sum_{\ell=0}^{L} w_\ell \mathbf{D}^{-\frac{1}{2}} \sum_{t=0}^{\ell} (\mathbf{A}\mathbf{D}^{-1})^{\ell-t} \boldsymbol{r}^{(t)}.
$$

Since the largest eigenvalue of $\mathbf{A}\mathbf{D}^{-1}$ is no greater than 1, we have $\left\|(\mathbf{A}\mathbf{D}^{-1})^\ell\right\|_2 \leq 1$ for $\ell \geq 0$. $\mathbf{D}$ is a diagonal matrix, and $d(u)^{\frac{-1}{2}}$ is no greater than 1 for all $u$. Thus, $\left\|\mathbf{D}^{-\frac{1}{2}}\right\|_2 \leq 1$. Therefore, we have

$$
\begin{aligned}
\|\hat{\mathbf{z}} - \mathbf{z}\| &= \left\| \sum_{\ell=0}^{L} w_\ell \mathbf{D}^{-\frac{1}{2}} \sum_{t=0}^{\ell} (\mathbf{A}\mathbf{D}^{-1})^{\ell-t} \boldsymbol{r}^{(t)} \right\| \\
&\leq \sum_{\ell=0}^{L} |w_\ell| \sum_{t=0}^{\ell} \left\| \mathbf{D}^{-\frac{1}{2}} (\mathbf{A}\mathbf{D}^{-1})^{\ell-t} \boldsymbol{r}^{(t)} \right\| \\
&\leq \sum_{\ell=0}^{L} |w_\ell| \sum_{t=0}^{\ell} \left\| \mathbf{D}^{-\frac{1}{2}} \right\| \left\| (\mathbf{A}\mathbf{D}^{-1})^{\ell-t} \right\| \left\| \boldsymbol{r}^{(t)} \right\|.
\end{aligned}
$$

The norm of $\boldsymbol{r}^{(t)}$ can be derived as $\sqrt{n}r_{\max}$ since each item of $\boldsymbol{r}^{(t)}$ is no greater than $r_{\max}$. Given that $\sum_{\ell=0}^{L} |w_\ell| \leq 1$, we conclude that $\|\hat{\mathbf{z}} - \mathbf{z}\|_2 \leq \sqrt{n}Lr_{\max}$. $\qquad\square$

### E.3 PROOF OF THEOREM D.1

*Proof.* First, consider the initialization cost for generating $\hat{\mathbf{z}}_0$. Recall that we have $\boldsymbol{r}^{(0)} = \mathbf{D}^{\frac{1}{2}}\mathbf{x}$ before we invoke Algorithm 2. In Algorithm 2, we push the residue $\boldsymbol{r}^{(\ell)}(u)$ of node $u$ to its neighbors whenever $\boldsymbol{r}^{(\ell)}(u) > r_{\max}$. Each node is pushed at most once at each level. Thus, for level 0, there are at most $\left\|\mathbf{D}^{\frac{1}{2}}\mathbf{x} - \boldsymbol{r}_0^{(0)}\right\|_1 / r_{\max}$ nodes with residues greater than $r_{\max}$, where $\boldsymbol{r}_0^{(0)}$ is the residues of level 0 after the generation. The average cost for one push operation is $O(d)$, the average degree of the graph. Therefore, it costs at most $T^{(0)} = \left\|\mathbf{D}^{\frac{1}{2}}\mathbf{x} - \boldsymbol{r}_0^{(0)}\right\|_1 d/r_{\max}$ time to finish the push operations for level 0. In Algorithm 2, a total of $\boldsymbol{r}^{(\ell)}(u)$ will be added to the residues at the next level once we perform *push* on node $u$. Therefore, after the push operations for level 0, $\left\|\boldsymbol{r}^{(1)}\right\|_1$ is no greater than $\left\|\mathbf{D}^{\frac{1}{2}}\mathbf{x} - \boldsymbol{r}_0^{(0)}\right\|_1$. Thus, the cost of the push operations at level 1 satisfies that

$$T^{(1)} \leq \left(\left\|\mathbf{D}^{\frac{1}{2}}\mathbf{x} - \boldsymbol{r}_0^{(0)}\right\|_1 - \left\|\boldsymbol{r}_0^{(1)}\right\|_1\right) \cdot \frac{d}{r_{\max}}$$
$$= T^{(0)} - \left\|\boldsymbol{r}_0^{(1)}\right\|_1 \cdot \frac{d}{r_{\max}},$$

Similarly, the cost of the push operations at level $\ell$ is bounded by $T^{(\ell)} = T^{(\ell-1)} - \left\|\boldsymbol{r}_0^{(\ell)}\right\|_1 \frac{d}{r_{\max}}$. Therefore, the total cost of generating $\hat{\mathbf{z}}$ is $O\left(\left(L - \sum_{\ell=0}^{L-1}(L-\ell)\left\|\boldsymbol{r}_0^{(\ell)}\right\|_1\right) \cdot \frac{d}{r_{\max}}\right)$.

Second, consider the update cost for a single removal. For ease of analysis, we design the following update procedure: First, we update the residues of the nodes whose degrees have changed at all levels, corresponding to Algorithm 3 Line 1-11. We denote the residues as $\{r_{\pm i}^{(\ell)}\}$ after the first step. Then, we push the non-negative residues for all levels. We denote the residues as $\{r_{-i}^{(\ell)}\}$ after pushing the non-negative residues. Finally, we push the negative residues at all levels, resulting in the approximate embedding vector $\hat{\mathbf{z}}_i$ and its residues $\{r_i^{(\ell)}\}$. The cost of the first step is $O(d(u) + d(v)) = O(d)$. Denote the cost of the push operations at level $\ell$ as $T^{(\ell)}$, which contains the cost of pushing non-negative residues $T_+^{(\ell)}$ and the cost of pushing negative residues $T_-^{(\ell)}$.

Consider the cost of pushing non-negative residues. For level 0, we have

$$T_+^{(0)} = \left\|r_{-i}^{(0)} - r_{\pm i}^{(0)}\right\|_1 \cdot \frac{d}{r_{\max}} \leq \left(\left\|r_{\pm i}^{(0)}\right\|_1 - \left\|r_{-i}^{(0)}\right\|_1\right) \cdot \frac{d}{r_{\max}}.$$

Note that the residues at level 1 will be modified since the non-negative residues at level 0 are pushed. Specifically, let the modification be $\tilde{r}^{(1)}$, that is, the residues of level 1 of this time is $r_{\pm i}^{(1)} + \tilde{r}^{(1)}$. Thus, the cost of pushing non-negative residues at level 1 is

$$T_+^{(1)} = \left\|r_{-i}^{(1)} - \left(r_{\pm i}^{(1)} + \tilde{r}^{(1)}\right)\right\|_1 \cdot \frac{d}{r_{\max}}$$
$$\leq \left(\left\|r_{\pm i}^{(1)}\right\|_1 - \left\|r_{-i}^{(1)}\right\|_1\right) \cdot \frac{d}{r_{\max}} + \left\|\tilde{r}^{(1)}\right\|_1 \cdot \frac{d}{r_{\max}}.$$

Note that $\left\|\tilde{r}^{(1)}\right\|_1$ is exactly the sum of the pushed residues from level 0, which is no greater than $\left\|r_{\pm i}^{(0)}\right\|_1 - \left\|r_{-i}^{(0)}\right\|_1$. Thus, we have

$$T_+^{(1)} \leq \left(\left\|r_{\pm i}^{(1)}\right\|_1 - \left\|r_{-i}^{(1)}\right\|_1\right) \cdot \frac{d}{r_{\max}} + T_+^{(0)}.$$

Similarly, we have

$$T_+^{(\ell)} \leq \left(\left\|r_{\pm i}^{(\ell)}\right\|_1 - \left\|r_{-i}^{(\ell)}\right\|_1\right) \cdot \frac{d}{r_{\max}} + T_+^{(\ell-1)}.$$

Therefore, we derive that

$$T_+ \leq \sum_{\ell=0}^{L-1} T_+^{(\ell)} \leq \sum_{\ell=0}^{L-1}(L-\ell)\left(\left\|r_{\pm i}^{(\ell)}\right\|_1 - \left\|r_{-i}^{(\ell)}\right\|_1\right) \cdot \frac{d}{r_{\max}}. \tag{6}$$

The cost of pushing negative residues is similar. We have

$$T_-^{(0)} = \left\| r_i^{(0)} - r_{-i}^{(0)} \right\|_1 \cdot \frac{d}{r_{\max}} \le \left( \left\| r_{-i}^{(0)} \right\|_1 - \left\| r_i^{(0)} \right\|_1 \right) \cdot \frac{d}{r_{\max}}.$$

Thus $T_-$ can be derived as

$$T_- \le \sum_{\ell=0}^{L-1} T_-^{(\ell)} = \sum_{\ell=0}^{L-1} (L - \ell) \left( \left\| r_{-i}^{(\ell)} \right\|_1 - \left\| r_i^{(\ell)} \right\|_1 \right) \cdot \frac{d}{r_{\max}}. \tag{7}$$

Summing up Equation (6) and Equation (7), the $i$-th update costs can be derived as

$$T_i \le 2d + \sum_{\ell=0}^{L-1} (L - \ell) \left( \left\| r_{\pm i}^{(\ell)} \right\|_1 - \left\| r_i^{(\ell)} \right\|_1 \right) \cdot \frac{d}{r_{\max}}$$

Recall that $r_{\pm i}^{(\ell)}$ can be derived from $r_{i-1}^{(\ell)}$ by the first step. Let $r_{\pm i}^{(\ell)} = r_{i-1}^{(\ell)} + \Delta r_i^{(\ell)}$. We can conclude that

$$T_i \le O \left( d + \frac{d}{r_{\max}} \cdot \sum_{\ell=0}^{L-1} (L - \ell) \left( \left\| r_{i-1}^{(\ell)} \right\|_1 + \left\| \Delta r_i^{(\ell)} \right\|_1 - \left\| r_i^{(\ell)} \right\|_1 \right) \right).$$

Finally, consider the total cost for $K$ removals. The total cost is the sum of the initial and update costs for $K$ updates.

$$
\begin{aligned}
T =& T_{\text{init}} + \sum_{i=1}^{K} T_i \\
\le& \left( L \left\| \mathbf{D}^{\frac{1}{2}} \mathbf{x} \right\|_1 - \sum_{\ell=0}^{L-1} (L - \ell) \left\| r_0^{(\ell)} \right\|_1 \right) \cdot \frac{d}{r_{\max}} \\
&+ \sum_{i=1}^{K} d + \sum_{i=1}^{K} \frac{d}{r_{\max}} \cdot \sum_{\ell=0}^{L-1} (L - \ell) \left( \left\| r_{i-1}^{(\ell)} \right\|_1 + \left\| \Delta r_i^{(\ell)} \right\|_1 - \left\| r_i^{(\ell)} \right\|_1 \right) \\
\le& L \left\| \mathbf{D}^{\frac{1}{2}} \mathbf{x} \right\|_1 \frac{d}{r_{\max}} + Kd \\
&+ \sum_{i=1}^{K} \frac{d}{r_{\max}} \cdot \sum_{\ell=0}^{L-1} (L - \ell) \left\| \Delta r_i^{(\ell)} \right\|_1 - \sum_{\ell=0}^{L-1} (L - \ell) \left\| r_i^{(\ell)} \right\|_1 \\
\overset{(a)}{\le}& L \left\| \mathbf{D}^{\frac{1}{2}} \mathbf{x} \right\|_1 \frac{d}{r_{\max}} + Kd + \sum_{i=1}^{K} L \sum_{\ell=0}^{L-1} \left\| \Delta r_i^{(\ell)} \right\|_1 \frac{d}{r_{\max}},
\end{aligned}
$$

where we omit the $- \sum_{\ell=0}^{L-1} (L - \ell) \left\| r_K^{(\ell)} \right\|_1$ in (a). This completes the proof. $\qquad \square$

### E.4 Proof of Theorem 3.3

**Theorem.** *For a sequence of $m$ removal requests that remove all edges of the graph, the amortized cost per edge removal is $O\left(L^2 d\right)$. For a sequence of $K$ random edge removals, the expected cost per edge removal is $O\left(L^2 d\right)$.*

With the help of Theorem D.1, we derive the proof of Theorem 3.3 as follows.

*Proof.* Consider the amortized cost for $m$ removal requests. First, we observe that $\sum_{i=1}^{m} \left\| \Delta r_i^{(0)} \right\|_1 = \left\| \mathbf{D}_0^{\frac{1}{2}} \mathbf{x} \right\|_1$, where we denote the degree matrix of the initial graph as $\mathbf{D}_0$. To explain, $\left\| \Delta r^{(0)} \right\|_1 = ((d(u) + 1)^{\frac{1}{2}} - d(u)^{\frac{1}{2}}) |\mathbf{x}(u)| + ((d(v) + 1)^{\frac{1}{2}} - d(v)^{\frac{1}{2}}) |\mathbf{x}(v)|$ if we remove edge $(u, v)$, where $d(u)$ is the new degree of node $u$. All nodes have no edges at the end of the $m$ removals. Thus,

the sum of $\left\|\Delta \boldsymbol{r}_m^{(0)}\right\|_1$ for the $m$ removals is $\left\|\mathbf{D}_0^{\frac{1}{2}}\mathbf{x}\right\|_1$. And we observe that $\left\|\Delta \boldsymbol{r}^{(\ell)}\right\|_1$ for $\ell > 0$ is $O\left(\left\|\Delta \boldsymbol{r}^{(0)}\right\|_1\right)$ for each removal. First, we have $\left\|\Delta \boldsymbol{r}^{(\ell)}\right\|_1 = \frac{2|\boldsymbol{q}^{(\ell-1)}(u)|}{d(u)+1} + \frac{2|\boldsymbol{q}^{(\ell-1)}(v)|}{d(v)+1}$. Note that $\sum_{t=1}^L \left|\boldsymbol{q}^{(t-1)}(u)\right| \leq 1$ and $|\mathbf{x}(u)| \leq 1$ since $\mathbf{x}$ is normalized such that $\left\|\mathbf{D}^{\frac{1}{2}}\mathbf{x}\right\|_1 \leq 1$. Thus, $\left\|\Delta \boldsymbol{r}^{(\ell)}\right\|_1 \leq \frac{2}{d(u)+1} + \frac{2}{d(v)+1}$. On the other hand, since $(d(u)+1)^{\frac{1}{2}} - d(u)^{\frac{1}{2}} \leq \frac{1}{2\sqrt{d(u)}}$, we have $\left\|\Delta \boldsymbol{r}^{(0)}\right\|_1 \leq \frac{1}{2\sqrt{d(u)}} + \frac{1}{2\sqrt{d(v)}}$ due to $|\mathbf{x}(u)| \leq 1$ for any node $u$. Therefore, $\left\|\Delta \boldsymbol{r}^{(\ell)}\right\|_1 \leq 4\left\|\Delta \boldsymbol{r}^{(0)}\right\|_1$ for any removal and any $\ell > 0$. Thus, the sum $\sum_{\ell=0}^{L-1} \left\|\Delta \boldsymbol{r}^{(\ell)}\right\|_1 = O\left(L\left\|\Delta \boldsymbol{r}^{(0)}\right\|_1\right)$. Plugging the result into the total cost in Theorem D.1, the total cost for $m$ removals is

$$T = L\left\|\mathbf{D}^{\frac{1}{2}}\mathbf{x}\right\|_1 \frac{d}{r_{\max}} + md + \sum_{i=1}^m L \sum_{\ell=0}^{L-1} \left\|\Delta \boldsymbol{r}_i^{(\ell)}\right\|_1 \frac{d}{r_{\max}}$$

$$\leq L\left\|\mathbf{D}^{\frac{1}{2}}\mathbf{x}\right\|_1 \frac{d}{r_{\max}} + md + O\left(\sum_{i=1}^m L^2 \left\|\Delta \boldsymbol{r}_i^{(0)}\right\|_1 \frac{d}{r_{\max}}\right)$$

$$\leq L\left\|\mathbf{D}^{\frac{1}{2}}\mathbf{x}\right\|_1 \frac{d}{r_{\max}} + md + O\left(L^2 \left\|\mathbf{D}^{\frac{1}{2}}\mathbf{x}\right\|_1 \frac{d}{r_{\max}}\right)$$

We set $r_{\max} = 1/\sqrt{n_t n}$ by the certified unlearning requirement, where $n_t$ represents the size of the training set. Thus, $r_{\max} \geq 1/n$. The amortized cost per edge removal is

$$\frac{T}{m} = O\left(\frac{1}{m} \cdot L^2 \left\|\mathbf{D}^{\frac{1}{2}}\mathbf{x}\right\|_1 dn\right).$$

Since we have $m \geq n$ and $\left\|\mathbf{D}^{\frac{1}{2}}\mathbf{x}\right\|_1 \leq 1$, we conclude that the amortized cost is $O(L^2 d)$.

Consider the expected cost for $K$ random edge removals. Each node has the same probability of being the endpoint of a removed edge. When removing $e = (u, v)$, the part induced by node $u$ in $\sum_{\ell=0}^{L-1} \left\|\Delta \boldsymbol{r}^{(\ell)}\right\|_1$ can be expressed as

$$S_u = \left((d(u)+1)^{\frac{1}{2}} - d(u)^{\frac{1}{2}}\right) |\mathbf{x}(u)|$$
$$+ \sum_{t=1}^L \frac{|\boldsymbol{q}^{(t-1)}(u)|}{d(u)+1} + \sum_{w \in \mathcal{N}(u)} \frac{|\boldsymbol{q}^{(t-1)}(u)|}{d(u)(d(u)+1)}.$$

The first term $\left((d(u)+1)^{\frac{1}{2}} - d(u)^{\frac{1}{2}}\right) |\mathbf{x}(u)|$ is less than $|\mathbf{x}(u)|$ and the second term equals $\frac{2|\boldsymbol{q}^{(t-1)}(u)|}{d(u)+1}$. Therefore, the expected value of $S_u$ is

$$\mathrm{E}[S_u] = \frac{1}{n} \sum_{u \in V} S_u = \frac{1}{n} \sum_{u \in V} |\mathbf{x}(u)| + \frac{2}{n} \sum_{u \in V} \sum_{t=1}^L \frac{|\boldsymbol{q}^{(t-1)}(u)|}{d(u)+1}.$$

The first term is $O(\frac{1}{n})$ and $\sum_{u \in V} \left|\boldsymbol{q}^{(t-1)}(u)\right|$ in second term is less than 1 for all level $t$ both due to the fact that $\left\|\mathbf{D}^{\frac{1}{2}}\mathbf{x}\right\|_1 \leq 1$. Thus, the second term is less than $\frac{2L}{n}$. The expected value of $S_u$ is $O(\frac{1}{n})$. Therefore, for one edge removal, the expected value of $\sum_{\ell=0}^{L-1} \left\|\Delta \boldsymbol{r}^{(\ell)}\right\|_1$ is $O(\frac{1}{n})$. Note that we set $r_{\max} = O(1/\sqrt{n_t n}) \geq 1/n$. Plugging the result into the total cost in Theorem D.1, the expected cost per edge removal is $O\left(L^2 d\right)$. □

### E.5 PROOF OF THEOREM 4.2

**Theorem.** *Suppose that Assumption 4.1 holds and the feature of node $u$ is to be unlearned. If $\forall j \in [F]$, $\left\|\hat{\mathbf{Z}}\mathbf{e}_j - \mathbf{Z}\mathbf{e}_j\right\| \leq \epsilon_1$, we have $\|\nabla \mathcal{L}(\mathbf{w}^-, \mathcal{D}')\|$ is less than*

$$\left(\frac{c\gamma_1}{\lambda} F + c_1\sqrt{F(n_t-1)}\right)\left(\epsilon_1 + \frac{8\gamma_1 F}{\lambda(n_t-1)} \cdot \sqrt{d(u)}\right).$$

For the sake of simplicity, the upper bound of $\|\mathbf{Z}\mathbf{e}_j - \mathbf{Z}'\mathbf{e}_j\|$ for all $j \in [F]$ is denoted as $\epsilon_2$ in the following analysis. At the end of the proof, we will plug the bound in Lemma E.1 into the result. We also assume that node $u$ is the $n_t$-th node in the training. We state the following lemmas to support the proof.

**Lemma E.1.** *Suppose that the feature of node $u$ is to be removed. Then, we have $\|\mathbf{Z}\mathbf{e}_j - \mathbf{Z}'\mathbf{e}_j\| \leq \sqrt{d(u)}$ for all $j \in [F]$.*

**Lemma E.2.** $\forall j \in [F]$, *suppose that $\left\|\hat{\mathbf{Z}}\mathbf{e}_j - \mathbf{Z}\mathbf{e}_j\right\| \leq \epsilon_1$. We have $\left\|\sum_{i \in [n_t-1]} \left(\hat{\mathbf{Z}}_i - \mathbf{Z}_i\right)\right\| \leq \sqrt{F(n_t-1)}\epsilon_1$.*

**Lemma E.3.** $\forall j \in [F]$, *suppose that $\|\mathbf{Z}'\mathbf{e}_j - \mathbf{Z}\mathbf{e}_j\| \leq \epsilon_2$. We have $\left\|\sum_{i \in [n_t-1]} \left(\mathbf{Z}'_i - \mathbf{Z}_i\right)\right\| \leq \sqrt{F(n_t-1)}\epsilon_2$.*

**Lemma E.4.** *For the embedding vector $\mathbf{z} = \sum_{\ell=0}^{L} w_\ell \left(\mathbf{D}^{-\frac{1}{2}}\mathbf{A}\mathbf{D}^{-\frac{1}{2}}\right)^{\ell} \mathbf{x}$, where $\|\mathbf{x}\| \leq 1$ and $\sum_{\ell=0}^{L} w_\ell \leq 1$, we have $\|\mathbf{z}\| \leq 1$.*

*Proof.* We note that $\|\nabla \mathcal{L}(\mathbf{w}^-, \mathcal{D}')\|$ can be written as

$$\left\|\nabla \mathcal{L}(\mathbf{w}^-, \mathcal{D}') + \nabla \mathcal{L}(\mathbf{w}^-, \hat{\mathcal{D}}') - \nabla \mathcal{L}(\mathbf{w}^-, \hat{\mathcal{D}}')\right\|,$$

which is less than $\left\|\nabla \mathcal{L}(\mathbf{w}^-, \mathcal{D}') - \nabla \mathcal{L}(\mathbf{w}^-, \hat{\mathcal{D}}')\right\| + \left\|\nabla \mathcal{L}(\mathbf{w}^-, \hat{\mathcal{D}}')\right\|$ due to the Minkowski inequality.

We start with the term $\left\|\nabla \mathcal{L}(\mathbf{w}^-, \mathcal{D}') - \nabla \mathcal{L}(\mathbf{w}^-, \hat{\mathcal{D}}')\right\|$. Observe that

$$\left\|\nabla \mathcal{L}(\mathbf{w}^-, \mathcal{D}') - \nabla \mathcal{L}(\mathbf{w}^-, \hat{\mathcal{D}}')\right\|$$

$$= \left\|\sum_{i \in [n_t-1]} \left(\nabla l(\mathbf{Z}', \mathbf{w}^-, i) - \nabla l(\hat{\mathbf{Z}}', \mathbf{w}^-, i)\right)\right\|$$

$$= \left\|\sum_{i \in [n_t-1]} \left(l'(\mathbf{Z}', \mathbf{w}^-, i)\mathbf{Z}'_i - l'(\hat{\mathbf{Z}}', \mathbf{w}^-, i)\hat{\mathbf{Z}}'_i\right)\right\|$$

$$\overset{(c)}{\leq} \left(\left\|\sum_{i \in [n_t-1]} \left(l'(\mathbf{Z}', \mathbf{w}^-, i)\mathbf{Z}'_i - l'(\hat{\mathbf{Z}}', \mathbf{w}^-, i)\mathbf{Z}'_i\right)\right\|\right. \tag{8}$$

$$\left. + \left\|\sum_{i \in [n_t-1]} \left(l'(\hat{\mathbf{Z}}', \mathbf{w}^-, i)\mathbf{Z}'_i - l'(\hat{\mathbf{Z}}', \mathbf{w}^-, i)\hat{\mathbf{Z}}'_i\right)\right\|\right). \tag{9}$$

where the inequality (c) is due to the Minkowski inequality. Consider Term (8), we have

$$\left\|\sum_{i \in [n_t-1]} \left(l'(\mathbf{Z}', \mathbf{w}^-, i)\mathbf{Z}'_i - l'(\hat{\mathbf{Z}}', \mathbf{w}^-, i)\mathbf{Z}'_i\right)\right\|$$

$$= \left\|\sum_{i \in [n_t-1]} \left(l'(\mathbf{Z}', \mathbf{w}^-, i) - l'(\hat{\mathbf{Z}}', \mathbf{w}^-, i)\right)\mathbf{Z}'_i\right\|$$

$$= \sqrt{\sum_{j \in [F]} \left(\sum_{i \in [n_t-1]} \left(l'(\mathbf{Z}', \mathbf{w}^-, i) - l'(\hat{\mathbf{Z}}', \mathbf{w}^-, i)\right)\mathbf{Z}'_{ij}\right)^2}$$

$$\overset{(a)}{\leq} \sqrt{\sum_{j \in [F]}\sum_{i \in [n_t-1]} (l'(\mathbf{Z}', \mathbf{w}^-, i) - l'(\hat{\mathbf{Z}}', \mathbf{w}^-, i))^2 \sum_{i \in [n_t-1]} \mathbf{Z}'^2_{ij}}, \tag{10}$$

where $(a)$ is due to the Cauchy-Schwarz inequality. Recalling that $l'$ is $\gamma_1$-Lipschitz as stated in the Assumption 4.1, we derive that

$$\sum_{i \in [n_t-1]} (l'(\mathbf{Z}', \mathbf{w}^-, i) - l'(\hat{\mathbf{Z}}', \mathbf{w}^-, i))^2$$

$$\leq \sum_{i \in [n_t-1]} \gamma_1^2 \left\| \mathbf{e}_i^\top \mathbf{Z}' \mathbf{w}^- - \mathbf{e}_i^\top \hat{\mathbf{Z}}' \mathbf{w}^- \right\|^2$$

$$\leq \sum_{i \in [n_t-1]} \gamma_1^2 \left\| \mathbf{e}_i^\top \mathbf{Z}' - \mathbf{e}_i^\top \hat{\mathbf{Z}}' \right\|^2 \left\| \mathbf{w}^- \right\|^2$$

$$\leq \left\| \mathbf{w}^- \right\|^2 \sum_{i \in [n_t-1]} \gamma_1^2 \left\| \mathbf{e}_i^\top \mathbf{Z}' - \mathbf{e}_i^\top \hat{\mathbf{Z}}' \right\|^2$$

$$\overset{(a)}{\leq} \gamma_1^2 F \epsilon_1^2 \left\| \mathbf{w}^- \right\|^2. \tag{11}$$

For inequality $(a)$, note that

$$\sum_{i \in [n_t-1]} \left\| \mathbf{e}_i^\top \mathbf{Z}' - \mathbf{e}_i^\top \hat{\mathbf{Z}}' \right\|^2 \leq \sum_{j \in [F]} \left\| \mathbf{Z}' \mathbf{e}_j - \hat{\mathbf{Z}}' \mathbf{e}_j \right\|^2 \leq F \epsilon_1^2,$$

which follows from the assumption that $\left\| \mathbf{Z}' \mathbf{e}_j - \hat{\mathbf{Z}}' \mathbf{e}_j \right\| \leq \epsilon_1$ for all $j$. Since $\mathbf{w}^- = \mathbf{w}^\star + \hat{\mathbf{H}}_{\mathbf{w}^\star}^{-1} \Delta$, we have

$$\left\| \mathbf{w}^- \right\| = \left\| \mathbf{w}^\star + \hat{\mathbf{H}}_{\mathbf{w}^\star}^{-1} \Delta \right\| \leq \left\| \mathbf{w}^\star \right\| + \left\| \hat{\mathbf{H}}_{\mathbf{w}^\star}^{-1} \Delta \right\|. \tag{12}$$

Note that $\left\| \nabla \ell(\mathbf{e}_i^\top \mathbf{Z} \mathbf{w}, \mathbf{e}_i^\top Y) \right\| \leq c$ holds as stated in the Assumption 4.1. Thus,

$$\left\| \mathbf{w}^\star \right\| = \frac{\left\| \sum_{i \in [n_t]} \nabla \ell(\mathbf{e}_i^\top \hat{\mathbf{Z}} \mathbf{w}^-, \mathbf{e}_i^\top \mathbf{Y}) \right\|}{\lambda n_t} \leq \frac{c}{\lambda}. \tag{13}$$

Also, following from Lemma E.4, we have $\sum_{j \in [F]} \sum_{i \in [n_t-1]} \mathbf{Z}_{ij}'^2 \leq F$. Plugging Inequality (11), Inequality (12) and Inequality (13) into Inequality (10), we have

$$\left\| \sum_{i \in [n_t-1]} \left( l'(\mathbf{Z}', \mathbf{w}^-, i) \mathbf{Z}_i' - l'(\hat{\mathbf{Z}}', \mathbf{w}^-, i) \mathbf{Z}_i' \right) \right\| \leq \gamma_1 F \epsilon_1 \left( \frac{c}{\lambda} + \left\| \hat{\mathbf{H}}_{\mathbf{w}^\star}^{-1} \Delta \right\| \right).$$

Next, considering Term (9), we have

$$\left\| \sum_{i \in [n_t-1]} \left( l'(\hat{\mathbf{Z}}', \mathbf{w}^-, i) \mathbf{Z}_i' - l'(\hat{\mathbf{Z}}', \mathbf{w}^-, i) \hat{\mathbf{Z}}_i' \right) \right\|$$

$$= \left\| \sum_{i \in [n_t-1]} l'(\hat{\mathbf{Z}}', \mathbf{w}^-, i) \left( \mathbf{Z}_i' - \hat{\mathbf{Z}}_i' \right) \right\|$$

$$\overset{(a)}{\leq} c_1 \left\| \sum_{i \in [n_t-1]} \left( \mathbf{Z}_i' - \hat{\mathbf{Z}}_i' \right) \right\|$$

$$\overset{(b)}{\leq} c_1 \sqrt{F(n_t - 1)} \epsilon_1,$$

where Inequality $(a)$ is due to the assumption that $\ell' \leq c_1$ and $(b)$ follows from Lemma E.2 by substituting $\mathcal{D}$ with $\mathcal{D}'$. Therefore, we have

$$\left\| \nabla \mathcal{L}(\mathbf{w}^-, \mathcal{D}') - \nabla \mathcal{L}(\mathbf{w}^-, \hat{\mathcal{D}}') \right\|$$

$$\leq \gamma_1 \epsilon_1 F \left( \frac{c}{\lambda} + \left\| \hat{\mathbf{H}}_{\mathbf{w}^\star}^{-1} \Delta \right\| \right) + c_1 \sqrt{F(n_t - 1)} \epsilon_1. \tag{14}$$

Next, our focus shifts to analyzing $\left\|\nabla\mathcal{L}(\mathbf{w}^-, \hat{\mathcal{D}}')\right\|$. While our approach draws partial inspiration from the proof of Theorem 1 in (Chien et al., 2022a), it significantly diverges due to the approximation involved in the embedding matrix. This necessitates a distinct analytical framework. Let $\hat{G}(\mathbf{w}) = \nabla\mathcal{L}(\mathbf{w}, \hat{\mathcal{D}}')$. Note that $\hat{G} : \mathbb{R}^d \to \mathbb{R}^d$ is a vector-valued function. By Taylor's Theorem, there exists some $\eta \in [0, 1]$ such that:

$$
\begin{aligned}
\hat{G}\left(\mathbf{w}^-\right) = \hat{G}\left(\mathbf{w}^\star + \hat{\mathbf{H}}_{\mathbf{w}^\star}^{-1}\Delta\right) &= \hat{G}\left(\mathbf{w}^\star\right) + \nabla\hat{G}\left(\mathbf{w}^\star + \eta\hat{\mathbf{H}}_{\mathbf{w}^\star}^{-1}\Delta\right)\hat{\mathbf{H}}_{\mathbf{w}^\star}^{-1}\Delta \\
&= \hat{G}(\mathbf{w}^\star) + \hat{\mathbf{H}}_{\mathbf{w}_\eta}\hat{\mathbf{H}}_{\mathbf{w}^\star}^{-1}\Delta \\
&= \hat{G}(\mathbf{w}^\star) + \Delta + \hat{\mathbf{H}}_{\mathbf{w}_\eta}\hat{\mathbf{H}}_{\mathbf{w}^\star}^{-1}\Delta - \Delta \\
&\overset{(a)}{=} \hat{\mathbf{H}}_{\mathbf{w}_\eta}\hat{\mathbf{H}}_{\mathbf{w}^\star}^{-1}\Delta - \Delta \\
&= (\hat{\mathbf{H}}_{\mathbf{w}_\eta} - \hat{\mathbf{H}}_{\mathbf{w}^\star})\hat{\mathbf{H}}_{\mathbf{w}^\star}^{-1}\Delta.
\end{aligned}
\tag{15}
$$

Here, we denote $\hat{\mathbf{H}}_{\mathbf{w}_\eta} = \nabla\hat{G}\left(\mathbf{w}^\star + \eta\hat{\mathbf{H}}_{\mathbf{w}^\star}^{-1}\Delta\right)$ with $\mathbf{w}_\eta = \mathbf{w}^\star + \eta\hat{\mathbf{H}}_{\mathbf{w}^\star}^{-1}\Delta$. Equality (a) is due to $\Delta = \nabla\mathcal{L}(\mathbf{w}^\star, \hat{\mathcal{D}}) - \nabla\mathcal{L}(\mathbf{w}^\star, \hat{\mathcal{D}}')$ and $\mathbf{w}^\star$ is the unique minimizer of $\mathcal{L}(\cdot, \hat{\mathcal{D}})$. This gives:

$$
\left\|\hat{G}\left(\mathbf{w}^-\right)\right\| = \left\|(\hat{\mathbf{H}}_{\mathbf{w}_\eta} - \hat{\mathbf{H}}_{\mathbf{w}^\star})\hat{\mathbf{H}}_{\mathbf{w}^\star}^{-1}\Delta\right\| \leq \left\|(\hat{\mathbf{H}}_{\mathbf{w}_\eta} - \hat{\mathbf{H}}_{\mathbf{w}^\star})\right\|\left\|\hat{\mathbf{H}}_{\mathbf{w}^\star}^{-1}\Delta\right\|.
$$

First, we consider the term $\left\|(\hat{\mathbf{H}}_{\mathbf{w}_\eta} - \hat{\mathbf{H}}_{\mathbf{w}^\star})\right\|$. Let $\tilde{\mathbf{Z}}$ consist of the first $n_t - 1$ rows of $\hat{\mathbf{Z}}'$, $\mathbf{D}_1 = \mathrm{diag}(l''(\hat{\mathbf{Z}}', \mathbf{w}_\eta, 1), \cdots, l''(\hat{\mathbf{Z}}', \mathbf{w}_\eta, n_t - 1))$ and $\mathbf{D}_2 = \mathrm{diag}(l''(\hat{\mathbf{Z}}', \mathbf{w}^\star, 1), \cdots, l''(\hat{\mathbf{Z}}', \mathbf{w}^\star, n_t - 1))$. By the definition, we have

$$
\begin{aligned}
&\left\|\hat{\mathbf{H}}_{\mathbf{w}_\eta} - \hat{\mathbf{H}}_{\mathbf{w}^\star}\right\| \\
&= \left\|\nabla^2\mathcal{L}(\mathbf{w}_\eta, \hat{\mathcal{D}}') - \nabla^2\mathcal{L}(\mathbf{w}^\star, \hat{\mathcal{D}}')\right\| \\
&= \left\|\tilde{\mathbf{Z}}^\top\mathbf{D}_1\tilde{\mathbf{Z}} - \tilde{\mathbf{Z}}^\top\mathbf{D}_2\tilde{\mathbf{Z}}\right\| \\
&\leq \left\|\tilde{\mathbf{Z}}^\top\right\|\|\mathbf{D}_1 - \mathbf{D}_2\|\left\|\tilde{\mathbf{Z}}\right\|.
\end{aligned}
$$

Note that $\left\|\tilde{\mathbf{Z}}^\top\right\|\left\|\tilde{\mathbf{Z}}\right\| = \left\|\tilde{\mathbf{Z}}^\top\tilde{\mathbf{Z}}\right\|$, and the trace of $\tilde{\mathbf{Z}}^\top\tilde{\mathbf{Z}}$ has the following upper bound:

$$
\mathbf{tr}(\tilde{\mathbf{Z}}^\top\tilde{\mathbf{Z}}) = \sum_{j\in[F]}\left\|\tilde{\mathbf{Z}}\mathbf{e}_j\right\|^2 \leq \sum_{j\in[F]}\left(\|\mathbf{Z}\mathbf{e}_j\| + \left\|\hat{\mathbf{Z}}\mathbf{e}_j - \mathbf{Z}\mathbf{e}_j\right\|\right)^2 \leq (1 + \epsilon_1)^2 F.
$$

Thus, we have $\left\|\tilde{\mathbf{Z}}^\top\tilde{\mathbf{Z}}\right\| \leq F(1 + \epsilon_1)^2$. Since $\mathbf{D}_1 - \mathbf{D}_2$ is a diagonal matrix, its norm equals the maximum absolute value of its diagonal elements. Due to the assumption that $l'$ is $\gamma_1$-Lipschitz, we have $l''(\mathbf{Z}, \mathbf{w}, i) \leq \gamma_1$ for any $\mathbf{Z}, \mathbf{w}, i$. Thus, $\|\mathbf{D}_1 - \mathbf{D}_2\|$ is upper bounded by $\gamma_1$. Therefore, we have

$$
\left\|\hat{\mathbf{H}}_{\mathbf{w}_\eta} - \hat{\mathbf{H}}_{\mathbf{w}^\star}\right\| \leq \gamma_1(1 + \epsilon_1)^2 F.
\tag{16}
$$

Since $\mathcal{L}(\cdot, \hat{\mathcal{D}}')$ is $\lambda(n_t - 1)$-strongly convex, we have $\left\|\hat{\mathbf{H}}_{\mathbf{w}^\star}\right\| \geq \lambda(n_t - 1)$, hence $\left\|\hat{\mathbf{H}}_{\mathbf{w}^\star}^{-1}\right\| \leq \frac{1}{\lambda(n_t - 1)}$. Next, we focus on bounding $\Delta = \nabla\mathcal{L}(\mathbf{w}^\star, \hat{\mathcal{D}}) - \nabla\mathcal{L}(\mathbf{w}^\star, \hat{\mathcal{D}}')$. In the feature unlearning scenario, we have

$$
\Delta = \lambda\mathbf{w}^\star + \nabla l(\hat{\mathbf{Z}}, \mathbf{w}^\star, n_t) + \sum_{i\in[n_t-1]}\left(\nabla l(\hat{\mathbf{Z}}, \mathbf{w}^\star, i) - \nabla l(\hat{\mathbf{Z}}', \mathbf{w}^\star, i)\right).
$$

By assumption that $\|\nabla l(\mathbf{Z}, \mathbf{w}, i)\| \leq c$ and the fact that $\|\mathbf{w}^\star\| \leq \frac{c}{\lambda}$, we have $\left\|\lambda \mathbf{w}^\star + \nabla l(\hat{\mathbf{Z}}, \mathbf{w}^\star, n_t)\right\| \leq 2c$. For the last term, we derive that

$$
\left\| \sum_{i \in [n_t-1]} \nabla l(\hat{\mathbf{Z}}, \mathbf{w}^\star, i) - \sum_{i \in [n_t-1]} \nabla l(\hat{\mathbf{Z}}', \mathbf{w}^\star, i) \right\|
$$

$$
= \left\| \sum_{i \in [n_t-1]} \left( l'(\hat{\mathbf{Z}}, \mathbf{w}^\star, i)\hat{\mathbf{Z}}_i - l'(\hat{\mathbf{Z}}', \mathbf{w}^\star, i)\hat{\mathbf{Z}}'_i \right) \right\|
$$

$$
\leq \left\| \sum_{i \in [n_t-1]} \left( l'(\hat{\mathbf{Z}}, \mathbf{w}^\star, i)\hat{\mathbf{Z}}_i - l'(\hat{\mathbf{Z}}', \mathbf{w}^\star, i)\hat{\mathbf{Z}}_i \right) \right\| \tag{17}
$$

$$
+ \left\| \sum_{i \in [n_t-1]} \left( l'(\hat{\mathbf{Z}}', \mathbf{w}^\star, i)\hat{\mathbf{Z}}_i - l'(\hat{\mathbf{Z}}', \mathbf{w}^\star, i)\hat{\mathbf{Z}}'_i \right) \right\|. \tag{18}
$$

For Term (17), it can be bounded by

$$
\left\| \sum_{i \in [n_t-1]} \left( l'(\hat{\mathbf{Z}}, \mathbf{w}^\star, i)\hat{\mathbf{Z}}_i - l'(\hat{\mathbf{Z}}', \mathbf{w}^\star, i)\hat{\mathbf{Z}}_i \right) \right\|
$$

$$
\leq \left\| \sum_{i \in [n_t-1]} \left( l'(\hat{\mathbf{Z}}, \mathbf{w}^\star, i) - l'(\mathbf{Z}, \mathbf{w}^\star, i) \right) \hat{\mathbf{Z}}_i \right\|
$$

$$
+ \left\| \sum_{i \in [n_t-1]} \left( l'(\mathbf{Z}, \mathbf{w}^\star, i) - l'(\mathbf{Z}', \mathbf{w}^\star, i) \right) \hat{\mathbf{Z}}_i \right\|
$$

$$
+ \left\| \sum_{i \in [n_t-1]} \left( l'(\mathbf{Z}', \mathbf{w}^\star, i) - l'(\hat{\mathbf{Z}}', \mathbf{w}^\star, i) \right) \hat{\mathbf{Z}}_i \right\|.
$$

For the first term, similar to Term (8), we have

$$
\left\| \sum_{i \in [n_t-1]} \left( l'(\hat{\mathbf{Z}}, \mathbf{w}^\star, i) - l'(\mathbf{Z}, \mathbf{w}^\star, i) \right) \hat{\mathbf{Z}}_i \right\|
$$

$$
\leq \sqrt{ \sum_{j \in [F]} \sum_{i \in [n_t-1]} (l'(\hat{\mathbf{Z}}, \mathbf{w}^\star, i) - l'(\mathbf{Z}, \mathbf{w}^\star, i))^2 \sum_{i \in [n_t-1]} \hat{\mathbf{Z}}_{ij}^2 },
$$

and

$$
\sum_{i \in [n_t-1]} (l'(\hat{\mathbf{Z}}, \mathbf{w}^\star, i) - l'(\mathbf{Z}, \mathbf{w}^\star, i))^2
$$

$$
\leq \sum_{i \in [n_t-1]} \gamma_1^2 \left\| \mathbf{e}_i^\top \hat{\mathbf{Z}} \mathbf{w}^\star - \mathbf{e}_i^\top \mathbf{Z} \mathbf{w}^\star \right\|^2
$$

$$
\leq \sum_{i \in [n_t-1]} \frac{c^2 \gamma_1^2}{\lambda^2} \left\| \mathbf{e}_i^\top \hat{\mathbf{Z}} - \mathbf{e}_i^\top \mathbf{Z} \right\|^2
$$

$$
\leq \frac{c^2 \gamma_1^2}{\lambda^2} F \epsilon_1^2.
$$

To derive the Frobenius norm of $\hat{\mathbf{Z}}$, $\sqrt{\sum_{j \in [F]} \sum_{i \in [n_t-1]} \hat{\mathbf{Z}}_{ij}^2}$, note that

$$
\sum_{i \in [n_t-1]} \hat{\mathbf{Z}}_{ij}^2 = \left\| \hat{\mathbf{Z}} \mathbf{e}_j \right\|^2 \leq \left( \|\mathbf{Z} \mathbf{e}_j\| + \left\| \hat{\mathbf{Z}} \mathbf{e}_j - \mathbf{Z} \mathbf{e}_j \right\| \right)^2 = (1 + \epsilon_1)^2,
$$

which implies that $\sqrt{\sum_{j\in[F]}\sum_{i\in[n_t-1]}\hat{\mathbf{Z}}_{ij}^2} \leq \sqrt{F}(1+\epsilon_1)$. Thus, we have the first term bounded as

$$\left\|\sum_{i\in[n_t-1]}\left(l'(\hat{\mathbf{Z}},\mathbf{w}^\star,i)-l'(\mathbf{Z},\mathbf{w}^\star,i)\right)\hat{\mathbf{Z}}_i\right\| \leq \frac{c\gamma_1}{\lambda}F\epsilon_1(1+\epsilon_1).$$

The third term is the same as the first term, except that the embedding matrix is replaced by the updated one. Thus, we have

$$\left\|\sum_{i\in[n_t-1]}\left(l'(\mathbf{Z}',\mathbf{w}^\star,i)-l'(\hat{\mathbf{Z}}',\mathbf{w}^\star,i)\right)\hat{\mathbf{Z}}_i\right\| \leq \frac{c\gamma_1}{\lambda}F\epsilon_1(1+\epsilon_1).$$

The second term can be derived similarly,

$$\sum_{i\in[n_t-1]}(l'(\mathbf{Z},\mathbf{w}^\star,i)-l'(\mathbf{Z}',\mathbf{w}^\star,i))^2$$

$$\leq \sum_{i\in[n_t-1]}\frac{c^2\gamma_1^2}{\lambda^2}\left\|\mathbf{e}_i^\top\mathbf{Z}-\mathbf{e}_i^\top\mathbf{Z}'\right\|^2.$$

According to Lemma E.3, we derive that

$$\sum_{i\in[n_t-1]}(l'(\mathbf{Z},\mathbf{w}^\star,i)-l'(\mathbf{Z}',\mathbf{w}^\star,i))^2 \leq \frac{c^2\gamma_1^2}{\lambda^2}F\epsilon_2^2.$$

Therefore, the second term can be bounded as

$$\left\|\sum_{i\in[n_t-1]}(l'(\mathbf{Z},\mathbf{w}^\star,i)-l'(\mathbf{Z}',\mathbf{w}^\star,i))\hat{\mathbf{Z}}_i\right\| \leq \frac{c\gamma_1}{\lambda}F\epsilon_2(1+\epsilon_1).$$

Plugging the above results into Term (17), we have

$$\left\|\sum_{i\in[n_t-1]}\nabla l(\hat{\mathbf{Z}},\mathbf{w}^\star,i)-\sum_{i\in[n_t-1]}\nabla l(\hat{\mathbf{Z}}',\mathbf{w}^\star,i)\right\|$$

$$\leq\frac{c\gamma_1}{\lambda}F(1+\epsilon_1)(2\epsilon_1+\epsilon_2).$$

For Term (18), we have

$$\left\|\sum_{i\in[n_t-1]}\left(\ell'(\mathbf{e}_i^\top\hat{\mathbf{Z}}'\mathbf{w}^\star,\mathbf{e}_i^\top\mathbf{Y})\hat{\mathbf{Z}}_i-\ell'(\mathbf{e}_i^\top\hat{\mathbf{Z}}'\mathbf{w}^\star,\mathbf{e}_i^\top\mathbf{Y})\hat{\mathbf{Z}}_i'\right)\right\|$$

$$=\left\|\sum_{i\in[n_t-1]}\ell'(\mathbf{e}_i^\top\hat{\mathbf{Z}}'\mathbf{w}^\star,\mathbf{e}_i^\top\mathbf{Y})(\hat{\mathbf{Z}}_i-\hat{\mathbf{Z}}_i')\right\|$$

$$\overset{(a)}{\leq}c_1\left\|\sum_{i\in[n_t-1]}(\hat{\mathbf{Z}}_i-\hat{\mathbf{Z}}_i')\right\|$$

$$\leq c_1\left\|\sum_{i\in[n_t-1]}(\hat{\mathbf{Z}}_i-\mathbf{Z}_i)\right\|$$

$$+c_1\left\|\sum_{i\in[n_t-1]}(\mathbf{Z}_i-\mathbf{Z}_i')\right\|+c_1\left\|\sum_{i\in[n_t-1]}(\mathbf{Z}_i'-\hat{\mathbf{Z}}_i')\right\|,$$

where $(a)$ is due to the assumption that $\ell' \leq c_1$. Plugging the results of Lemma E.2 and Lemma E.3 into the above inequality, we have $\left\|\sum_{i\in[n_t-1]}\left(\ell'(\mathbf{e}_i^\top\hat{\mathbf{Z}}'\mathbf{w}^\star,\mathbf{e}_i^\top\mathbf{Y})\hat{\mathbf{Z}}_i-\ell'(\mathbf{e}_i^\top\hat{\mathbf{Z}}'\mathbf{w}^\star,\mathbf{e}_i^\top\mathbf{Y})\hat{\mathbf{Z}}_i'\right)\right\| \leq$ $c_1\sqrt{F(n_t-1)}(2\epsilon_1+\epsilon_2)$.

Now we are ready to bound $\Delta$ for feature unlearning scenarios as follows:

$$\|\Delta\| = 2c + \frac{c\gamma_1}{\lambda}F(1+\epsilon_1)(2\epsilon_1+\epsilon_2) + c_1\sqrt{F(n_t-1)}(2\epsilon_1+\epsilon_2).$$

Consequently, combining the results in Inequality (14) and Inequality (16), we have

$$\left\|\nabla\mathcal{L}(\mathbf{w}^-,\mathcal{D}')\right\|$$
$$\leq \gamma_1\epsilon_1 F\left(\frac{c}{\lambda}+\left\|\hat{\mathbf{H}}_{\mathbf{w}^\star}^{-1}\Delta\right\|\right) + c_1\sqrt{F(n_t-1)}\epsilon_1 + \gamma_1(1+\epsilon_1)^2 F\left\|\hat{\mathbf{H}}_{\mathbf{w}^\star}^{-1}\Delta\right\|$$
$$\leq \frac{c\gamma_1}{\lambda}F\epsilon_1 + c_1\sqrt{F(n_t-1)}\epsilon_1 + \left(\gamma_1\epsilon_1 F + \gamma_1(1+\epsilon_1)^2 F\right)\frac{1}{\lambda(n_t-1)}\Delta$$
$$\overset{(a)}{\leq} \frac{c\gamma_1}{\lambda}F\epsilon_1 + c_1\sqrt{F(n_t-1)}\epsilon_1 + 2\gamma_1 F\frac{1}{\lambda(n_t-1)}\Delta.$$

Here, we omit the small term $(1+\epsilon_1)$ for simplicity. Then $(a)$ follows from $\gamma_1\epsilon_1 F \leq \gamma_1 F$. Similarly, $\Delta$ can be derived as

$$\Delta \leq 2c + \left(\frac{c\gamma_1}{\lambda}F + c_1\sqrt{F(n_t-1)}\right)(2\epsilon_1+\epsilon_2).$$

Thus, we have

$$\left\|\nabla\mathcal{L}(\mathbf{w}^-,\mathcal{D}')\right\|$$
$$\leq \frac{c\gamma_1}{\lambda}F\epsilon_1 + c_1\sqrt{F(n_t-1)}\epsilon_1 + \frac{4c\gamma_1 F}{\lambda(n_t-1)}$$
$$+ \frac{2\gamma_1 F}{\lambda(n_t-1)}\left(\frac{c\gamma_1}{\lambda}F + c_1\sqrt{F(n_t-1)}\right)(2\epsilon_1+\epsilon_2)$$
$$\leq \frac{4c\gamma_1 F}{\lambda(n_t-1)} + \left(\frac{c\gamma_1}{\lambda}F + c_1\sqrt{F(n_t-1)}\right)\left(\epsilon_1 + \frac{2\gamma_1 F}{\lambda(n_t-1)}(2\epsilon_1+\epsilon_2)\right).$$

Plugging Lemma E.5 into the above inequality, we have

$$\left\|\nabla\mathcal{L}(\mathbf{w}^-,\mathcal{D}')\right\|$$
$$\leq \frac{4c\gamma_1 F}{\lambda(n_t-1)} + \left(\frac{c\gamma_1}{\lambda}F + c_1\sqrt{F(n_t-1)}\right)\left(\epsilon_1 + \frac{2\gamma_1 F}{\lambda(n_t-1)}(2\epsilon_1+\sqrt{d(u)})\right).$$

Note that $c_1\sqrt{F(n_t-1)}\cdot\sqrt{d(u)}$ is much larger than $2c$, and thus $\frac{4c\gamma_1 F}{\lambda(n_t-1)}$ is much smaller than $c_1\sqrt{F(n_t-1)}\cdot\frac{2\gamma_1 F}{\lambda(n_t-1)}\cdot\sqrt{d(u)}$, which is include in the second term. Therefore, we omit $\frac{4c\gamma_1 F}{\lambda(n_t-1)}$ and adjust by enlarging $\frac{2\gamma_1 F}{\lambda(n_t-1)}$ to $\frac{4\gamma_1 F}{\lambda(n_t-1)}$ for simplicity. Similarly, we omit the $2epsilon_1$ term and enlarge $\sqrt{d(u)}$ to $2\sqrt{d(u)}$. Finally, we have the upper bound of $\|\nabla\mathcal{L}(\mathbf{w}^-,\mathcal{D}')\|$ as

$$\left(\frac{c\gamma_1}{\lambda}F + c_1\sqrt{F(n_t-1)}\right)\left(\epsilon_1 + \frac{8\gamma_1 F}{\lambda(n_t-1)}\cdot\sqrt{d(u)}\right).$$

$\square$

### E.6 Proof of Theorem 4.3

**Theorem.** *Suppose that Assumption 4.1 holds, and the edge $(u,v)$ is to be unlearned. If $\forall j \in [F]$, $\left\|\hat{\mathbf{Z}}\mathbf{e}_j - \mathbf{Z}\mathbf{e}_j\right\| \leq \epsilon_1$, we can bound $\|\nabla\mathcal{L}(\mathbf{w}^-,\mathcal{D}')\|$ by*

$$\frac{4c\gamma_1 F}{\lambda n_t} + \left(\frac{c\gamma_1}{\lambda}F + c_1\sqrt{Fn_t}\right)\left(\epsilon_1 + \frac{4\gamma_1 F}{\lambda n_t}(\epsilon_1 + \frac{4}{\sqrt{d(u)}} + \frac{4}{\sqrt{d(v)}})\right).$$

In the proof of Theorem 4.3, we also denote the upper bound of $\|\mathbf{Z}\mathbf{e}_j - \mathbf{Z}'\mathbf{e}_j\|$ for all $j \in [F]$ as $\epsilon_2$ before the conclusion is derived.

**Lemma E.5.** *Suppose that edge $e = (u,v)$ is to be removed. Then, we have $\|\mathbf{Z}\mathbf{e}_j - \mathbf{Z}'\mathbf{e}_j\| \leq \frac{4}{\sqrt{d(u)}} + \frac{4}{\sqrt{d(v)}}$ for all $j \in [F]$.*

*Proof.* We can derive that $\left\|\sum_{i\in[n_t]}\left(\hat{\mathbf{Z}}_i - \mathbf{Z}_i\right)\right\| \leq \sqrt{Fn_t}\epsilon_1$ and $\left\|\sum_{i\in[n_t]}\left(\mathbf{Z}'_i - \mathbf{Z}_i\right)\right\| \leq \sqrt{Fn_t}\epsilon_2$ by substituting $n_t - 1$ by $n_t$ in the proofs of Lemma E.2 and Lemma E.3. Thus, by plugging the above results into the proof of Theorem 4.2, we derive that $\|\nabla\mathcal{L}(\mathbf{w}^-, \mathcal{D}')\|$ is less than

$$\gamma_1\epsilon_1 F\left(\frac{c}{\lambda} + \frac{1}{n_t}\|\Delta\|\right) + c_1\sqrt{Fn_t}\epsilon_1 + \gamma_1(1+\epsilon_1)^2 F\left\|\frac{1}{n_t}\Delta\right\|$$

In the edge unlearning scenario, we have

$$\Delta = \sum_{i\in[n_t]}\left(\nabla l(\hat{\mathbf{Z}}, \mathbf{w}^\star, i) - \nabla l(\hat{\mathbf{Z}}', \mathbf{w}^\star, i)\right).$$

Similar to the analysis of the third term of $\Delta$ in the proof of Theorem 4.2, we have

$$\|\Delta\| \leq \frac{c\gamma_1}{\lambda}F(1+\epsilon_1)(2\epsilon_1 + \epsilon_2) + c_1\sqrt{Fn_t}(2\epsilon_1 + \epsilon_2).$$

Plugging Lemma E.5, we derive the upper bound of $\|\nabla\mathcal{L}(\mathbf{w}^-, \mathcal{D}')\|$ as

$$\frac{4c\gamma_1 F}{\lambda n_t} + \left(\frac{c\gamma_1}{\lambda}F + c_1\sqrt{Fn_t}\right)\left(\epsilon_1 + \frac{2\gamma_1 F}{\lambda n_t}(2\epsilon_1 + \frac{4}{\sqrt{d(u)}} + \frac{4}{\sqrt{d(v)}})\right).$$

$\square$

### E.7 PROOF OF THEOREM 4.4

**Theorem.** *Suppose that Assumption 4.1 holds and the feature of node $u$ is to be unlearned. If $\forall j \in [F]$, $\left\|\hat{\mathbf{Z}}\mathbf{e}_j - \mathbf{Z}\mathbf{e}_j\right\| \leq \epsilon_1$, we can bound $\|\nabla\mathcal{L}(\mathbf{w}^-, \mathcal{D}')\|$ by*

$$\frac{4c\gamma_1 F}{\lambda(n_t-1)} + \left(\frac{c\gamma_1}{\lambda}F + c_1\sqrt{F(n_t-1)}\right)\left(\epsilon_1 + \frac{2\gamma_1 F}{\lambda(n_t-1)}(2\epsilon_1 + 4\sqrt{d(u)} + \sum_{w\in\mathcal{N}(u)}\frac{4}{\sqrt{d(w)}})\right).$$

**Lemma E.6.** *Suppose that node $u$ is to be removed. Then, for all $j \in [F]$, it holds that:*

$$\|\mathbf{Z}\mathbf{e}_j - \mathbf{Z}'\mathbf{e}_j\| \leq 4\sqrt{d(u)} + \sum_{w\in\mathcal{N}(u)}\frac{4}{\sqrt{d(w)}}$$

*Proof.* The node unlearning scenario is identical to the node feature unlearning scenario except for $\|\mathbf{Z}\mathbf{e}_j - \mathbf{Z}'\mathbf{e}_j\|$. According to the proof of Theorem 4.2, we have

$$\left\|\nabla\mathcal{L}(\mathbf{w}^-, \mathcal{D}')\right\| \leq \frac{4c\gamma_1 F}{\lambda(n_t-1)} + \left(\frac{c\gamma_1}{\lambda}F + c_1\sqrt{F(n_t-1)}\right)\left(\epsilon_1 + 2\gamma_1 F\frac{1}{\lambda(n_t-1)}(2\epsilon_1 + \epsilon_2)\right).$$

Plugging Lemma E.6 into the above inequality, we derive the conclusion that $\|\mathbf{Z}\mathbf{e}_j - \mathbf{Z}'\mathbf{e}_j\|$ is bounded by

$$\frac{4c\gamma_1 F}{\lambda(n_t-1)} + \left(\frac{c\gamma_1}{\lambda}F + c_1\sqrt{F(n_t-1)}\right)\left(\epsilon_1 + \frac{2\gamma_1 F}{\lambda(n_t-1)}(2\epsilon_1 + 4\sqrt{d(u)} + \sum_{w\in\mathcal{N}(u)}\frac{4}{\sqrt{d(w)}})\right)$$

$\square$

### E.8 PROOF OF LEMMA E.1

**Lemma.** *Suppose that the feature of node $u$ is to be removed. Then, we have $\|\mathbf{Z}\mathbf{e}_j - \mathbf{Z}'\mathbf{e}_j\| \leq \sqrt{d(u)}$ for all $j \in [F]$.*

*Proof.* Let $\mathbf{z}$ and $\mathbf{z}'$ be the $j$-th column of $\mathbf{Z}$ and $\mathbf{Z}'$, respectively. Correspondingly, let $\{\boldsymbol{r}^{(\ell)}\}$, $0 \leq t \leq L$ are the residues of $\mathbf{z}$. And $\{\boldsymbol{r}'^{(\ell)}\}$ denotes the residues after the update operation for

residues but before invoking Algorithm 2 to reduce the error further. Thus, $\mathbf{z}$ and $\mathbf{z}'$ shares the same $\{\boldsymbol{q}^{(\ell)}\}$. First, we consider the difference between $\mathbf{z}$ and $\mathbf{z}'$. We have

$$\mathbf{z} - \mathbf{z}' = \sum_{\ell=0}^{L} w_\ell \sum_{t=0}^{\ell} \mathbf{D}^{-\frac{1}{2}} (\mathbf{A}\mathbf{D}^{-1})^{\ell-t} \left( \boldsymbol{r}^{(\ell)} - \boldsymbol{r}'^{(\ell)} \right).$$

Let $\Delta \boldsymbol{r}$ is the difference between $\boldsymbol{r}$ and $\boldsymbol{r}'$, that is, $\Delta \boldsymbol{r}^{(\ell)} = \boldsymbol{r}'^{(\ell)} - \boldsymbol{r}^{(\ell)}$. Note that the node feature unlearning scenario does not revise the graph structure. Thus, the residues of the neighbors of $u$ and their neighbors are not updated. Since only $\boldsymbol{r}^{(0)}(u)$ is updated, we have

$$\|\mathbf{z} - \mathbf{z}'\|_1 \leq \sum_{\ell=0}^{L} |w_\ell| \, \mathbf{D}^{-\frac{1}{2}} (\mathbf{A}\mathbf{D}^{-1})^\ell \left| \Delta \boldsymbol{r}^{(0)}(u) \right| \mathbf{e}_u.$$

Since $\mathbf{A}\mathbf{D}^{-1}$ is a left stochastic matrix and $\mathbf{D}^{-\frac{1}{2}}$ will not increase the result, we derive that

$$S \leq \sum_{\ell=0}^{L} |w_\ell| \left| \Delta \boldsymbol{r}^{(0)}(u) \right| \mathbf{e}_u.$$

Due to the fact that $\sum_{\ell=0}^{L} |w_\ell| \leq 1$, it can be further bounded by $\left| \Delta \boldsymbol{r}^{(0)}(u) \right|$. As illustrated in Section 3, $\boldsymbol{r}'^{(0)}$ is modified to $-\boldsymbol{q}^{(0)}(u)$. Note that Equation (1) holds for all settings of $r_{\max}$. When $r_{\max} = 0$, $\boldsymbol{r}^{(0)}$ is a zero vector. Combining Lemma 3.2, we have $\boldsymbol{q}^{(0)}(u) = d(u)^{\frac{1}{2}} \mathbf{x}(u)$. Therefore, $\left| \Delta \boldsymbol{r}^{(0)} \right| = d(u)^{\frac{1}{2}} |\mathbf{x}(u)| \leq d(u)^{\frac{1}{2}}$ due to the fact that $\mathbf{x}$ is normalized such that $\left\| \mathbf{D}^{\frac{1}{2}} \mathbf{x} \right\|_1 \leq 1$. Consequently, we have $\|\mathbf{z} - \mathbf{z}'\|_1 \leq d(u)^{\frac{1}{2}}$. Thus, the $L2$-norm, which is always less than $L1$-norm, is bounded by $d(u)^{\frac{1}{2}}$, too. This holds for all $j \in [F]$, which completes the proof. $\qquad \square$

### E.9    PROOF OF LEMMA E.5

**Lemma.** *Suppose that edge $e = (u, v)$ is to be removed. Then, we have $\|\mathbf{Z}\mathbf{e}_j - \mathbf{Z}'\mathbf{e}_j\| \leq \frac{4}{\sqrt{d(u)}} + \frac{4}{\sqrt{d(v)}}$ for all $j \in [F]$.*

*Proof.* Similar to the proof of Lemma E.1, we have

$$\mathbf{z} - \mathbf{z}' = \sum_{\ell=0}^{L} w_\ell \mathbf{D}^{-\frac{1}{2}} \left( \boldsymbol{q}^{(\ell)} + \sum_{t=0}^{\ell} (\mathbf{A}\mathbf{D}^{-1})^{\ell-t} \boldsymbol{r}^{(t)} \right)$$

$$- \sum_{\ell=0}^{L} w_\ell \mathbf{D}'^{-\frac{1}{2}} \left( \boldsymbol{q}^{(\ell)} + \sum_{t=0}^{\ell} (\mathbf{A}'\mathbf{D}'^{-1})^{\ell-t} \boldsymbol{r}'^{(t)} \right),$$

where $\mathbf{A}'$ and $\mathbf{D}'$ are the adjacency matrix and the degree matrix of the graph after the edge removal, respectively. Let $\Delta \boldsymbol{r}$ is the difference between $\boldsymbol{r}$ and $\boldsymbol{r}'$, that is, $\Delta \boldsymbol{r}^{(t)} = \boldsymbol{r}'^{(t)} - \boldsymbol{r}^{(t)}$. Then, we have

$$\mathbf{z} - \mathbf{z}'$$

$$= \sum_{t=0}^{\ell} w_\ell \sum_{t=0}^{\ell} \left( \mathbf{D}^{-\frac{1}{2}} (\mathbf{A}\mathbf{D}^{-1})^{\ell-t} \boldsymbol{r}^{(t)} - \mathbf{D}'^{-\frac{1}{2}} (\mathbf{A}'\mathbf{D}'^{-1})^{\ell-t} (\boldsymbol{r}^{(t)} + \Delta \boldsymbol{r}^{(t)}) \right)$$

$$+ \sum_{\ell=0}^{L} w_\ell \mathbf{D}^{-\frac{1}{2}} \boldsymbol{q}^{(\ell)} - \sum_{\ell=0}^{L} w_\ell \mathbf{D}'^{-\frac{1}{2}} \boldsymbol{q}^{(\ell)}$$

Note that the above equation holds under all settings of $r_{\max}$. Thus, we simply set $r_{\max} = 0$ and then $\{\boldsymbol{r}^{(t)}\}$ are a zero vector for $0 \leq t \leq L$. Therefore, it can be simplified as

$$\mathbf{z} - \mathbf{z}' = \sum_{\ell=0}^{L} w_\ell \sum_{t=0}^{\ell} \mathbf{D}'^{-\frac{1}{2}} (\mathbf{A}'\mathbf{D}'^{-1})^{\ell-t} \Delta \boldsymbol{r}^{(t)}$$

$$+ \sum_{\ell=0}^{L} w_\ell \mathbf{D}^{-\frac{1}{2}} \boldsymbol{q}^{(\ell)} - \sum_{\ell=0}^{L} w_\ell \mathbf{D}'^{-\frac{1}{2}} \boldsymbol{q}^{(\ell)}$$

Consider the $L1$-norm of $\mathbf{z} - \mathbf{z}'$. Note that the $L1$-norm of the first term is upper bounded by

$$S = \sum_{\ell=0}^{L} |w_\ell| \sum_{t=0}^{\ell} \sum_{i \in [n_t]} \mathbf{e}_i^\top \mathbf{D}'^{-\frac{1}{2}} (\mathbf{A}' \mathbf{D}'^{-1})^{\ell-t} \left| \Delta \boldsymbol{r}^{(t)} \right|,$$

where $\left| \Delta \boldsymbol{r}^{(t)} \right|$ represents the vector with the absolute value of each element in $\Delta \boldsymbol{r}^{(t)}$. Since $\mathbf{A}' \mathbf{D}'^{-1}$ is a left stochastic matrix and $\mathbf{D}'^{-\frac{1}{2}}$ will not increase the result, we derive that

$$S \le \sum_{\ell=0}^{L} |w_\ell| \sum_{t=0}^{\ell} \sum_{i \in [n_t]} \mathbf{e}_i^\top \left| \Delta \boldsymbol{r}^{(t)} \right|.$$

Due to the fact that $\sum_{\ell=0}^{L} |w_\ell| \le 1$, it can be further bounded by $\sum_{t=0}^{L} \sum_{i \in [n_t]} \mathbf{e}_i^\top \left| \Delta \boldsymbol{r}^{(t)} \right|$, that is, the sum of the absolute value of $\boldsymbol{r}^{(t)}$ for $0 \le t \le L$. According to Algorithm 3, only the residues of node $u$, node $v$, and their neighbors will be updated. The sum induced by node $u$ can be written as

$$S_u = \left( (d(u) + 1)^{\frac{1}{2}} - d(u)^{\frac{1}{2}} \right) |\mathbf{x}(u)|$$

$$+ \sum_{t=1}^{L} \frac{\left| \boldsymbol{q}^{(t-1)}(u) \right|}{d(u) + 1} + \sum_{w \in \mathcal{N}(u)} \frac{\left| \boldsymbol{q}^{(t-1)}(u) \right|}{d(u)(d(u) + 1)},$$

where the first term is the decreasement of $\boldsymbol{r}^{(0)}(u)$, the second term is the decreasement of the residues of $v$, and the third term is the increase of the residues of the neighbors of $u$. Note that $S_u \le \frac{|\mathbf{x}(u)|}{2\sqrt{d(u)}} + \frac{2}{d(u)+1} \sum_{t=1}^{L} \left| \boldsymbol{q}^{(t-1)}(u) \right|$ due to the fact that $\left( (d(u) + 1)^{\frac{1}{2}} - d(u)^{\frac{1}{2}} \right) = \frac{1}{\sqrt{d(u)} + \sqrt{d(u)+1}} \le \frac{1}{2\sqrt{d(u)}}$ and the new degree of $u$ is $d(u)$. Note that $\sum_{t=1}^{L} \left| \boldsymbol{q}^{(t-1)}(u) \right| \le 1$ and $|\mathbf{x}(u)| \le 1$ since $\mathbf{x}$ is normalized such that $\left\| \mathbf{D}^{\frac{1}{2}} \mathbf{x} \right\|_1 \le 1$. Thus, we have $S_u \le \frac{1}{2\sqrt{d(u)}} + \frac{2}{d(u)+1} \le \frac{3}{\sqrt{d(u)}}$. Similarly, the sum induced by node $v$ can be bounded by $\frac{1}{2\sqrt{d(v)}} + \frac{2}{d(v)+1} \le \frac{3}{\sqrt{d(v)}}$.

Consider the second term. Only the degree of node $u$ and node $v$ is modified. Thus, we have

$$\left\| \sum_{\ell=0}^{L} w_\ell \mathbf{D}^{-\frac{1}{2}} \boldsymbol{q}^{(\ell)} - \sum_{\ell=0}^{L} w_\ell \mathbf{D}'^{-\frac{1}{2}} \boldsymbol{q}^{(\ell)} \right\|_1$$

$$= \sum_{\ell=0}^{L} |w_\ell| \left( d(u)^{-\frac{1}{2}} - (d(u) + 1)^{-\frac{1}{2}} \right) \left| \boldsymbol{q}^{(\ell)}(u) \right|$$

$$+ \sum_{\ell=0}^{L} |w_\ell| \left( d(v)^{-\frac{1}{2}} - (d(v) + 1)^{-\frac{1}{2}} \right) \left| \boldsymbol{q}^{(\ell)}(v) \right|.$$

The term $\left( d(u)^{-\frac{1}{2}} - (d(u) + 1)^{-\frac{1}{2}} \right)$ is less than $1/d(u)$. Thus, the second term is upper bounded by $\frac{1}{d(u)} + \frac{1}{d(v)}$. Sum up the two terms, we have $\|\mathbf{z} - \mathbf{z}'\|_1 \le \frac{4}{\sqrt{d(u)}} + \frac{4}{\sqrt{d(v)}}$. Thus, the $L2$-norm, which is always less than $L1$-norm, is bounded by $\frac{4}{\sqrt{d(u)}} + \frac{4}{\sqrt{d(v)}}$, too. The lemma follows. $\qquad \square$

## E.10 PROOF OF LEMMA E.6

**Lemma.** *Suppose that node $u$ is to be removed. Then, for all $j \in [F]$, it holds that:*

$$\|\mathbf{Z} \mathbf{e}_j - \mathbf{Z}' \mathbf{e}_j\| \le 4\sqrt{d(u)} + \sum_{w \in \mathcal{N}(u)} \frac{4}{\sqrt{d(w)}}$$

*Proof.* Removing a node is equivalent to removing all edges connected to the node. Thus, $\|\mathbf{Z} \mathbf{e}_j - \mathbf{Z}' \mathbf{e}_j\|$ can be bounded by the sum of the bounds of removing all edges from node $u$. Drawing the result from Lemma E.5, the bound induced by removing one single edge can be bounded by $\frac{4}{\sqrt{d(u)}} + \frac{4}{\sqrt{d(w)}}$. Thus, the total sum is less than $4\sqrt{d(u)} + \sum_{w \in \mathcal{N}(u)} \frac{4}{\sqrt{d(w)}}$ by the triangle inequality. The lemma follows. $\qquad \square$

### E.11 PROOF OF THEOREM 4.5

**Theorem.** *Let $\boldsymbol{R}^{(t)} \in \mathbb{R}^{n \times F}$ denote the residue matrix at level $t$, where the $j$-th column $\boldsymbol{R}^{(t)}\mathbf{e}_j$ represents the residue at level $t$ for the $j$-th feature vector. Let $\boldsymbol{R}$ be defined as the sum $\sum_{t=0}^{L} \boldsymbol{R}^{(t)}$. Consequently, we can establish the following data-dependent bound:*

$$\left\|\nabla \mathcal{L}(\mathbf{w}^-, \mathcal{D}')\right\| \le 2c_1 \left\|\mathbf{1}^\top \boldsymbol{R}\right\| + \gamma_2 \left\|\hat{\mathbf{Z}}'\right\| \left\|\hat{\mathbf{H}}_{\mathbf{w}^\star}^{-1}\Delta\right\| \left\|\hat{\mathbf{Z}}'\hat{\mathbf{H}}_{\mathbf{w}^\star}^{-1}\Delta\right\|.$$

*Proof.* As demonstrated in the proof of Theorem 4.2, it is established that

$$\left\|\nabla \mathcal{L}(\mathbf{w}^-, \mathcal{D}')\right\| \le \left\|\nabla \mathcal{L}(\mathbf{w}^-, \mathcal{D}') - \nabla \mathcal{L}(\mathbf{w}^-, \hat{\mathcal{D}}')\right\| + \left\|\nabla \mathcal{L}(\mathbf{w}^-, \hat{\mathcal{D}}')\right\|.$$

Initially, the data-dependent bound of the second term $\left\|\nabla \mathcal{L}(\mathbf{w}^-, \hat{\mathcal{D}}')\right\|$ can be inferred directly from Corollary 1 in (Guo et al., 2019), by substituting $\mathcal{D}'$ with $\hat{\mathcal{D}}'$. This leads to the conclusion that $\left\|\nabla \mathcal{L}(\mathbf{w}^-, \hat{\mathcal{D}}')\right\| \le \gamma_2 \left\|\hat{\mathbf{Z}}'\right\| \left\|\hat{\mathbf{H}}_{\mathbf{w}^\star}^{-1}\Delta\right\| \left\|\hat{\mathbf{Z}}'\hat{\mathbf{H}}_{\mathbf{w}^\star}^{-1}\Delta\right\|$.

Focusing now on the first term.

$$\left\|\nabla \mathcal{L}(\mathbf{w}^-, \mathcal{D}') - \nabla \mathcal{L}(\mathbf{w}^-, \hat{\mathcal{D}}')\right\|$$
$$= \left\| \sum_{i \in [n_t]} \left( l'(\mathbf{Z}', \mathbf{w}^-, i)\mathbf{Z}'_i - l'(\hat{\mathbf{Z}}', \mathbf{w}^-, i)\hat{\mathbf{Z}}'_i \right) \right\|.$$

Since $l'$ is bounded by $c_1$, we have

$$\left\| \sum_{i \in [n_t]} \left( l'(\mathbf{Z}', \mathbf{w}^-, i)\mathbf{Z}'_i - l'(\hat{\mathbf{Z}}', \mathbf{w}^-, i)\hat{\mathbf{Z}}'_i \right) \right\|$$
$$\le 2c_1 \left\| \sum_{i \in [n_t]} \left( \mathbf{Z}'_i - \hat{\mathbf{Z}}'_i \right) \right\|$$
$$= 2c_1 \sqrt{ \sum_{j \in [F]} \left( \sum_{i \in [n_t]} \mathbf{Z}'_{ij} - \hat{\mathbf{Z}}'_{ij} \right)^2 }. \tag{19}$$

Let $\mathbf{z}' = \mathbf{Z}'\mathbf{e}_j$ and $\hat{\mathbf{z}}' = \hat{\mathbf{Z}}'\mathbf{e}_j$. According to Equation (1), we have

$$\mathbf{z}' - \hat{\mathbf{z}}' = \sum_{\ell=0}^{L} w_\ell \sum_{t=0}^{\ell} \mathbf{D}^{-\frac{1}{2}}(\mathbf{A}\mathbf{D}^{-1})^{\ell-t}\boldsymbol{r}^{(t)}.$$

Given that $\mathbf{A}\mathbf{D}^{-1}$ forms a left stochastic matrix, the sum of the entries in $\boldsymbol{r}^{(t)}$ is equivalent to the sum of entries in $(\mathbf{A}\mathbf{D}^{-1})^{\ell-t}\boldsymbol{r}^{(t)}$. And $\mathbf{D}^{-\frac{1}{2}}$ will not increase the result. Thus, we have

$$\sum_{i \in [n_t]} \mathbf{z}'_i - \hat{\mathbf{z}}'_i \le \sum_{\ell=0}^{L} w_\ell \sum_{t=0}^{\ell} \mathbf{1}^\top \boldsymbol{r}^{(t)} \le \sum_{t=0}^{L} \mathbf{1}^\top \boldsymbol{r}^{(t)},$$

due to the fact that $\sum_{\ell=0}^{L} |w_\ell| \le 1$. Therefore, Term (19) can be bounded by $2c_1 \left\|\mathbf{1}^\top \boldsymbol{R}\right\|$. This completes the proof.

$\square$

### E.12 PROOF OF LEMMA E.2 AND LEMMA E.3

We prove these two lemmas directly by the Cauchy-Schwarz inequality. The followings show the proof of Lemma E.2 and the other can be derived similarly.

*Proof.* Note that

$$\left\| \sum_{i \in [n_t - 1]} \left( \hat{\mathbf{Z}}_i - \mathbf{Z}_i \right) \right\| = \sqrt{\sum_{j \in [F]} \left( \sum_{i \in [n_t - 1]} (\hat{\mathbf{Z}}_{ij} - \mathbf{Z}_{ij}) \right)^2}.$$

With the Cauchy-Schwarz inequality, we derive that

$$\sqrt{\sum_{j \in [F]} \left( \sum_{i \in [n_t - 1]} (\hat{\mathbf{Z}}_{ij} - \mathbf{Z}_{ij}) \right)^2} \leq \sqrt{\sum_{j \in [F]} (n_t - 1) \sum_{i \in [n_t - 1]} (\hat{\mathbf{z}}_{ij} - \mathbf{z}_{ij})^2}.$$

Given that $\left\| \hat{\mathbf{Z}} \mathbf{e}_j - \mathbf{Z} \mathbf{e}_j \right\| \leq \epsilon_1$, we have

$$\left\| \sum_{i \in [n_t - 1]} \left( \hat{\mathbf{Z}}_i - \mathbf{Z}_i \right) \right\| \leq \sqrt{F(n_t - 1)} \epsilon_1.$$

$\square$

