# OpenReview forum: "Scalable and Certifiable Graph Unlearning: Overcoming the Approximation Error Barrier"
_ICLR.cc/2025/Conference — ICLR 2025 Spotlight_

### Official Review · Reviewer_N1R8 · 2024-10-18

**Soundness:** 3
**Presentation:** 2
**Contribution:** 3
**Rating:** 8
**Confidence:** 3

**Summary:**

This paper examines and highlights the efficiency issue of existing certified graph unlearning approaches. The authors show that the culprit is the costly re-computation of graph propagation. To address this challenge, the authors propose ScaleGUN, an efficient certified graph unlearning method that integrates the approximate graph propagation technique to speed up the process. Despite being introduced with additional approximation errors, ScaleGUN's model error is bounded with theoretical guarantees.

**Strengths:**

1. Certified graph unlearning is an important research topic with potentially broach impact in both academia and industry.
2. The theoretical analyses and proofs in this paper are mostly rigorous.

**Weaknesses:**

1. The assumption, analysis, and proposed methodology seem to be limited to SGC-like models (GNNs with pre-computed graph propagation). The efficiency issue may also not be evident when the backbone model is not SGC (e.g., multi-layer GCN).
2. The main contribution of this paper mainly lies in theory, while the methodology part seems to be just a simple combination of InstantGNN and CGU.
3. The clarity and presentation of this paper can be further improved. For example, it feels quite strange when the authors suddenly bring up PageRank in Line 100 without any bedding or motivation.

**Questions:**

1. Would the efficiency issue still be a problem when CGU/CEU uses other GNNs as the backbone model (e.g., multi-layer GCN)? Is ScaleGUN compatible with other GNN models? Would the model error of ScaleGUN be still bounded when using other GNNs as the backbone model?
2. What is the main difference between the approximate propagation module in ScaleGUN and that in InstantGNN?
3. It would be better if the authors can also plot the bounds of other baseline methods (CGU and CEU) in Figure 2, so as to show that ScaleGUN is on par with them.
4. To avoid ambiguity and improve readability, the authors are advised to explain or formally define $\nabla\mathcal{L}(\mathbf{w}^\star,\mathcal{D})-\nabla\mathcal{L}(\mathbf{w}^\star,\mathcal{D^\prime})$. The reader may incorrectly interpret that $\mathcal{L}(\cdot)$ here is the loss function of all training samples, whereas it is actually the loss function of those removed.
5. There are a few typos or grammatical errors, such as
    1. Page 1 Line 049 & Page 2 Line 83: the gradient residual norm is missing the vector differential operator $\nabla$.
    2. Page 4 Line 188: "parallel" should be corrected as "parallelly"

---

> ### Author Response · Authors · 2024-11-20
> **Response to Reviewer N1R8 (Part 1/4)**
>
> Thank you for these helpful comments. Our detailed answers are provided below.
>
> > Q1-1. "*Would the efficiency issue still be a problem when CGU/CEU uses other GNNs as the backbone model (e.g., multi-layer GCN)?*"
>
> First, we would like to clarify that CGU/CEU cannot use other GNNs as backbone models if certified guarantees are required. This limitation arises because the theoretical analysis of certified unlearning relies on the assumption of a strongly convex loss function. To the best of our knowledge, all certified unlearning methods are restricted to linear models with strongly convex loss functions.
> Second, the efficiency issue remains significant even when using other GNNs as the backbone model. Training GNNs on large graphs is inherently challenging due to the time-consuming and memory-intensive graph propagation, a core operation in nearly all GNNs. In this paper, we address the efficiency issue in certified unlearning by introducing an approximate propagation technique. While certified unlearning is limited to linear models, ScaleGUN can also be applied to deep models when theoretical guarantees are not required. We elaborate on these points below.
>
> **The efficiency issue is severe regardless of whether SGC is the backbone.** This is because graph propagation, serving as a key component in GNN models, incurs substantial time and memory costs. This challenge has spurred extensive research into accelerating GNN training [R1, R2, R3]. For instance, as noted in ClusterGCN [R1], the time complexity of GCN is $O(LmF)$, where $L$ represents the number of graph convolution layers, $m$ is the number of edges, and $F$ is the feature dimension. For large-scale graphs with billions of edges, such as ogbn-papers100M, this computational cost is prohibitive. Other GNN models face similar efficiency challenges, as they typically adopt similar graph propagation schemes akin to GCN.
>
> **Efficiency is an open problem in (certified) graph unlearning.** In the context of graph unlearning, most methods (including those that do not use SGC as a backbone [R4, R5]) first perform graph re-propagation after the graph data is modified, followed by updates to the model parameters. While existing unlearning methods focus on optimizing parameter updates, they overlook the significant time costs of graph propagation for each unlearning request, as discussed in our motivations (Lines 63–76). Consequently, these methods fail to achieve efficient unlearning on large-scale graphs, where re-propagation costs dominate total unlearning costs, as seen with ogbn-papers100M in Figure 1. To our best knowledge, no existing graph unlearning methods can handle billion-edge graphs efficiently, even without certified guarantees. In this paper, ScaleGUN solves the efficiency issue for linear models with certified unlearning guarantees (scaling to ogbn-papers100M in Table 1), and can apply to nonlinear models when certified guarantees are relaxed (scaling to ogbn-papers100M in Table 3).
>
> [R1] Wei-Lin Chiang, Xuanqing Liu, Si Si, Yang Li, Samy Bengio, and Cho-Jui Hsieh. Cluster-gcn: An efficient algorithm for training deep and large graph convolutional networks.
>
> [R2] Hanqing Zeng, Hongkuan Zhou, Ajitesh Srivastava, Rajgopal Kannan, and Viktor Prasanna. Graph-saint: Graph sampling based inductive learning method.
>
> [R3] William L. Hamilton, Rex Ying, Jure Leskovec. Inductive Representation Learning on Large Graphs.
>
> [R4] Kun Wu, Jie Shen, Yue Ning, Ting Wang, and Wendy Hui Wang. Certified edge unlearning for graph neural networks.
>
> [R5] Jiancan Wu, Yi Yang, Yuchun Qian, Yongduo Sui, Xiang Wang, and Xiangnan He. Gif: A general graph unlearning strategy via influence function.

---

> ### Author Response · Authors · 2024-11-20
> **Response to Reviewer N1R8 (Part 2/4)**
>
> > W1. "*The assumption, analysis, and proposed methodology seem to be limited to SGC-like models (GNNs with pre-computed graph propagation)...*"
> >
> > Q1-2. "*Is ScaleGUN compatible with other GNN models? Would the model error of ScaleGUN be still bounded when using other GNNs as the backbone model?*"
>
> When theoretical guarantees are not required, ScaleGUN achieves competitive model utility using deep models as backbones. This is demonstrated in Table 3 (using deep decoupled models) and Table 14 (using spectral GNNs). To the best of our knowledge, ScaleGUN is the first scalable and effective unlearning mechanism for heuristic (non-certified) graph unlearning capable of handling billion-edge graphs.
> We have not derived the bounds for model error when ScaleGUN utilizes nonlinear models as backbones, as this is a nontrivial task and beyond the scope of this paper. However, we empirically evaluate the unlearning efficacy on deep models in Figure 6, which shows that ScaleGUN can effectively unlearn graph data in nonlinear models, behaving similarly to retraining.
>
> Next, we explain why ScaleGUN is limited to linear models (SGC-like models) when theoretical guarantees are required. While our ultimate goal is to develop a scalable and certified graph unlearning mechanism for nonlinear GNNs and apply it to real-world applications, to the best of our knowledge, no existing certified graph unlearning methods can apply to general GNNs. Achieving certified unlearning guarantees for nonlinear models, including nonlinear GNNs, remains an open and highly nontrivial challenge. In this work, we focus on certified unlearning and prioritize scalability in order to make it applicable to real-world applications. We take the first step by providing scalable and certified graph unlearning for linear models. We elaborate on these points as follows:
>
> 1. **Achieving certified unlearning for nonlinear models is highly nontrivial and beyond the scope of this paper.** Specifically, certified unlearning requires that the unlearned model is approximately equivalent to retraining in terms of their probability distributions (Line 127-131 of our manuscript). This indicates that the unlearned model must be able to approximate the new optimal point of empirical risk within a certain margin of error. However, existing studies[R6] have proven that even approximating the optimal value of a 2-layer ReLU neural network is NP-hard in worst cases. This demonstrates that it is nontrivial to achieve certified unlearning for nonlinear models and explains why existing certififed unlearning studies are limited to linear models.
>
> 2. **ScaleGUN aims at taking the first step toward making graph unlearning applicable to real-world applications.** While regulatory requirements for “the right to be forgotten” are driving research on graph unlearning, no existing methods scale to billion-edge graphs to our best knowledge, even without certified guarantees. This limitation hinders the applicability of graph unlearning in real-world scenarios (e.g., recommender systems, online social networks), as these applications often involve large-scale networks with billions of edges. As the first scalable and certified graph unlearning framework, ScaleGUN enables unlearning on billion-edge graphs and paves the way for graph unlearning to be applied in real-world applications.
>
> [R6] Yifei Wang and Mert Pilanci. Polynomial-Time Solutions for ReLU Network Training: A Complexity Classification via Max-Cut and Zonotopes.

---

> ### Author Response · Authors · 2024-11-20
> **Response to Reviewer N1R8 (Part 3/4)**
>
> > Q2. "*What is the main difference between the approximate propagation module in ScaleGUN and that in InstantGNN?*"
>
> 1. Propagation scheme
>
>     The approximate propagation module in ScaleGUN is based on the Generalized PageRank (GPR) propagation scheme, i.e., $\mathbf{Z}=\sum_{\ell=0}^L w_{\ell}(\mathbf{D}^{-1/2}\mathbf{A}\mathbf{D}^{-1/2})^{\ell} \mathbf{X}$, whereas InstantGNN and other existing dynamic propagation methods use the Personalized PageRank (PPR) propagation scheme, i.e., $\mathbf{Z}=\sum_{\ell=0}^\infty w_{\ell}(\mathbf{D}^{-1/2}\mathbf{A}\mathbf{D}^{-1/2})^{\ell} \mathbf{X}$ with $w_\ell=\alpha(1-\alpha)^\ell$, where $0<\alpha<1$ denotes the decay factor.
>     We choose GPR for the following reasons:
>
>     1. Most (certified) graph unlearning methods use GNN backbones with layered propagation schemes like GPR instead of PPR. Adopting GPR allows ScaleGUN to align with these methods and scale them to billion-edge graphs as well.
>     2. GPR provides a more general framework, as PPR is a special case of GPR. Theoretical analysis on GPR can be directly applied to PPR. Thus, ScaleGUN can support existing dynamic PPR approaches to create a certified unlearning method for PPR-based models.
>     3. Prior studies indicate that PPR-based GNNs behave as low-pass filters [R7] and are less effective for heterogeneous graphs [R8].
>
> 2. On theoretical analysis and implementations
>
>     PPR-based methods can simply maintain a reserve vector $\boldsymbol{q}$ and a residue vector $\boldsymbol{r}$. In contrast, GPR-based propagation requires $\boldsymbol{q}^{(\ell)}$ and $\boldsymbol{r}^{(\ell)}$ for each level $\ell$. These differences introduce unique challenges to algorithm implementation, the theoretical analysis of correctness and time complexity. To our knowledge, this is the first dynamic propagation algorithm designed for the GPR-based propagation scheme.
>
> Kindly note that we also provided discussions on the differences between ScaleGUN and existing dynamic propagation methods in Appendix D.1.
>
> [R7] Meiqi Zhu, Xiao Wang, Chuan Shi, Houye Ji, and Peng Cui. Interpreting and unifying graph neural networks with an optimization framework.
>
> [R8] Mingguo He, Zhewei Wei, and Ji-Rong Wen. Convolutional neural networks on graphs with chebyshev approximation, revisited.
>
> ---
>
> > W2. "*The main contribution of this paper mainly lies in theory, while the methodology part seems to be just a simple combination of InstantGNN and CGU.*"
>
> We appreciate your recognition of our contributions to graph unlearning theory. However, we would like to clarify that the methodology part is not a simple combination of InstantGNN and CGU.
> On one hand, our theoretical analysis directly affects the design of the unlearning algorithm. Without these theoretical bounds operating during the unlearning process, simply applying the CGU unlearning algorithm would not guarantee certified unlearning.
> On the other hand, as explained in our response to Q2, the propagation algorithm is distinct from InstantGNN. Below, we elaborate further on the differences in the unlearning algorithm.
>
> Simply combining InstantGNN and CGU cannot achieve scalable and certified graph unlearning, either in terms of theoretical guarantees for certified unlearning or the practical implementation of unlearning algorithms. This combination raises two key questions: 1. Will the approximation error introduced by approximate propagation lead to an unacceptably large model error? 2. What are the new model error bounds (both worst-case and data-dependent) after incorporating approximate propagation? These bounds are essential for selecting suitable privacy parameters in the unlearning algorithm, particularly the data-dependent bounds, which help guide decisions on when to retrain the model automatically during sequential unlearning. Without such bounds, it would be impossible to guarantee certified unlearning throughout the unlearning process.
>
> Through non-trivial theoretical analysis, we demonstrate that approximation error only marginally impacts the model error, while ensuring the total model error remains bounded. This critical insight allows us to achieve scalable and certified graph unlearning via approximate propagation. Moreover, we derive the worst-case and data-dependent bounds on model error in the context of approximate propagation. These finish the missing pieces in the puzzle of scalable and certified graph unlearning.

---

> ### Author Response · Authors · 2024-11-20
> **Response to Reviewer N1R8 (Part 4/4)**
>
> > Q3. "*It would be better if the authors can also plot the bounds of other baseline methods (CGU and CEU) in Figure 2*"
>
> We present the bounds of CGU and CEU in Figure 2 of our revised PDF. The figure illustrates the following key observations:
> 1. For ScaleGUN, CGU and CEU, the worst-case bounds and the data-dependent bounds validly upper bound the true value, and the worst-case bounds are looser than the data-dependent bounds. This confirms the correctness of the bounds for all methods.
> 2. The worst-case bound of ScaleGUN is larger than that of CGU and slightly larger than that of CEU. This difference arises because ScaleGUN introduces approximation error, which is not present in CGU and CEU. However, note that this large worst-case bound does not impact the update efficiency of ScaleGUN, as decisions on retraining during sequential unlearning rely on data-dependent bounds rather than worst-case bounds.
> 3. In edge and node unlearning scenarios, the data-dependent bound of ScaleGUN is slightly larger than that of CGU in the early stages of sequential unlearning. This occurs because the approximation error introduced by ScaleGUN dominates the model error when only a small amount of data has been unlearned. As the number of unlearning requests increases, the unlearning error accumulates, eventually dominating the model error, while the approximation error does not accumulate since we meet the approximation error bound in each unlearning step. Consequently, the data-dependent bound of ScaleGUN becomes comparable to that of CGU over time. In the edge unlearning scenario, the data-dependent bound of CEU is larger than that of both ScaleGUN and CGU, primarily due to CEU employing a different unlearning mechanism.
> 4. The true norm of ScaleGUN is slightly larger than that of CGU in edge and node unlearning scenarios, again due to the introduction of approximation error by ScaleGUN.
>
> > W3. "*The clarity and presentation of this paper can be further improved... bring up PageRank in Line 100 without any bedding or motivation*"
>
> Thank you for the suggestion. To improve clarity, we will revise the introduction of our contribution on the lazy local propagation framework in the following way:
>
> Among the existing approximate propagation techniques for decoupled models, algorithms for the Personalized PageRank (PPR) propagation scheme are well studied. However, existing certified graph unlearning models typically adopt the Generalized PageRank (GPR) propagation scheme. To bridge this gap, we extend these techniques to Generalized PageRank, enabling efficient propagation in widely used unlearning models.
>
> > Q4. notations
>
> The term $\nabla \mathcal{L}\left(\mathbf{w}^{\star}, \mathcal{D}\right)-\nabla \mathcal{L}\left(\mathbf{w}^{\star}, \mathcal{D}^{\prime}\right)$ represents the difference between the gradient of the loss of $\mathbf{w}^{\star}$ on the dataset before unlearning ( $\mathcal{D}$) and after unlearning ($\mathcal{D}^{\prime}$). This term quantifies the impact of the removed data on the gradient of the loss. Here, $\mathbf{w}^{\star}$ represents the optimal model paramter on the dataset before unlearning as stated in Line 138.
> As stated in Line 253 in our manuscript, $\mathcal{L}(\mathbf{w}, \mathcal{D})$ is defined as $\sum_{i \in\left[n_t\right]}\left(l\left(\mathbf{e}_i^{\top} {\mathbf{Z}} \mathbf{w}, \mathbf{e}_i^{\top} \mathbf{Y}\right)+\frac{\lambda}{2}\|\mathbf{w}\|^2\right)$.
>
> To improve readability, we will explain this term formally in our revision and add a notation table in the appendix for easy reference.
>
> > Q5. typos or grammatical errors
>
> Thank you for spotting the typos and grammatical errors. We will correct them in our revision.

---

> > ### Comment · Reviewer_N1R8 · 2024-11-20
> >
> > Thanks for the detailed answers to my questions. Most of my concerns have been properly addressed, so I decided to raise my rating accordingly.

---

> > > ### Author Response · Authors · 2024-11-25
> > >
> > > We sincerely thank you for your positive evaluation and for recognizing the value of our work. Your insightful comments have been instrumental in improving our manuscript. We truly appreciate the time and effort you dedicated to reviewing our submission.

---

### Official Review · Reviewer_skqz · 2024-10-26

**Soundness:** 3
**Presentation:** 3
**Contribution:** 4
**Rating:** 8
**Confidence:** 4

**Summary:**

The authors study the important problem of certified graph unlearning, where they focus on developing scalable graph unlearning algorithms that can work on extremely large graphs such as ogbn-papers100M. They identify the scalability issue of prior works that require recomputing the graph propagation, and alleviate such issue with an efficient push-forward algorithm that is commonly utilized in efficient PageRank computation. The certified unlearning guarantees are established and extensive experiments are done for large graph datasets, which demonstrates the superiority of the proposed method compared to prior works.

**Strengths:**

- Identify the scalability issue of the prior graph unlearning works.
- Combining error analysis of push-forward operation and certified graph unlearning framework in a clever way.
- Extensive experiment, especially conducted on graphs with at most 100M nodes.

**Weaknesses:**

I personally do not find major weaknesses in the paper. Yet some minor points can be addressed to improve the manuscript further.

- The notations can be hard to digest for readers who are not familiar with prior works.
- The difference in analysis compared to prior works should be further emphasized and elaborated.
- I feel the main experiments should be based on the setting $\delta=1/n$ or $\delta=1/|E|$ instead (currently deferred to Appendix), which provides a more meaningful privacy implication.

## Detail comments

Overall it is a joyful read. The authors study the important certified graph unlearning problem, where they identify the scalability issue in prior works and resolve it by introducing approaches from efficient PageRank computation methods. In theory, while both the error analysis of updating push-forward operation and certified graph unlearning are known, it is still not trivial to combine them correctly. I feel the authors made a great contribution here. On the practice side, the authors test with large graphs such as ogbn-papers100M, which is the largest graph to be tested in the context of certified graph unlearning to the best of my knowledge. The proposed algorithm makes an important step toward the practical certified graph unlearning approach, which I find it as another great contribution of the paper.

There are still some minor comments that can further improve the clarity and quality of the papers. First, I feel the notations can be further improved. Some notations are not even clearly defined, albeit I roughly understand the meaning since I am familiar with the field. For instance, what exactly is the definition of $\hat{D}$ and the other quantities with \hat? I guess it implies the quantities related to approximate propagation regarding equation (2) but it would be great to further elaborate the details. Also, it would be weird to say we have some “approximate dataset $\hat{D}$”. If I understand correctly, we only have $D,D^\prime$ in practice, and the “approximate quantity” is only regrading to $Z, Z^\prime$. I encourage the authors to think more carefully about the choice of the notation here.

Another comment is that I feel the difference in analysis compared to prior works is not emphasized enough and clearly explained. For instance, I cannot fully understand why the authors can deal with propagation matrix $P = D^{-1/2}A D^{-1/2}$ and the prior works only can do $P=D^{-1}A$. What exactly is the issue and what techniques that the authors use to resolve such issue? Now I only roughly know that it is handled by examining the approximation error of equation (2), but I feel the explanation can be clearer.

Finally, I personally think the authors should focus on the privacy setting of $\delta = 1/n$ or $1/|E|$ instead of a fix $\delta = 10^{-4}$, especially the number of edges and nodes are larger than 100M in the largest graph. As also pointed out by the authors, this is the common standard in the privacy graph learning literature. While the claim would not change and is fair (since all tested approaches follow the same privacy constraint), it is still preferable to make the setting as meaningful as possible in terms of privacy.

**Questions:**

1.	What is the definition of $\hat{D}$ and other quantities with \hat?

---

> ### Author Response · Authors · 2024-11-20
> **Response to Reviewer skqz (Part 1/2)**
>
> Thank you for these helpful comments and positive feedback. Our detailed answers are provided below.
>
> > Q1. "*the definition of $\hat{\mathcal{D}}$ and other quantities with \hat*"
>
> Your understanding is correct. Quantities with \hat represent the values computed using approximate propagation with our lazy local propagation framework. The term $\hat{\mathcal{D}}$ denotes that we use the approximate embedding in the context. For instance, $\mathcal{L}(\mathbf{w},\hat{\mathcal{D}})$ is defined as $\sum_{i \in\left[n_t\right]}\left(l\left(\mathbf{e}_i^{\top} \hat{\mathbf{Z}} \mathbf{w}, \mathbf{e}_i^{\top} \mathbf{Y}\right)+\frac{\lambda}{2}\|\mathbf{w}\|^2\right)$, where $\hat{\mathbf{Z}}$ represents the approximate embedding.
>
> > W1. improving notations
>
> Thank you for the suggestion. We have provided the explanation of \hat and \prime in Line 122-126 of our manuscript, however, we agree that the clarity could be further improved. In our revision, we will use $\mathcal{L}\_{\rm appr}$ to explicitly indicate that approximate propagation is used when computing the loss, instead of using $\hat{\mathcal{D}}$. Similarly, we will replace $\hat{\mathbf{Z}}$ with $\mathbf{Z}_{\rm appr}$ to denote approximate embeddings in order to prevent any potential confusion with $\hat{\mathbf{Z}}$ and $\mathbf{Z}'$. Furthermore, we will add a notation table in the appendix for easy reference.
>
> > W2. "*The difference in analysis compared to prior works should be further emphasized and elaborated.*"
>
> Thank you for the suggestion. The difference in analysis between prior works and ScaleGUN is significant, arising from two key aspects: 1. ScaleGUN computes approximate embeddings, whereas prior works compute exact embeddings. However, exact embeddings are necessary to bound the model error, making this a new challenge for ScaleGUN. 2. ScaleGUN adopts the propagation matrix $\mathbf{P}=\mathbf{D}^{-1/2}\mathbf{A}\mathbf{D}^{-1/2}$, while prior works (i.e., CGU) adopt $\mathbf{P}=\mathbf{D}^{-1}\mathbf{A}$. The first challenge can be addressed through approximation error analysis, as explained in Lines 301–313 of our manuscript. Below, we focus on the second challenge in detail.
>
> Choosing $\mathbf{P}=\mathbf{D}^{-1/2}\mathbf{A}\mathbf{D}^{-1/2}$ introduces additional difficulties in deriving bounds for $||\Delta||$ (a key component for bounding the model error, as defined in Line 258), requiring entirely different techniques compared to CGU. Specifically, the main challenge of bounding $||\Delta||$ lies in connecting the pre- and post-unlearning embeddings ($\mathbf{Z}$ and $\mathbf{Z}'$). In CGU, the authors choose to bound $||e_i (\mathbf{Z} - \mathbf{Z}')||$, i.e., the difference between the pre- and post-unlearning embedding for each node $i$. Because CGU adopts $\mathbf{P} = \mathbf{D}^{-1}\mathbf{A}$, a right stochastic matrix, it benefits from useful properties to simplify the analysis. For example, the row norms of $\mathbf{Z}$ are bounded by 1. The CGU authors emphasize that this choice of $\mathbf{P}$ is critical, as other choices will lead to worse bounds on the norm of each row of $\mathbf{Z}$.
> Therefore, we adopt a different approach due to our choice of $\mathbf{P}$. Instead of bounding $||e_i (\mathbf{Z} - \mathbf{Z}')||$ as in CGU, we leverage the local lazy framework to neatly bound $||(\mathbf{Z} - \mathbf{Z}') e_j||$, i.e., the difference in an embedding dimension pre- and post- unlearning. The details of this method are provided in Line 324-337 in our manuscript.
> We compute $(\mathbf{Z} - \mathbf{Z}') e_i$ using the equation in Line 329-330 for three graph unlearning scenarios in a unified way. Specifically, we compute ${r}'^{(t)}$ following Equation (2), while ${r}^{(t)}$ is zero for any layer $t$, as we could assume $r_{\rm max} = 0$ in the analysis. In this way, we derive bounds for $||( \mathbf{Z} - \mathbf{Z}')\mathbf{e}_i||$ in edge, node, and node feature unlearning scenarios. These results are presented in Lemmas E.5, E.6, and E.1, respectively.
> We believe this is an interesting and innovative aspect of our analysis and plan to further emphasize the differences between ScaleGUN and CGU in our revision.
>
> We hope the above explanation clarifies the difference between ScaleGUN and CGU in terms of the analysis and answers your question of how we overcome the challenges of a different choice of $\mathbf{P}$.

---

> ### Author Response · Authors · 2024-11-20
> **Response to Reviewer skqz (Part 2/2)**
>
> > W3. focus on the setting $\delta=1/n$ or $\delta=1/|E|$
>
> Thank you for the suggestion. We agree that setting $\delta=1/n$ or $\delta=1/|E|$ is more meaningful in the context of privacy. Based on this, we have conducted new experiments on node and feature unlearning with ogbn-papers100M. The experimental setting mirrors that of Table 2, except that we use $\delta=1/n$ instead. In our revision, we will replace Table 1 (results for edge unlearning) with Table 12, and replace Table 2 with the results presented below. Additionally, we will also update the results on Table 9 and Table 10 with $\delta=1/n$ or $\delta=1/|E|$ in the future version.
>
> The following table reports test accuracy (%), total unlearning cost (s) and propagation cost (s) per node feature removal for linear models on ogbn-papers100M:
>
> | $N (\times 10^3)$ | Retrain | ScaleGUN |
> |-|-|-|
> | 0 | 59.99   | 59.72    |
> | 2 | 59.99   | 59.72    |
> | 4 | 59.99   | 59.65    |
> | 6 | 59.99   | 59.47    |
> | 8 | 59.99   | 59.54    |
> | 10| 59.99   | 59.45    |
> | **Total** | 5400.45 | 45.29    |
> | **Prop**  | 5352.84 | 6.89     |
>
> The following table reports test accuracy (%), total unlearning cost (s) and propagation cost (s) per node removal for linear models on ogbn-papers100M:
>
> | $N (\times 10^3)$ | Retrain | ScaleGUN |
> |-|-|-|
> | 0 | 59.99   | 59.72    |
> | 2 | 59.99   | 59.75    |
> | 4 | 59.99   | 59.58    |
> | 6 | 59.99   | 59.80    |
> | 8 | 59.99   | 59.56    |
> | 10| 60.00   | 59.63    |
> | **Total** | 5201.88 | 60.08    |
> | **Prop**  | 5139.09 | 21.61    |

---

> > ### Comment · Reviewer_skqz · 2024-11-20
> >
> > I thank the authors for the response. I have no more questions and I will keep my score. Wish all the best to the authors!

---

> > > ### Author Response · Authors · 2024-11-25
> > >
> > > Thank you for your positive feedback. We greatly appreciate your recognition of our contributions and the constructive suggestions that have helped us refine our manuscript. Thank you again for your efforts, and we wish you all the best as well!

---

### Official Review · Reviewer_qZtw · 2024-11-04

**Soundness:** 3
**Presentation:** 3
**Contribution:** 2
**Rating:** 8
**Confidence:** 4

**Summary:**

This paper proposes ScaleGUN, a scalable certified unlearning framework for PPR-based GNNs. In particular, the authors propose a lazy local propagation mechanism to improve the efficiency of the forward pass. In a theoretical perspective, this paper proposes theoretical guarantees on the unlearning efficacy of the proposed lazy local propagation framework. Experimental results show that ScaleGUN largely reduces the latency for unlearning.

**Strengths:**

- The proposed ScaleGUN achieves an impressive speedup on some of the largest graph datasets while ensuring a theoretical guarantee.
- In the theoretical perspective, ScaleGUN extends previous certified unlearning for linear GNNs to PPR-based GNNs which includes a forward approximation error.
- The paper is overall well-organized. Technical details are introduced. Open-source codes are provided.

**Weaknesses:**

- The scope of ScaleGUN is still limited to linear GNNs. However, existing studies [1,2,3] have already extended certified graph unlearning to general GNN architectures. This limitation can reduce the significance of the proposed ScaleGUN.
- For the unlearning technique, ScaleGUN simply follows previous designs (based on influence function). The technical novelty is limited.
- Several latest certified graph unlearning baselines [1,3] are missing in the experiments.
- The noise variance is determined by some hyperparameters, e.g., $c_1$ and $\gamma_2$. How to determine the value of them in the practical scenarios?
- Some typos need to be fixed, such as the reference to CEU is incorrect.

[1] Wu, Jiancan, et al. "Gif: A general graph unlearning strategy via influence function." Proceedings of the ACM Web Conference 2023. 2023.

[2] Wu, Kun, et al. "Certified edge unlearning for graph neural networks." Proceedings of the 29th ACM SIGKDD Conference on Knowledge Discovery and Data Mining. 2023.

[3] Dong, Yushun, et al. "Idea: A flexible framework of certified unlearning for graph neural networks." Proceedings of the 30th ACM SIGKDD Conference on Knowledge Discovery and Data Mining. 2024.

**Questions:**

Please see the Weaknesses

---

> ### Author Response · Authors · 2024-11-20
> **Response to Reviewer qZtw (Part 1/3)**
>
> Thank you for these helpful comments. Our detailed answers are provided below.
>
> > W1. "*existing studies [1,2,3] have already extended certified graph unlearning to general GNN architectures*"
>
> We appreciate the reviewer's acknowledgment of relevant works. However, we believe there may be a misunderstanding regarding the contributions of [1,2,3]. They have **NOT** extended certified graph unlearning to general GNNs. Specifically,
>
> [1] GIF extends traditional influence functions to graph-structured data, proposing a graph unlearning mechanism. However, GIF does not offer certified guarantees, and the work does not claim any theorems on certified graph unlearning.
>
> [2] CEU proposes a certified graph unlearning mechanism for edge unlearning. However, its theoretical analysis relies on the strongly convex loss function assumption, similar to ScaleGUN. Consequently, CEU's certified guarantees are also restricted to linear models, as stated in Assumption 6 of the CEU paper.
>
> [3] IDEA provides a certified graph unlearning framework for several graph unlearning scenarios, but its theoretical analysis also depends on the strongly convex loss function assumption (Assumption 1 in the IDEA paper). As such, its certified guarantees are also restricted to linear models.
>
> **To the best of our knowledge, no existing certified graph unlearning methods can apply to general GNNs.** While our ultimate goal is to develop a scalable and certified graph unlearning mechanism for general GNNs and apply it to real-world applications, achieving certified unlearning guarantees for nonlinear models, including nonlinear GNNs, remains an open and highly nontrivial challenge. In this work, we focus on certified unlearning and prioritize scalability in order to make it applicable to real-world applications. We take the first step by providing scalable and certified graph unlearning for linear models. Additionally, ScaleGUN can serve as a scalable and effective unlearning mechanism for nonlinear models in scenarios where theoretical guarantees are not required. We elaborate on these points below.
>
> **Achieving certified unlearning for nonlinear models is highly nontrivial and beyond the scope of this paper.** Specifically, certified unlearning requires that the unlearned model is approximately equivalent to retraining in terms of their probability distributions (Line 127-131 of our manuscript). This indicates that the unlearned model must be able to approximate the new optimal point of empirical risk within a certain margin of error. However, existing studies[R1] have proven that even approximating the optimal value of a 2-layer ReLU neural network is NP-hard in worst cases. This demonstrates that it is nontrivial to achieve certified unlearning for nonlinear models and explains why existing certified unlearning studies are limited to linear models.
>
> **ScaleGUN aims at taking the first step toward making graph unlearning applicable to real-world applications.** While regulatory requirements for “the right to be forgotten” are driving research on graph unlearning, no existing methods scale to billion-edge graphs to our best knowledge, even without certified guarantees. This limitation hinders the applicability of graph unlearning in real-world scenarios (e.g., recommender systems, online social networks), as these applications often involve large-scale networks with billions of edges. As the first scalable and certified graph unlearning framework, ScaleGUN enables unlearning on billion-edge graphs and paves the way for graph unlearning to be applied in real-world applications.
>
> **ScaleGUN can also serve as a scalable and effective unlearning mechanism for nonlinear models in scenarios where theoretical guarantees are not required.** ScaleGUN achieves competitive performance on large graphs when applied to deep decoupled models (Table 3) and spectral GNNs (Table 14). Figure 6 further demonstrates that ScaleGUN can effectively unlearn graph data in nonlinear models, behaving similarly to retraining.
>
> Kindly note that we provided detailed discussions on the limitations of existing certified graph unlearning methods and potential improvements in Line 785-831 of our manuscript. We hope this clarifies your concerns and highlights the contributions of our work.
>
> [R1] Yifei Wang and Mert Pilanci. Polynomial-Time Solutions for ReLU Network Training: A Complexity Classification via Max-Cut and Zonotopes.

---

> ### Author Response · Authors · 2024-11-20
> **Response to Reviewer qZtw (Part 2/3)**
>
> > W2. "*For the unlearning technique, ScaleGUN simply follows previous designs (based on influence function)*"
>
> We respectfully disagree that ScaleGUN simply follows previous designs. Please note that simply adopting existing certified unlearning methods cannot achieve scalable and certified graph unlearning, either in terms of theoretical guarantees for certified unlearning or the practical implementation of unlearning algorithms. Since we integrate an approximate propagation algorithm to accelerate graph propagation, two key questions arise: 1. Will the approximation error introduced by approximate propagation lead to an unacceptably large model error? 2. What are the new model error bounds (both worst-case and data-dependent) after incorporating approximate propagation? These bounds are essential for selecting suitable privacy parameters in the algorithm, particularly the data-dependent bounds, which help guide decisions on when to retrain the model automatically during sequential unlearning. Without such bounds, it would be impossible to guarantee certified unlearning throughout the process.
>
> Through non-trivial theoretical analysis, we demonstrate that approximation error only marginally impacts the model error, while ensuring the total model error remains bounded. This critical insight allows us to achieve scalable and certified graph unlearning via approximate propagation. Moreover, we derive the worst-case and data-dependent bounds on model error in the context of approximate propagation. These finish the missing pieces in the puzzle of scalable and certified graph unlearning.
>
> Influence function[R2] is a heuristic technique that estimates how the model parameters change when a single data point is removed, but it does not provide theoretical guarantees for unlearning. It is important to note that existing certified graph unlearning techniques, such as CGU, CEU, and IDEA, as well as certified unlearning methods for unstructured data [R3], also originate from the influence function. One key contribution of these works is to provide theoretical guarantees for certified unlearning based on the influence function technique. ScaleGUN distinguishes itself by introducing approximate propagation to achieve scalability, which is absent in these prior methods. Without the non-trivial theoretical analysis we provide, it would not be possible to achieve scalable graph unlearning with certified guarantees by relying solely on influence functions. Additionally, as mentioned in our response to W2 from Reviewer skqz, the detailed analytical techniques in ScaleGUN are fundamentally different from existing certified unlearning approaches, too.
>
> [R2] Understanding Black-box Predictions via Influence Functions. Pang Wei Koh, Percy Liang.
>
> [R3] Certified Data Removal from Machine Learning Models. Chuan Guo, Tom Goldstein, Awni Hannun, Laurens van der Maaten.
>
> > W3. "*[1,3] are missing in the experiments*"
>
> The edge unlearning results of GIF [1] and IDEA [3] on the ogbn-arxiv and ogbn-products datasets are provided below. The experimental settings for GIF and IDEA are consistent with those used in Table 1. For comparison, the results of Retrain and ScaleGUN have been copied directly from Table 1.
>
> ogbn-arxiv:
> |N|Retrain|IDEA|GIF|ScaleGUN|
> |-|-|-|-|-|
> |0|57.83|57.82|57.83|57.84|
> |25|57.83|57.81|57.83|57.84|
> |50|57.82|57.81|57.82|57.83|
> |75|57.82|57.80|57.82|57.83|
> |100|57.81|57.80|57.81|57.82|
> |125|57.81|57.79|57.81|57.81|
> |Total|2.66|3.88|3.61|**0.91**|
> |Prop|1.73|1.66|1.61|**0.70**|
>
> ogbn-products:
> ||Retrain|IDEA|GIF|ScaleGUN|
> |-|-|-|-|-|
> |0|56.24|56.15|56.24|56.23|
> |1000|56.23|56.14|56.23|56.22|
> |2000|56.22|56.13|56.22|56.21|
> |3000|56.21|56.13|56.21|56.20|
> |4000|56.20|56.12|56.20|56.20|
> |5000|56.19|56.11|56.19|56.19|
> |Total|101.90|86.46|87.97|**8.76**|
> |Prop|98.48|83.74|85.32|**8.35**|
>
> These results demonstrate the following:
> 1. GIF achieves slightly better model accuracy, primarily because it does not introduce noise perturbation to ensure certified guarantees.
> 2. Both IDEA and GIF incur high propagation costs, similar to Retrain, as they require recomputing exact graph propagation for each unlearning request, which is time-intensive.
> 3. IDEA and GIF require higher unlearning costs (= Total - Prop) compared to ScaleGUN. This is likely because both IDEA and GIF rely on iterative algorithms to compute the influence function, whereas ScaleGUN employs conjugate gradient (CG) for large graphs and directly computes the inverse of the Hessian for small graphs, improving efficiency.
>
> Note that GIF does not provide certified guarantees, as mentioned in our reply to W1. Therefore, we did not include GIF as a baseline, consistent with other certified graph unlearning studies, such as CGU and CEU, which also exclude heuristic methods as baselines.
> IDEA is a recent work published on August 24, 2024, which is not incorporated in our initial submission. We will include IDEA in our revision.

---

> ### Author Response · Authors · 2024-11-20
> **Response to Reviewer qZtw (Part 3/3)**
>
> > W4. "*The noise variance is determined by some hyperparameters, e.g., $c_1$ and $\gamma_2$. How to determine the value of them in the practical scenarios?*"
>
> 1.  $c_1$ and $\gamma_2$
>
>     All the parameters in Assumption 4.1, including $c_1$ and $\gamma_2$, are the assumptions of the loss function's properties. Thus, they are determined by the loss function we adopted. We provide the exact values of these parameters for binary logistic regression problems with log-sigmoid loss in Line 745-751, Appendix B.1. These values are adopted in our experiments for linear models, too.
>
> 2. the noise variance
>
>    The noise variance is determined by the practical privacy requirements and the desired privacy budget. To determine the noise variance, we first decide the $(\epsilon, \delta)$-certified requirement (as defined in Line 127-131 of our manuscript). In differential privacy (DP) literature, $\delta$ is typically set to one over the number of relevant items in the dataset, such as $1/|\mathcal{E}|$ for edge-DP, while $\epsilon$ is usually set to 0.
>
>    Then, we determine the noise variance according to Theorem 2.1 and the desired privacy budget. Details on how the privacy budget works during sequential unlearning are provided in Line 390-398, Appendix B.2 and Algorithm 1. Intuitively, the privacy budget represents the amount of model error that can be tolerated. During sequential unlearning, each unlearning request incrementally increases the model error, thereby consuming the privacy budget. When the privacy budget is exhausted, retraining the model becomes necessary to satisfy $(\epsilon,\delta)$-certified unlearning requirements. The privacy budget is computed as $\alpha\epsilon\sqrt{2\log (1.5/\delta)}$, which means that a larger noise variance can reduce the frequency of retraining when fixing $\epsilon, \delta$.
>
> > W5. typos
>
> Thank you for spotting the typos. We will correct the citation of CEU in Line 409 in our revision.

---

> ### Comment · Reviewer_qZtw · 2024-11-23
>
> I appreciate the authors' efforts for further clarification. The theoretical contributions emphasized in the response make sense to me. For the experiments of deep nonlinear model (Figure 6), It would be helpful to add more details of the experimental settings. Overall, I believe this work can be a good start to bridge the gap between theoretically sound graph unlearning and practical use in the real world. Hence, I have raised my score.

---

> > ### Author Response · Authors · 2024-11-25
> >
> > We deeply appreciate your positive evaluation and recognition of our work. In response to your feedback, we have revised our manuscript to include additional details about the experimental settings for Figure 6, including the selection of adversarial edges and the formulation of the deep models employed. The description of Table 3 has also been revised accordingly for improved clarity. These updates have been highlighted in blue for your convenience.
> >
> > Your valuable comments and suggestions have greatly contributed to improving the clarity and quality of our manuscript. Thank you once again for your thoughtful review.

---

### Official Review · Reviewer_ABLM · 2024-11-04

**Soundness:** 3
**Presentation:** 4
**Contribution:** 3
**Rating:** 6
**Confidence:** 4

**Summary:**

This paper studies the certified graph unlearning problem with controlled propagation approximation to accelerate the unlearning speed over large-scale datasets. The main contribution of this paper comes from combining the ideas from forward push and graph unlearning with theoretical guarantees. Specifically, the forward push is designed to control the impact when we try to remove an edge or node feature from the graph dataset. Overall the idea of this paper is intuitive and the presentation is easy to follow. The empirical evaluations also show significant improvement in the unlearning efficiency compared to previous graph unlearning baselines when the model performance is comparable.

**Strengths:**

1. This paper aims to resolve one of the major bottlenecks of current certified graph unlearning methods, which is the high complexity of repropagating the features over the new graph topology. Approximate updates, adapted from dynamic PPR propagation techniques, are used to accelerate this repropagation process. To align with the certified (graph) unlearning analysis, the authors also provide theoretical analysis to bound the extra approximation error, validating that the privacy requirement is satisfied.

2. The empirical performance on large-scale datasets show promising gains in the unlearning efficiency while the model performance is comparable.

**Weaknesses:**

1. The linear model assumption is still needed with the new analysis. Also, most of the analysis (i.e., forward push part and graph unlearning part) seems similar to previous methods.

2. One of the key points, why updating the embeddings of a small local neighborhood is enough, is not rigorously explained in the main text. I am not sure the statement "Consider an edge removal scenario, for example, where edge (u, v) is targeted for removal. Only node u, node v, and their neighbors fail to meet Equation (2) due to the altered degrees of u and v." is enough here. Please see the Questions Section for more details.

**Questions:**

In Lemma 3.2 Eq (2), the authors claim that "Consider an edge removal scenario, for example, where edge (u, v) is targeted for removal. Only node u, node v, and their neighbors fail to meet Eq (2) due to the altered degrees of u and v." But I think besides the one-hop neighbors, more nodes could be affected thus violating the Eq (2). This is mainly because both $q^{l-1}(t)$ and $d(t)$ can change in this process, not just $d(t)$. For example, suppose there is a line connection between $a-b-c$, and we remove one edge incident to $a$. In this case, $a$ and $b$ would definitely violate Eq (2). But for $c$, since $q^{l-1}(b)$ would change, $q^{l}(c)$ and $r^{l}(c)$ would also need to adjust. I am wondering if the authors can explain the details here in a more rigorous way and maybe discuss how it will affect the following analysis.

---

> ### Author Response · Authors · 2024-11-20
> **Response to Reviewer ABLM (Part 1/3)**
>
> Thank you for these helpful comments. Our detailed answers are provided below.
> > W1-1. "*The linear model assumption is still needed with the new analysis.*"
>
> We agree that the linear model assumption is necessary for our current analysis. However, this is because our paper focuses on certified graph unlearning, where the linear model assumption is required to ensure certified guarantees. **To the best of our knowledge, no existing certified graph unlearning methods are applicable to general GNNs**. While our ultimate goal is to develop a scalable and certified graph unlearning mechanism for general GNNs and apply it to real-world applications, achieving certified guarantees for nonlinear models, including nonlinear GNNs, remains an open and highly nontrivial challenge. In this work, we prioritize scalability in order to make certified unlearning applicable to real-world applications and take the first step by providing scalable and certified graph unlearning for linear models. Additionally, when theoretical guarantees are not required, our proposed method can serve as a scalable and effective unlearning mechanism for nonlinear models. We elaborate on these points below.
>
> **Achieving certified unlearning for nonlinear models is highly nontrivial and beyond the scope of this paper.** Specifically, certified unlearning requires that the unlearned model is approximately equivalent to retraining in terms of their probability distributions (Line 127-131 of our manuscript). This indicates that the unlearned model must be able to approximate the new optimal point of empirical risk within a certain margin of error. However, existing studies[R1] have proven that even approximating the optimal value of a 2-layer ReLU neural network is NP-hard in worst cases. This demonstrates that it is nontrivial to achieve certified unlearning for nonlinear models and explains why existing certified unlearning studies are all limited to linear models.
>
> **ScaleGUN aims at taking the first step toward making graph unlearning applicable to real-world applications.** While regulatory requirements for “the right to be forgotten” are driving research on graph unlearning, no existing methods scale to billion-edge graphs to our best knowledge, even without certified guarantees. This limitation hinders the applicability of graph unlearning in real-world scenarios (e.g., recommender systems, online social networks), as these applications often involve large-scale networks with billions of edges. As the first scalable and certified graph unlearning framework, ScaleGUN enables unlearning on billion-edge graphs and paves the way for graph unlearning to be applied in real-world applications.
>
> **ScaleGUN can also serve as a scalable and effective unlearning mechanism for nonlinear models in scenarios where theoretical guarantees are not required.** ScaleGUN achieves competitive performance on large graphs when applied to deep decoupled models (Table 3) and spectral GNNs (Table 14). Figure 6 further demonstrates that ScaleGUN can effectively unlearn graph data in nonlinear models, behaving similarly to retraining.
>
> Kindly note that we provided detailed discussions on the limitations of existing certified graph unlearning methods and potential improvements in Line 785-831 of our manuscript. We hope this clarifies your concerns and highlights the contributions of our work.
>
> [R1] Yifei Wang and Mert Pilanci. Polynomial-Time Solutions for ReLU Network Training: A Complexity Classification via Max-Cut and Zonotopes.

---

> ### Author Response · Authors · 2024-11-20
> **Response to Reviewer ABLM (Part 2/3)**
>
> > W1-2. "*Also, most of the analysis (i.e., forward push part and graph unlearning part) seems similar to previous methods.*"
>
> In fact, the analysis of ScaleGUN is fundamentally different from previous methods in three key aspects:
> 1. the integration of approximate propagation into certified unlearning, which requires a non-trivial theoretical analysis to understand the impact of approximation error on model error. This is entirely absent in prior methods.
> 2. the unlearning part, which requires different techniques to derive bounds for model error due to the introduction of approximation error and our choice of the propagation matrix.
> 3. the propagation (forward push) part, which is designed for the GPR-based propagation scheme, whereas existing dynamic propagation methods rely on the PPR scheme.
>
> We elaborate on these differences below.
>
> **The integration of approximate propagation into certified unlearning:** Please note that simply combining existing approximate propagation and existing certified unlearning methods cannot achieve scalable and certified graph unlearning, either in terms of theoretical guarantees for certified unlearning or the practical implementation of unlearning algorithm. This integration raises two key questions: 1. Will the approximation error introduced by approximate propagation lead to an unacceptably large model error? 2. What are the new model error bounds (both worst-case and data-dependent) after incorporating approximate propagation? These bounds are essential for selecting suitable privacy parameters in the algorithm, particularly the data-dependent bounds, which guide decisions on when to retrain the model automatically during sequential unlearning. Without such bounds, it would be impossible to guarantee certified unlearning throughout the process.
>
> Through non-trivial theoretical analysis, we demonstrate that approximation error only marginally impacts the model error, while ensuring the total model error remains bounded. This critical insight allows us to achieve scalable and certified graph unlearning via approximate propagation. Moreover, we derive the worst-case and data-dependent bounds on model error in the context of approximate propagation. These finish the missing pieces in the puzzle of scalable and certified graph unlearning.
>
> **Unlearning part**: The difference in analysis between prior works and ScaleGUN arises from two key aspects: 1. ScaleGUN computes approximate embeddings, whereas prior works compute exact embeddings. However, exact embeddings are necessary to bound the model error, making this a new challenge for ScaleGUN. 2. ScaleGUN adopts the propagation matrix $\mathbf{P}=\mathbf{D}^{-1/2}\mathbf{A}\mathbf{D}^{-1/2}$, while prior works (i.e., CGU) adopt $\mathbf{P}=\mathbf{D}^{-1}\mathbf{A}$. The first challenge can be addressed through approximation error analysis, as explained in Lines 301–313 of our manuscript. Below, we focus on the second challenge in details.
>
> Choosing $\mathbf{P}=\mathbf{D}^{-1/2}\mathbf{A}\mathbf{D}^{-1/2}$ introduces additional difficulties in deriving bounds for $||\Delta||$ (a key component for bounding the model error, as defined in Line 258), requiring entirely different techniques compared to CGU. Specifically, the main challenge of bounding $||\Delta||$ lies in connecting the pre- and post-unlearning embeddings ($\mathbf{Z}$ and $\mathbf{Z}'$). In CGU, the authors choose to bound $||\mathbf{e}_i (\mathbf{Z} - \mathbf{Z}')||$, i.e., the difference between the pre- and post-unlearning embedding for each node $i$. Because CGU adopts $\mathbf{P} = \mathbf{D}^{-1}\mathbf{A}$, a right stochastic matrix, it benefits from useful properties to simplify the analysis. The CGU authors emphasize that this choice of $\mathbf{P}$ is critical, as other choices will lead to worse bounds of the intermediate results. Therefore, instead of bounding $||\mathbf{e}_i (\mathbf{Z} - \mathbf{Z}')||$ as in CGU, we leverage the local lazy framework to neatly bound $||(\mathbf{Z} - \mathbf{Z}')\mathbf{e}_j||$ (i.e., the difference in an embedding dimension pre- and post- unlearning). The details of this method are provided in Line 324-337 in our manuscript.
>
> **Propagation part**: As we mentioned in the response to Q3 from Reviewer N1R8, ScaleGUN adopts a GPR-based propagation scheme, whereas existing dynamic propagation methods rely on the PPR scheme. We choose GPR for its generalizability, alignment with existing unlearning methods, and enhanced expressive power. PPR-based methods can simply maintain a reserve vector $q$ and a residue vector $r$. In contrast, GPR-based propagation requires $q^{(\ell)}$ and $r^{(\ell)}$ for each level $\ell$. These differences introduce unique challenges to algorithm implementation, the theoretical analysis of correctness and time complexity. To our best knowledge, ScaleGUN provides the first dynamic propagation algorithm designed for GPR-based propagation scheme.

---

> ### Author Response · Authors · 2024-11-20
> **Response to Reviewer ABLM (Part 3/3)**
>
> > W2/Q. Details on lazy local propagation framework
>
> The update operation consists of two steps: 1. updating $r^{(\ell)}$ for any nodes that violate Eq (2). 2. performing local propagation to further reduce the error.
>
> A key point in this process is that we do not update $q^{(\ell)}$ in Step 1. Instead, only $r^{(\ell)}$ is updated for nodes $a$ and $b$. It is important to note that only $q^{(\ell)}(b)$ appears in node $c$'s Eq (2), while $r^{(\ell)}(b)$ does not. As a result, node $c$'s Eq (2) is not violated during Step 1. In Step 2, the propagation remains local, as we only perform a push operation for a node $u$ with $r^{(\ell)}(u) > r_{\rm max}$.
>
> Kindly note that the detailed update procedure is provided in Algorithm 3, where Line 1-10 correspond to Step 1 and Line 11 corresponds to Step 2. We hope this clarifies your question and we will add this explanation to the manuscript in our revision.

---

> ### Author Response · Authors · 2024-11-25
> **Summary and Looking Forward to Further Discussions**
>
> Thank you again for your great efforts and valuable comments. As the author-reviewer discussion phase comes to a close, we look forward to any additional feedback you may have. Below, we summarize our previous responses for your convenience:
>
> 1. W1-1: Regarding the linear model assumption, we have explained the necessity of this assumption for our current analysis and the challenges of extending certified graph unlearning to general GNNs. We have also emphasized ScaleGUN’s contributions in bridging the gap between existing theoretical analyses and the real-world necessity for scalability, as well as its applicability to nonlinear models when theoretical guarantees are not required.
> 2. W1-2: Regarding the differences in analysis between ScaleGUN and prior works, we have highlighted the unique challenges ScaleGUN addresses to achieve scalable and certified graph unlearning, focusing on three key aspects, particularly the integration of approximate propagation into certified unlearning.
> 3. Q1: Regarding the details of the lazy local propagation framework, we have provided a detailed explanation of the update operation.
>
> If you have any further questions or require additional clarification, we welcome continued discussion and engagement.

---

> > ### Comment · Reviewer_ABLM · 2024-11-29
> >
> > I appreciate the authors' efforts for the detailed response and summary. I am more clear about the paper contribution now. I would keep the score and vote for accepting the paper.

---

### Comment · Area_Chair_x9RB · 2024-11-28

I would like to encourage the reviewers to engage with the author's replies if they have not already done so. At the very least, please
acknowledge that you have read the rebuttal.

---

### Meta-Review · Area_Chair_x9RB · 2024-12-19

**Metareview:**

This paper introduces ScaleGUN, a scalable and certifiable graph unlearning model. It overcomes scalabilit challenges by integrating approximate graph propagation (a la forward push), while ensuring bounded approximation errors. The authors tackle various unlearning scenarios, including edge and node feature removal. Experimental evidence on large graphs such ogbn-papers100M validate the scalability claims. The method is restricted to linear models, but this limitation is shared by all existing baselines. No existing methods achieve similar scalability, even without guarantees.

**Additional Comments On Reviewer Discussion:**

Doubts about some of the theoretical results were resolved after the authors' response. Some reviewers mentioned that the analysis seems similar to previous methods and that the integration of approximate propagation and unlearning seems incremental. However, the authors successfully clarified the unique challenges that they address. In general, most of the concerns raised by the reviewers were addressed, and subsequently two reviewers raised their score.

---

### Decision · Program_Chairs · 2025-01-22

Accept (Spotlight)